# A Finite-Time Analysis of Two Time-Scale Actor-Critic Methods

**Yue Wu**
Department of Computer Science
University of California, Los Angeles
Los Angeles, CA 90095
ywu@cs.ucla.edu

**Weitong Zhang**
Department of Computer Science
University of California, Los Angeles
Los Angeles, CA 90095
weightzero@cs.ucla.edu

**Pan Xu**
Department of Computer Science
University of California, Los Angeles
Los Angeles, CA 90095
panxu@cs.ucla.edu

**Quanquan Gu**
Department of Computer Science
University of California, Los Angeles
Los Angeles, CA 90095
qgu@cs.ucla.edu

## Abstract

Actor-critic (AC) methods have exhibited great empirical success compared with other reinforcement learning algorithms, where the actor uses the policy gradient to improve the learning policy and the critic uses temporal difference learning to estimate the policy gradient. Under the two time-scale learning rate schedule, the asymptotic convergence of AC has been well studied in the literature. However, the non-asymptotic convergence and finite sample complexity of actor-critic methods are largely open. In this work, we provide a non-asymptotic analysis for two time-scale actor-critic methods under non-i.i.d. setting. We prove that the actor-critic method is guaranteed to find a first-order stationary point (i.e., $\|\nabla J(\boldsymbol{\theta})\|_2^2 \leq \epsilon$) of the non-concave performance function $J(\boldsymbol{\theta})$, with $\widetilde{\mathcal{O}}(\epsilon^{-2.5})$ sample complexity. To the best of our knowledge, this is the first work providing finite-time analysis and sample complexity bound for two time-scale actor-critic methods.

## 1 Introduction

Actor-Critic (AC) methods [2, 16] aim at combining the advantages of actor-only methods and critic-only methods, and have achieved great empirical success in reinforcement learning [31, 1]. Specifically, actor-only methods, such as policy gradient [28] and trust region policy optimization [24], utilize a parameterized policy function class and improve the policy by optimizing the parameters of some performance function using gradient ascent, whose exact form is characterized by the Policy Gradient Theorem [28]. Actor-only methods can be naturally applied to continuous setting but suffer from high variance when estimating the policy gradient. On the other hand, critic-only methods, such as temporal difference learning [26] and Q-learning [32], focus on learning a value function (expected cumulative rewards), and determine the policy based on the value function, which is recursively approximated based on the Bellman equation. Although the critic-only methods can efficiently learn a satisfying policy under tabular setting [14], they can diverge with function approximation under continuous setting [33]. Therefore, it is natural to combine actor and critic based methods to achieve the best of both worlds. The principal idea behind actor-critic methods is simple: the critic tries to learn the value function, given the policy from the actor, while the actor can estimate the policy gradient based on the approximate value function provided by the critic.

If the actor is fixed, the policy remains unchanged throughout the updates of the critic. Thus one can use policy evaluation algorithm such as temporal difference (TD) learning [27] to estimate the value function (critic). After many steps of the critic update, one can expect a good estimation of the value function, which in turn enables an accurate estimation of the policy gradient for the actor. A more favorable implementation is the so-called two time-scale actor-critic algorithm, where the actor and the critic are updated simultaneously at each iteration except that the actor changes more slowly (with a small step size) than the critic (with a large step size). In this way, one can hope the critic will be well approximated even after one step of update. From the theoretical perspective, the asymptotic analysis of two time-scale actor-critic methods has been established in [6, 16]. In specific, under the assumption that the ratio of the two time-scales goes to infinity (i.e. $\lim_{t\to\infty} \beta_t/\alpha_t = \infty$), the asymptotic convergence is guaranteed through the lens of the two time-scale ordinary differential equations(ODE), where the slower component is fixed and the faster component converges to its stationary point. This type of analysis was also applied in the context of generic two time-scale stochastic approximation [5].

However, finite-time analysis (non-asymptotic analysis) of two-time scale actor-critic is still largely missing in the literature, which is important because it can address the questions that how many samples are needed for two time-scale actor-critic to converge, and how to appropriately choose the different learning rates for the actor and the critic. Some recent work has attempted to provide the finite-time analysis for the "decoupled" actor-critic methods [18, 23]. The term "decoupled" means that before updating the actor at the $t$-th iteration, the critic starts from scratch to estimate the state-value (or Q-value) function. At each iteration, the "decoupled" setting requires the critic to perform multiple sampling and updating (often from another new sample trajectory). As we will see in the later comparison, this setting is sample-inefficient or even impractical. Besides, their analyses are based on either the i.i.d. assumption [18] or the partially i.i.d. assumption [23] (the actor receives i.i.d. samples), which is unrealistic in practice. In this paper, we present the first finite-time analysis on the convergence of the two time-scale actor-critic algorithm. We summarize our contributions as follows:

- We prove that, the actor in the two time-scale actor critic algorithm converges to an $\epsilon$-approximate stationary point of the non-concave performance function $J$ after accessing at most $\widetilde{\mathcal{O}}(\epsilon^{-2.5})$ samples. Compared with existing finite-time analysis of actor-critic methods [18, 23], the algorithm we analyzed is based on two time-scale update and therefore more practical and efficient than the "decoupled" version. Moreover, we do not need any i.i.d. data assumptions in the convergence analysis as required by Kumar et al. [18], Qiu et al. [23], which do not hold in real applications.
- From the technical viewpoint, we also present a new proof framework that can tightly characterize the estimation error in two time-scale algorithms. Compared with the proof technique used in [38], we remove the extra artificial factor $\mathcal{O}(t^\xi)$ in the convergence rate introduced by their "iterative refinement" technique. Therefore, our new proof technique may be of independent interest for analyzing the convergence of other two time-scale algorithms to get sharper rates.

**Notation** We use lower case letters to denote scalars, and use lower and upper case bold face letters to denote vectors and matrices respectively. For two sequences $\{a_n\}$ and $\{b_n\}$, we write $a_n = \mathcal{O}(b_n)$ if there exists an absolute constant $C$ such that $a_n \leq Cb_n$. We use $\widetilde{\mathcal{O}}(\cdot)$ to further hide logarithm factors. Without other specification, $\|\cdot\|$ denotes the $\ell_2$ norm of Euclidean vectors. $d_{TV}(P, Q)$ is the total variation norm between two probability measure $P$ and $Q$, which is defined as $d_{TV}(P, Q) = 1/2 \int_{\mathcal{X}} |P(dx) - Q(dx)|$.

## 2 Related work

In this section, we briefly review and discuss existing work, which is mostly related to ours.

**Stochastic bias characterization** The main difficulty in analyzing reinforcement learning algorithms under non-i.i.d. data assumptions is that the samples and the trainable parameters are correlated, which makes the noise term biased. Bhandari et al. [3] used information-theoretical techniques to bound the Markovian bias and provide a simple and explicit analysis for the temporal difference learning. Similar techniques were also established in [25] through the lens of stochastic approximation methods. Gupta et al. [12], Xu et al. [38] applied such methods to deriving the non-asymptotic convergence of two time-scale temporal difference learning algorithms (TDC). Zou et al. [44], Chen et al. [10], Xu and Gu [35] further applied these analysis methods to on-policy learning algorithms including SARSA

and Q-learning. In addition, Hu and Syed [13] formulated a family of TD learning algorithms as a Markov jump linear systems and analyzed the evolution of the mean and covariance matrix of the estimation error. Cai et al. [7] studied TD learning with neural network approximation, and proved its global convergence.

**Two time-scale reinforcement learning** The two time-scale stochastic approximation can be seen as a general framework for analyzing reinforcement learning [5, 29, 17]. Recently, the finite-time analysis of two time-scale stochastic approximation has gained much interest. Dalal et al. [11] proved convergence rate for the two time-scale linear stochastic approximation under i.i.d. assumption. Gupta et al. [12] also provided finite-time analysis for the two time-scale linear stochastic approximation algorithms. Both can be applied to analyze two time-scale TD methods like GTD, GTD2 and TDC. Xu et al. [38] proved convergence rate and sample complexity for the TDC algorithm over Markovian samples. [15] further improved the convergence rate of two time-scale linear stochastic approximation and removed the projection step. However, since the update rule for the actor is generally not linear, we cannot apply these results to the actor-critic algorithms.

**Analysis for actor-critic methods** The asymptotic analysis of actor-critic methods has been well established. Konda and Tsitsiklis [16] proposed the actor-critic algorithm, and established the asymptotic convergence for the two time-scale actor-critic, with TD($\lambda$) learning-based critic. Bhatnagar et al. [4] proved the convergence result for the original actor-critic and natural actor-critic methods. Castro and Meir [8] proposed a single time-scale actor-critic algorithm and proved its convergence. Recently, [43] proved convergence of two time-scale off-policy actor-critic with function approximation. Recently, there has emerged some works concerning the finite-time behavior of actor-critic methods. Yang et al. [41] studied the global convergence of actor-critic algorithms under the Linear Quadratic Regulator. Yang et al. [40] analyzed the finite-sample performance of batched actor-critic, where all samples are assumed i.i.d. and the critic performs several empirical risk minimization (ERM) steps. Qiu et al. [23] treated the actor-critic algorithms as a bilevel optimization problem and established a finite sample analysis under the "average-reward" setting, assuming that the actor has access to independent samples. Similar result has also been established by Kumar et al. [18], where they considered the sample complexity for the "decoupled" actor-critic methods under i.i.d. assumption. Wang et al. [30] also proved the global convergence of actor-critic algorithms with both actor and critic being approximated by overparameterized neural networks.

When we were preparing this work, we noticed that there is a concurrent and independent work [39] which also analyzes the non-asymptotic convergence of two time-scale actor-critic algorithms and achieves the same sample complexity, i.e., $\widetilde{\mathcal{O}}(\epsilon^{-2.5})$. However, there are two key differences between their work and ours. First, the two time-scale algorithms analyzed in both papers are very different. We analyze the classical two time-scale algorithm described in [27], where both actor and critic take one step update in each iteration. It is very easy to implement and has been widely used in practice, while the update rule in [39] for the critic needs to call a sub-algorithm, which involves generating a fresh episode to estimate the Q-function. Second, the analysis in [39] relies on the compatible function approximation [28], which requires the critic to be a specific linear function class, while our analysis does not require such specific approximation, and therefore is more general. This makes our analysis potentially extendable to non-linear function approximation such as neural networks [7].

## 3 Preliminaries

In this section, we present the background of the two time-scale actor-critic algorithm.

### 3.1 Markov decision processes

Reinforcement learning tasks can be modeled as a discrete-time Markov Decision Process (MDP) $\mathcal{M} = \{\mathcal{S}, \mathcal{A}, \mathcal{P}, r\}$, where $\mathcal{S}$ and $\mathcal{A}$ are the state and action spaces respectively. In this work we consider the finite action space $|\mathcal{A}| < \infty$. $\mathcal{P}(s'|s, a)$ is the transition probability that the agent transits to state $s'$ after taking action $a$ at state $s$. Function $r : \mathcal{S} \times \mathcal{A} \to [-U_r, U_r]$ emits a bounded reward after the agent takes action $a$ at state $s$, where $U_r > 0$ is a constant. A policy parameterized by $\boldsymbol{\theta}$ at state $s$ is a probability function $\pi_{\boldsymbol{\theta}}(a|s)$ over action space $\mathcal{A}$. $\mu_{\boldsymbol{\theta}}$ denotes the stationary distribution induced by the policy $\pi_{\boldsymbol{\theta}}$.

In this work we consider the "average reward" setting [28], where under the ergodicity assumption, the average reward over time eventually converges to the expected reward under the stationary

distribution:

$$r(\boldsymbol{\theta}) := \lim_{N \to \infty} \frac{\sum_{t=0}^{N} r(s_t, a_t)}{N} = \mathbb{E}_{s \sim \mu_{\boldsymbol{\theta}}, a \sim \pi_{\boldsymbol{\theta}}} \big[ r(s,a) \big].$$

To evaluate the overall rewards given a starting state $s_0$ and the behavior policy $\pi_{\boldsymbol{\theta}}$, we define the state-value function as

$$V^{\pi_{\boldsymbol{\theta}}}(\cdot) := \mathbb{E}\left[ \sum_{t=0}^{\infty} \big( r(s_t, a_t) - r(\boldsymbol{\theta}) \big) | s_0 = \cdot \right],$$

where the action follows the policy $a_t \sim \pi_{\boldsymbol{\theta}}(\cdot|s_t)$ and the next state follows the transition probability $s_{t+1} \sim \mathcal{P}(\cdot|s_t, a_t)$. Another frequently used function is the state-action value function, also called Q-value function:

$$Q^{\pi_{\boldsymbol{\theta}}}(s,a) := \mathbb{E}\left[ \sum_{t=0}^{\infty} \big( r(s_t, a_t) - r(\boldsymbol{\theta}) \big) | s_0 = s, a_0 = a \right]$$
$$= r(s,a) - r(\boldsymbol{\theta}) + \mathbb{E}\big[ V^{\pi_{\boldsymbol{\theta}}}(s') \big],$$

where the expectation is taken over $s' \sim \mathcal{P}(\cdot|s, a)$.

Throughout this paper, we use $O$ to denote the tuple $O = (s, a, s')$, some variants are like $O_t = (s_t, a_t, s_{t+1})$ and $\widetilde{O}_t = (\widetilde{s}_t, \widetilde{a}_t, \widetilde{s}_{t+1})$.

### 3.2 Policy gradient theorem

We define the performance function associated with policy $\pi_{\boldsymbol{\theta}}$ naturally as the expected reward under the stationary distribution $\mu_{\boldsymbol{\theta}}$ induced by $\pi_{\boldsymbol{\theta}}$, which takes the form

$$J(\boldsymbol{\theta}) := r(\boldsymbol{\theta}). \tag{3.1}$$

To maximize the performance function with respect to the policy parameters, Sutton et al. [28] proved the following policy gradient theorem.

**Lemma 3.1** (Policy Gradient). Consider the performance function defined in (3.1), its gradient takes the form

$$\nabla J(\boldsymbol{\theta}) = \mathbb{E}_{s \sim \mu_{\boldsymbol{\theta}}(\cdot)} \left[ \sum_{a \in \mathcal{A}} Q^{\pi_{\boldsymbol{\theta}}}(s,a) \nabla \pi(a|s) \right].$$

The policy gradient also admits a neat form in expectation:

$$\nabla J(\boldsymbol{\theta}) = \mathbb{E}_{s \sim \mu_{\boldsymbol{\theta}}(\cdot), a \sim \pi_{\boldsymbol{\theta}}(\cdot|s)} \big[ Q^{\pi_{\boldsymbol{\theta}}}(s,a) \nabla \log \pi_{\boldsymbol{\theta}}(a|s) \big].$$

A typical way to estimate the policy gradient $\nabla J(\boldsymbol{\theta})$ is by Monte Carlo method, namely using the summed return along the trajectory as the estimated Q-value, which is known as the "REINFORCE" method [34].

**Remark 3.2.** The problem formulation in this paper is what Sutton et al. [28] had defined as "average-reward" formulation. An alternative formulation is the "start-state" formulation, which avoids estimating the average reward, but gives a more complicated form for the policy-gradient algorithm and the AC algorithm.

### 3.3 REINFORCE with a baseline

Note that for any function $b(s)$ depending only on the state, which is usually called "baseline" function, we have

$$\sum_{a \in \mathcal{A}} b(s) \nabla \pi_{\boldsymbol{\theta}}(a|s) = b(s) \nabla \left( \sum_{a \in \mathcal{A}} \pi_{\boldsymbol{\theta}}(a|s) \right) = 0.$$

So we also have

$$\nabla J(\boldsymbol{\theta}) = \mathbb{E}\left[ \sum_{a \in \mathcal{A}} \big( Q^{\pi_{\boldsymbol{\theta}}}(s,a) - b(s) \big) \nabla \pi_{\boldsymbol{\theta}}(a|s) \right].$$

A popular choice of $b(s)$ is $b(s) = V^{\pi_\theta}(s)$ and $\Delta^{\pi_\theta}(s, a) = Q^{\pi_\theta}(s, a) - V^{\pi_\theta}(s)$ is viewed as the advantage of taking a specific action $a$, compared with the expected reward at state $s$. Also note that the expectation form still holds:

$$\nabla J(\boldsymbol{\theta}) = \mathbb{E}_{s,a} \big[ \Delta^{\pi_\theta}(s, a) \nabla \log \pi_{\boldsymbol{\theta}}(a|s) \big].$$

Based on this fact, Williams [34] also proposed a corresponding policy gradient algorithm named "REINFORCE with a baseline" which performs better due to the reduced variance.

In practice the policy gradient method could suffer from high variance. An alternative approach is to introduce another trainable model to approximate the state-value function, which is called the actor-critic methods.

### 3.4  The two time-scale actor-critic algorithm

In previous subsection, we have seen how the policy gradient theorem appears in the form of the advantage value instead of the Q-value. Assume the critic uses linear function approximation $\widehat{V}(\cdot; \boldsymbol{\omega}) = \boldsymbol{\phi}^\top(\cdot)\boldsymbol{\omega}$, and is updated by TD(0) algorithm, then this gives rise to Algorithm 1 that we are going to analyze.

Algorithm 1 has been proposed in many literature, and is clearly introduced in [27] as a classic on-line one-step actor-critic algorithm. It uses the advantage (namely temporal difference error) to update the critic and the actor simultaneously. Based on its on-line nature, this algorithm can be implemented both under episodic and continuing setting. In practice, the asynchronous variant of this algorithm, called Asynchronous Advantage Actor-Critic(A3C), is an empirically very successful parallel actor-critic algorithm.

Sometimes, Algorithm 1 is also called Advantage Actor-Critic (A2C) because it is the synchronous version of A3C and the name indicates its use of advantage instead of Q-value [20].

---

**Algorithm 1** Two Time-Scale Actor-Critic

1: **Input:** initial actor parameter $\boldsymbol{\theta}_0$, initial critic parameter $\boldsymbol{\omega}_0$, initial average reward estimator $\eta_0$, step size $\alpha_t$ for actor, $\beta_t$ for critic and $\gamma_t$ for the average reward estimator.
2: Draw $s_0$ from some initial distribution
3: **for** $t = 0, 1, 2, \ldots$ **do**
4:      Take the action $a_t \sim \pi_{\boldsymbol{\theta}_t}(\cdot|s_t)$
5:      Observe next state $s_{t+1} \sim \mathcal{P}(\cdot|s_t, a_t)$ and the reward $r_t = r(s_t, a_t)$
6:      $\delta_t = r_t - \eta_t + \boldsymbol{\phi}(s_{t+1})^\top \boldsymbol{\omega}_t - \boldsymbol{\phi}(s_t)^\top \boldsymbol{\omega}_t$
7:      $\eta_{t+1} = \eta_t + \gamma_t(r_t - \eta_t)$
8:      $\boldsymbol{\omega}_{t+1} = \Pi_{R_\omega}\big(\boldsymbol{\omega}_t + \beta_t \delta_t \boldsymbol{\phi}(s_t)\big)$
9:      $\boldsymbol{\theta}_{t+1} = \boldsymbol{\theta}_t + \alpha_t \delta_t \nabla_{\boldsymbol{\theta}} \log \pi_{\boldsymbol{\theta}_t}(a_t|s_t)$
10: **end for**

---

In Line 6 of Algorithm 1, the temporal difference error $\delta_t$ can be calculated based on the critic's estimation of the value function $\boldsymbol{\phi}(\cdot)^\top \boldsymbol{\omega}_t$, where $\boldsymbol{\omega}_t \in \mathbb{R}^d$ and $\phi(\cdot) : \mathcal{S} \to \mathbb{R}^d$ is a known feature mapping. Then the critic will be updated using the semi-gradient from TD(0) method. Line 8 in Algorithm 1 also contains a projection operator. This is required to control the algorithm's convergence which also appears in some other literature [3, 38]. The actor uses the advantage $\delta_t$ (estimated by critic) and the samples to get an estimation of the policy gradient.

Algorithm 1 is more general and practical than the algorithms analyzed in many previous work [23, 18]. In our algorithm, there is no need for independent samples or samples from the stationary distribution. There is only one naturally generated sample path. Also, the critic inherits from last iteration and continuously updates its parameter, without requiring a restarted sample path (or a new episode).

## 4  Main theory

In this section, we first discuss on some standard assumptions used in the literature for deriving the convergence of reinforcement learning algorithms and then present our theoretical results for two time-scale actor-critic methods.

## 4.1 Assumptions and propositions

We consider the setting where the critic uses TD [27] with linear function approximation to estimate the state-value function, namely $\widehat{V}(\cdot; \boldsymbol{\omega}) = \boldsymbol{\phi}^\top(\cdot)\boldsymbol{\omega}$. We assume that the feature mapping has bounded norm $\|\boldsymbol{\phi}(\cdot)\| \leq 1$. Denote by $\boldsymbol{\omega}^*(\boldsymbol{\theta})$ the limiting point of TD(0) algorithms under the behavior policy $\pi_{\boldsymbol{\theta}}$, and define $\mathbf{A}$ and $\mathbf{b}$ as:

$$\mathbf{A} := \mathbb{E}_{s,a,s'}\big[\boldsymbol{\phi}(s)\big(\boldsymbol{\phi}(s') - \boldsymbol{\phi}(s)\big)^\top\big],$$
$$\mathbf{b} := \mathbb{E}_{s,a,s'}[(r(s,a) - r(\boldsymbol{\theta}))\boldsymbol{\phi}(s)],$$

where $s \sim \mu_{\boldsymbol{\theta}}(\cdot), a \sim \pi_{\boldsymbol{\theta}}(\cdot|s), s' \sim \mathcal{P}(\cdot|s,a)$. It is known that the TD limiting point satisfies:

$$\mathbf{A}\boldsymbol{\omega}^*(\boldsymbol{\theta}) + \mathbf{b} = \mathbf{0}.$$

In the sequel, when there is no confusion, we will use a shorthand notation $\boldsymbol{\omega}^*$ to denote $\boldsymbol{\omega}^*(\boldsymbol{\theta})$. Based on the complexity of the feature mapping, the approximation error of this function class can vary. The approximation error of the linear function class is defined as follows:

$$\epsilon_{\text{app}}(\boldsymbol{\theta}) := \sqrt{\mathbb{E}_{s \sim \mu_{\boldsymbol{\theta}}}\big(\boldsymbol{\phi}(s)^\top \boldsymbol{\omega}^*(\boldsymbol{\theta}) - V^{\pi_{\boldsymbol{\theta}}}(s)\big)^2}.$$

Throughout this paper, we assume the approximation error for all potential policies is uniformly bounded,

$$\forall \boldsymbol{\theta}, \epsilon_{\text{app}}(\boldsymbol{\theta}) \leq \epsilon_{\text{app}},$$

for some constant $\epsilon_{\text{app}} \geq 0$.

In the analysis of TD learning, the following assumption is often made to ensure the uniqueness of the limiting point of TD and the problem's solvability.

**Assumption 4.1.** For all potential policy parameters $\boldsymbol{\theta}$, the matrix $\mathbf{A}$ defined above is negative definite and has the maximum eigenvalues as $-\lambda$.

Assumption 4.1 is often made to guarantee the problem's solvability [3, 44, 38]. Note that Algorithm 1 contains a projection step at Line 8. To guarantee convergence it is required all $\boldsymbol{\omega}^*$ lie within this projection radius $R_{\boldsymbol{\omega}}$. Assumption 4.1 indicates that a sufficient condition is to set $R_{\omega} = 2U_r/\lambda$ because $\|\mathbf{b}\| \leq 2U_r$ and $\|\mathbf{A}^{-1}\| \leq \lambda^{-1}$.

The next assumption, first adopted by Bhandari et al. [3] in TD learning, addresses the issue of Markovian noise.

**Assumption 4.2** (Uniform ergodicity)**.** For a fixed $\boldsymbol{\theta}$, denote $\mu_{\boldsymbol{\theta}}(\cdot)$ as the stationary distribution induced by the policy $\pi_{\boldsymbol{\theta}}(\cdot|s)$ and the transition probability measure $\mathcal{P}(\cdot|s,a)$. Consider a Markov chain generated by the rule $a_t \sim \pi_{\boldsymbol{\theta}}(\cdot|s_t), s_{t+1} \sim \mathcal{P}(\cdot|s_t, a_t)$. Then there exists $m > 0$ and $\rho \in (0,1)$ such that:

$$d_{TV}\big(\mathbb{P}(s_\tau \in \cdot|s_0 = s), \mu_{\boldsymbol{\theta}}(\cdot)\big) \leq m\rho^\tau, \forall \tau \geq 0, \forall s \in \mathcal{S}.$$

We also need some regularity assumptions on the policy.

**Assumption 4.3.** Let $\pi_{\boldsymbol{\theta}}(a|s)$ be a policy parameterized by $\boldsymbol{\theta}$. There exist constants $L, B, L_l > 0$ such that for all given state $s$ and action $a$ it holds

(a) $\big\|\nabla \log \pi_{\boldsymbol{\theta}}(a|s)\big\| \leq B, \forall \boldsymbol{\theta} \in \mathbb{R}^d$,

(b) $\big\|\nabla \log \pi_{\boldsymbol{\theta}_1}(a|s) - \nabla \log \pi_{\boldsymbol{\theta}_2}(a|s)\big\| \leq L_l\|\boldsymbol{\theta}_1 - \boldsymbol{\theta}_2\|, \forall \boldsymbol{\theta}_1, \boldsymbol{\theta}_2 \in \mathbb{R}^d$,

(c) $\big|\pi_{\boldsymbol{\theta}_1}(a|s) - \pi_{\boldsymbol{\theta}_2}(a|s)\big| \leq L\|\boldsymbol{\theta}_1 - \boldsymbol{\theta}_2\|, \forall \boldsymbol{\theta}_1, \boldsymbol{\theta}_2 \in \mathbb{R}^d$.

The first two inequalities are regularity conditions to guarantee actor's convergence in the literature of policy gradient [22, 42, 18, 36, 37]. The last inequality in Assumption 4.3 is also adopted by Zou et al. [44] when analyzing SARSA.

An important fact arises from our assumptions is that the limiting point $\boldsymbol{\omega}^*$ of TD(0), which can be viewed as a mapping of the policy's parameter $\boldsymbol{\theta}$, is Lipschitz.

**Proposition 4.4.** Under Assumptions 4.1 and 4.2, there exists a constant $L_* > 0$ such that

$$\left\|\boldsymbol{\omega}^*(\boldsymbol{\theta}_1) - \boldsymbol{\omega}^*(\boldsymbol{\theta}_2)\right\| \leq L_* \|\boldsymbol{\theta}_1 - \boldsymbol{\theta}_2\|, \forall \boldsymbol{\theta}_1, \boldsymbol{\theta}_2 \in \mathbb{R}^d.$$

Proposition 4.4 states that the target point $\boldsymbol{\omega}^*$ moves slowly compared with the actor's update on $\boldsymbol{\theta}$. This is an observation pivotal to the two time-scale analysis. Specifically, the two time-scale analysis can be informally described as "the actor moves slowly while the critic chases the slowly moving target determined by the actor".

Now we are ready to present the convergence result of two time-scale actor-critic methods. We first define an integer that depends on the learning rates $\alpha_t$ and $\beta_t$.

$$\tau_t := \min\left\{i \geq 0 | m\rho^{i-1} \leq \min\{\alpha_t, \beta_t\}\right\}, \tag{4.1}$$

where $m, \rho$ are defined as in Assumption 4.2. By definition, $\tau_t$ is a mixing time of an ergodic Markov chain. We will use $\tau_t$ to control the Markovian noise encountered in the training process.

## 4.2 Convergence of the actor

At the $k$-th iteration of the actor's update, $\boldsymbol{\omega}_k$ is the critic parameter estimated by Line 7 of Algorithm 1 and $\boldsymbol{\omega}_k^*$ is the unknown parameter of value function $V^{\pi_{\boldsymbol{\theta}_k}}(\cdot)$ defined in Assumption 4.1. The following theorem gives the convergence rate of the actor when the averaged mean squared error between $\boldsymbol{\omega}_k$ and $\boldsymbol{\omega}_k^*$ and the error between $\eta_k$ and $r(\boldsymbol{\theta}_k)$ from $k = \tau_t$ to $k = t$ are small.

**Theorem 4.5.** Suppose Assumptions 4.1-4.3 hold and we choose $\alpha_t = c_\alpha/(1+t)^\sigma$ in Algorithm 1, where $\sigma \in (0,1)$ and $c_\alpha > 0$ are constants. If we assume at the $t$-th iteration, the critic satisfies

$$\frac{8}{t}\sum_{k=1}^{t}\mathbb{E}\|\boldsymbol{\omega}_k - \boldsymbol{\omega}_k^*\|^2 + \frac{2}{t}\sum_{k=1}^{t}\mathbb{E}\big(\eta_k - r(\boldsymbol{\theta}_k)\big)^2 = \mathcal{E}(t), \tag{4.2}$$

where $\mathcal{E}(t)$ is a bounded sequence, then we have

$$\min_{0 \leq k \leq t}\mathbb{E}\left\|\nabla J(\boldsymbol{\theta}_k)\right\|^2 = \mathcal{O}(\epsilon_{\text{app}}) + \mathcal{O}\left(\frac{1}{t^{1-\sigma}}\right) + \mathcal{O}\left(\frac{\log^2 t}{t^\sigma}\right) + \mathcal{O}\big(\mathcal{E}(t)\big),$$

where $\mathcal{O}(\cdot)$ hides constants, whose exact forms can be found in the detailed proof in Appendix C.1.

Note that $\mathcal{E}(t)$ in Theorem 4.5 is the averaged estimation error made by the critic throughout the learning process, which will be bounded in the next Theorem 4.7.

**Remark 4.6.** Theorem 4.5 recovers the results for the decoupled case [23, 18] by setting $\sigma = 1/2$. Nevertheless, we are considering a much more practical and challenging case where the actor and critic are simultaneously updated under Markovian noises. It is worth noting that the non-i.i.d. data assumption leads to an additional logarithm term, which is also observed in [3, 44, 25, 10].

## 4.3 Convergence of the critic

The condition in (4.2) is guaranteed by the following theorem that characterizes the convergence of the critic.

**Theorem 4.7.** Suppose Assumptions 4.1-4.3 hold and we choose $\alpha_t = c_\alpha/(1+t)^\sigma$ and $\beta_t = c_\beta/(1+t)^\nu$ in Algorithm 1, where $0 < \nu < \sigma < 1$, $c_\alpha$ and $c_\beta \leq \lambda^{-1}$ are positive constants. Then we have

$$\frac{1}{1+t-\tau_t}\sum_{k=\tau_t}^{t}\mathbb{E}\|\boldsymbol{\omega}_k - \boldsymbol{\omega}_k^*\|^2 = \mathcal{O}\left(\frac{1}{t^{1-\nu}}\right) + \mathcal{O}\left(\frac{\log t}{t^\nu}\right) + \mathcal{O}\left(\frac{1}{t^{2(\sigma-\nu)}}\right), \tag{4.3}$$

$$\frac{1}{1+t-\tau_t}\sum_{k=\tau_t}^{t}\mathbb{E}\big(\eta_k - r(\boldsymbol{\theta}_k)\big)^2 = \mathcal{O}\left(\frac{1}{t^{1-\nu}}\right) + \mathcal{O}\left(\frac{\log t}{t^\nu}\right) + \mathcal{O}\left(\frac{1}{t^{2(\sigma-\nu)}}\right), \tag{4.4}$$

where $\mathcal{O}(\cdot)$ hides constants, whose exact forms can be found in the detailed proof in Appendix C.2 and C.3.

**Remark 4.8.** The first term $\mathcal{O}(t^{\nu-1})$ on the right hand side of (4.3) and (4.4) comes from loosely bounding the error's norm, and can be removed by applying the "iterative refinement" technique used in Xu et al. [38]. Using this technique, we can obtain a bound (also holds for $\eta_t$) $\mathbb{E}\|\omega_t - \omega_t^*\|^2 = \mathcal{O}(\log t/t^{\nu}) + \mathcal{O}(1/t^{2(\sigma-\nu)-\xi})$, where $\xi > 0$ is an arbitrarily small constant. The constant $\xi$ is an artifact due to the the "iterative refinement" technique. Similar simplification can be done for (4.4). Nevertheless, if we plug (4.3) and (4.4) (after some transformation) into the result of Theorem 4.5, it is easy to see that the term $\mathcal{O}(1/t^{1-\nu})$ is actually dominated by the term $\mathcal{O}(1/t^{1-\sigma})$. Thus this term makes no difference in the total sample complexity of Algorithm 1 and we choose not to complicate the proof or introduce the extra artificial parameter $\xi$ in the result of Theorem 4.7.

The second term in both (4.3) and (4.4) comes from the Markovian noise and the variance of the semi-gradient. The third term in these two equations comes from the slow drift of the actor. These two terms together can be interpreted as follows: if the actor moves much slower than the critic (i.e., $\sigma - \nu \gg \nu$), then the error is dominated by the Markovian noise and gradient variance; if the actor moves not too slowly compared with the critic (i.e. $\sigma - \nu \ll \nu$), then the critic's error is dominated by the slowly drifting effect of the actor.

### 4.4 Convergence rate and sample complexity

Combining Theorems 4.5 and 4.7 leads to the following convergence rate and sample complexity for Algorithm 1. Detailed proof is in Appendix C.4.

**Corollary 4.9.** Under the same assumptions of Theorems 4.5 and 4.7, we have

$$\min_{0 \le k \le t} \mathbb{E}\|\nabla J(\boldsymbol{\theta}_k)\|^2 = \mathcal{O}(\epsilon_{\text{app}}) + \mathcal{O}\left(\frac{1}{t^{1-\sigma}}\right) + \mathcal{O}\left(\frac{\log t}{t^{\nu}}\right) + \mathcal{O}\left(\frac{1}{t^{2(\sigma-\nu)}}\right).$$

If we set $\sigma = 3/5, \nu = 2/5$, leading to the actor step size $\alpha_t = O(1/t^{3/5})$ and the critic step size $\beta_t = O(1/t^{2/5})$, Algorithm 1 can find an $\epsilon$-approximate stationary point of $J(\cdot)$ within $T$ steps, namely,

$$\min_{0 \le k \le T} \mathbb{E}\left\|\nabla J(\boldsymbol{\theta}_k)\right\|^2 \le \mathcal{O}(\epsilon_{\text{app}}) + \epsilon,$$

where $T = \widetilde{\mathcal{O}}(\epsilon^{-2.5})$ is the total iteration number.

Corollary 4.9 combines the results of Theorems 4.5 and 4.7 and shows that the convergence rate of Algorithm 1 is $\widetilde{\mathcal{O}}(t^{-2/5})$. Since the per iteration sample is 1, the sample complexity of two time-scale actor-critic is $\widetilde{\mathcal{O}}(\epsilon^{-2.5})$.

**Remark 4.10.** We compare our results with existing results on the sample complexity of actor-critic methods in the literature. Kumar et al. [18] provided a general result that after $T = \mathcal{O}(\epsilon^{-2})$ updates for the actor, the algorithm can achieve $\min_{0 \le k \le T} \mathbb{E}\|\nabla J(\boldsymbol{\theta}_k)\|^2 \le \epsilon$ , as long as the estimation error of the critic can be bounded by $\mathcal{O}(t^{-1/2})$ at the $t$-th actor's update. However, to ensure such a condition on the critic, they need to draw $t$ samples to estimate the critic at the $t$-th actor's update. Therefore, the total number of samples drawn from the whole training process by the actor-critic algorithm in [18] is $\mathcal{O}(T^2)$, yielding a $\mathcal{O}(\epsilon^{-4})$ sample complexity. Under the similar setting, Qiu et al. [23] proved the same sample complexity $\widetilde{\mathcal{O}}(\epsilon^{-4})$ when TD(0) is used for estimating the critic. Thus Corollary 4.9 suggests that the sample complexity of Algorithm 1 is significantly better than the sample complexity presented in [18, 23] by a factor of $\mathcal{O}(\epsilon^{-1.5})$.

**Remark 4.11.** The gap between the "decoupled" actor-critic and the two time-scale actor-critic seems huge. Intuitively, this is due to the inefficient usage of the samples. At each iteration, the critic in the "decoupled" algorithm starts over to evaluate the policy's value function and discards the history information, regardless of the fact that the policy might only changed slightly. The two time-scale actor-critic keeps the critic's parameter and thus takes full advantage of each samples in the trajectory.

**Remark 4.12.** According to [22], the sample complexity of policy gradient methods such as RE-INFORCE is $\mathcal{O}(\epsilon^{-2})$. As a comparison, if the critic converges faster than $\mathcal{O}(t^{-1/2})$, namely $\mathcal{E}(t) = \mathcal{O}(t^{-1/2})$, then Theorem 4.5 combined with Corollary 4.9 implies that the complexity of two time-scale actor-critic is $\widetilde{\mathcal{O}}(\epsilon^{-2})$, which matches the result of policy gradient methods [22] up to logarithmic factors. Nevertheless, as we have discussed in the previous remarks, a smaller

estimation error for critic often comes at the cost of more samples needed for the critic update [23, 18], which eventually increases the total sample complexity. Therefore, the $\widetilde{\mathcal{O}}(\epsilon^{-2.5})$ sample complexity in Corollary 4.9 is indeed the lowest we can achieve so far for classic two time-scale actor-critic methods. However, it is possible to further improve the sample complexity by using policy evaluation algorithms better than vanilla TD(0), such as GTD and TDC methods.

## 5  Conclusion and discussion

In this paper, we provided the first finite-time analysis of the two time-scale actor-critic methods, with non-i.i.d. Markovian samples and linear function approximation. The algorithm we analyzed is an on-line, one-step actor-critic algorithm which is practical and efficient. We proved its non-asymptotic convergence rate as well as its sample complexity. Our proof technique can be potentially extended to analyze other two time-scale reinforcement learning algorithms.

As one of the anonymous reviewers suggested, the compatible features are useful tools to address the function approximation error of the critic [16]. This can leads to finite-time analysis for the natural actor-critic algorithm [39], which also relates to the more general natural policy gradient methods [9]. Another possible improvement is to use regularization( e.g. ridge) for the critic to ensure the boundedness of the critic and remove the assumption on the maximum eigenvalue. The analysis can also be applied to the infinite-horizon discounted MDP, where the framework of analysis essentially remains the same.

## Broader impact

This work could positively impact the industrial application of actor-critic algorithms and other reinforcement learning algorithms. The theorem exhibits the sample complexity of actor-critic algorithms, which could be used to estimate required training time of reinforcement learning models. Another direct application of our result is to set the learning rate according to the finite-time bound, by optimizing the constant factors of the dominant terms. In this sense, the result could potentially reduce the overhead of hyper-parameter tuning, thus saving both human and computational resources. Moreover, the new analysis in this paper can potentially help people in different fields to understand the broader class of two-time scale algorithms, in addition to actor-critic methods. To our knowledge, this algorithm and theory studied in our paper do not have any ethical issues.

## Acknowledgement

We would like to thank the anonymous reviewers for their helpful comments. This research was sponsored in part by the National Science Foundation IIS-1904183 and Adobe Data Science Research Award. The views and conclusions contained in this paper are those of the authors and should not be interpreted as representing any funding agencies.

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
