[Supplementary Material]

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

# A Proof Sketch

In this section, we provide the proof roadmap of the main theory. Detailed proofs can be found in Appendix C.

## A.1 Proof Sketch of Theorem 4.5

The following lemma is important in that it enables the analysis of policy gradient method:

**Lemma A.1** ([42]). For the performance function defined in (3.1), there exists a constant $L_J > 0$ such that for all $\boldsymbol{\theta}_1, \boldsymbol{\theta}_2 \in \mathbb{R}^d$, it holds that

$$\left\|\nabla J(\boldsymbol{\theta}_1) - \nabla J(\boldsymbol{\theta}_2)\right\| \leq L_J \|\boldsymbol{\theta}_1 - \boldsymbol{\theta}_2\|,$$

which by the definition of smoothness [21] is also equivalent to

$$J(\boldsymbol{\theta}_2) \geq J(\boldsymbol{\theta}_1) + \left\langle \nabla J(\boldsymbol{\theta}_1), \boldsymbol{\theta}_2 - \boldsymbol{\theta}_1 \right\rangle - \frac{L_J}{2} \|\boldsymbol{\theta}_1 - \boldsymbol{\theta}_2\|^2.$$

This lemma enables us to perform a gradient ascent style analysis on the non-concave function $J(\boldsymbol{\theta})$:

$$
\begin{aligned}
J(\boldsymbol{\theta}_{t+1}) &\geq J(\boldsymbol{\theta}_t) + \alpha_t \left\langle \nabla J(\boldsymbol{\theta}_t), \delta_t \nabla \log \pi_{\boldsymbol{\theta}_t}(a_t|s_t) \right\rangle - L_J \alpha_t^2 \left\| \delta_t \nabla \log \pi_{\boldsymbol{\theta}_t}(a_t|s_t) \right\|^2 \\
&= J(\boldsymbol{\theta}_t) + \alpha_t \left\langle \nabla J(\boldsymbol{\theta}_t), \Delta h(O_t, \eta_t, \boldsymbol{\omega}_t, \boldsymbol{\theta}_t) \right\rangle + \alpha_t \left\langle \nabla J(\boldsymbol{\theta}_t), \Delta h'(O_t, \boldsymbol{\theta}_t) \right\rangle \\
&\quad + \alpha_t \Gamma(O_t, \boldsymbol{\theta}_t) + \alpha_t \left\| \nabla J(\boldsymbol{\theta}_t) \right\|^2 - L_J \alpha_t^2 \left\| \delta_t \nabla \log \pi_{\boldsymbol{\theta}_t}(a_t|s_t) \right\|^2, \qquad \text{(A.1)}
\end{aligned}
$$

where $O_t = (s_t, a_t, s_{t+1})$ is a tuple of observations. The second term $\Delta h(O_t, \boldsymbol{\omega}_t, \boldsymbol{\theta}_t)$ on the right hand side of (A.1) is the bias introduced by the critic. The third term $\Delta h'(O_t, \boldsymbol{\theta}_t)$ is from the linear approximation error. The fourth term $\Gamma(O_t, \boldsymbol{\theta}_t)$ is due to the Markovian noise. The last term can be viewed as the variance of the stochastic gradient update. Please refer to (C.1) for the definition of each notation.

Now we bound each term's expectation in (A.1) respectively.

First, we have

$$\mathbb{E}\left\langle \nabla J(\boldsymbol{\theta}_t), \Delta h(O_t, \eta_t, \boldsymbol{\omega}_t, \boldsymbol{\theta}_t) \right\rangle \geq -B \sqrt{\mathbb{E}\left\| \nabla J(\boldsymbol{\theta}_t) \right\|^2} \sqrt{8\mathbb{E}\|\mathbf{z}_t\|^2 + 2\mathbb{E}[y_t^2]},$$

where $\mathbf{z}_t := \boldsymbol{\omega}_t - \boldsymbol{\omega}_t^*$ and $y_t := \eta_t - \eta_t^*$, and the inequality is due to Cauchy inequality and Lemma C.2.

Second, taking expectation over the approximation error term containing $\Delta h'$, we have

$$
\begin{aligned}
\mathbb{E}\left\langle \nabla J(\boldsymbol{\theta}_t), \Delta h'(O_t, \boldsymbol{\theta}_t) \right\rangle &\geq -G_{\boldsymbol{\theta}} \sqrt{\mathbb{E}\left\| \Delta h'(O_t, \boldsymbol{\theta}_t) \right\|^2} \\
&\geq -G_{\boldsymbol{\theta}} \cdot 2B \sqrt{\mathbb{E}\left(\phi(s)^\top \boldsymbol{\omega}_t^* - V^{\pi_{\boldsymbol{\theta}_t}}(s)\right)^2} \\
&\geq -2BG_{\boldsymbol{\theta}} \epsilon_{\mathrm{app}},
\end{aligned}
$$

Third, we have

$$
\begin{aligned}
\mathbb{E}[\Gamma(O_t, \boldsymbol{\theta}_t)] &\geq -G_{\boldsymbol{\theta}}\left( D_1(\tau+1) \sum_{k=t-\tau+1}^{t} \mathbb{E}\|\boldsymbol{\theta}_k - \boldsymbol{\theta}_{k-1}\| + D_2 m \rho^{\tau-1} \right), \\
&\geq -G_{\boldsymbol{\theta}}\left( D_1(\tau+1) G_{\boldsymbol{\theta}} \sum_{k=t-\tau+1}^{t-1} \alpha_k + D_2 m \rho^{\tau-1} \right),
\end{aligned}
$$

where the first inequality is due to Lemma C.3, and the second inequality is due to $\left\| \delta_t \nabla \log \pi_{\boldsymbol{\theta}_t}(a_t|s_t) \right\| \leq G_{\boldsymbol{\theta}}$ by Lemma C.3.

Taking the expectation of (C.2), plugging the above terms back into it and rearranging give

$$
\begin{aligned}
\mathbb{E}\left\| \nabla J(\boldsymbol{\theta}_t) \right\|^2 &\leq \frac{1}{\alpha_t}\left( \mathbb{E}[J(\boldsymbol{\theta}_{t+1})] - \mathbb{E}[J(\boldsymbol{\theta}_t)] \right) + B\sqrt{\mathbb{E}\left\| \nabla J(\boldsymbol{\theta}_t) \right\|^2}\sqrt{8\mathbb{E}\|\mathbf{z}_t\|^2 + 2\mathbb{E}[y_t^2]} \\
&\quad + D_1 G_{\boldsymbol{\theta}}^2 (\tau+1) \sum_{k=t-\tau}^{t-1} \alpha_k + D_2 G_{\boldsymbol{\theta}} m \rho^{\tau-1} + L_J G_{\boldsymbol{\theta}}^2 \alpha_t.
\end{aligned}
$$

Setting $\tau = \tau_t$ and summing over each term, and further dividing $(1 + t - \tau_t)$ at both sides and assuming $t > 2\tau_t - 1$, we can express the result as

$$\frac{1}{1 + t - \tau_t} \sum_{k=\tau_t}^{t} \mathbb{E}\big\|\nabla J(\boldsymbol{\theta}_t)\big\|^2 \leq \mathcal{O}\Big(\frac{1}{t^{1-\sigma}}\Big) + \mathcal{O}\Big(\frac{(\log t)^2}{t^\sigma}\Big) + \mathcal{O}(\epsilon_{\text{app}})$$

$$+ \frac{2B}{1 + t - \tau_t} \sum_{k=\tau_t}^{t} \sqrt{\mathbb{E}\big\|\nabla J(\boldsymbol{\theta}_t)\big\|^2} \sqrt{8\mathbb{E}\|\mathbf{z}_t\|^2 + 2\mathbb{E}[y_t^2]} \quad \text{(A.2)}$$

By Cauchy-Schwartz inequality, we have

$$\frac{1}{1 + t - \tau_t} \sum_{k=\tau_t}^{t} \sqrt{\mathbb{E}\big\|\nabla J(\boldsymbol{\theta}_t)\big\|^2} \sqrt{8\mathbb{E}\|\mathbf{z}_t\|^2 + 2\mathbb{E}[y_t^2]}$$

$$\leq \Big(\frac{1}{1 + t - \tau_t} \sum_{k=\tau_t}^{t} \mathbb{E}\big\|\nabla J(\boldsymbol{\theta}_t)\big\|^2\Big)^{\frac{1}{2}} \Big(\frac{1}{1 + t - \tau_t} \sum_{k=\tau_t}^{t} \big(8\mathbb{E}\|\mathbf{z}_t\|^2 + 2\mathbb{E}[y_t^2]\big)\Big)^{\frac{1}{2}}.$$

Now, denote $F(t) := 1/(1 + t - \tau_t) \sum_{k=\tau_t}^{t} \mathbb{E}\|\nabla J(\boldsymbol{\theta}_k)\|^2$ and $Z(t) := 1/(1 + t - \tau_t) \sum_{k=\tau_t}^{t} \big(8\mathbb{E}\|\mathbf{z}_t\|^2 + 2\mathbb{E}[y_t^2]\big)$, and putting them back to (A.2) ($\mathcal{O}$-notation for simplicity):

$$F(t) \leq \mathcal{O}\Big(\frac{1}{t^{1-\sigma}}\Big) + \mathcal{O}\Big(\frac{(\log t)^2}{t^\sigma}\Big) + \mathcal{O}(\epsilon_{\text{app}}) + 2B\sqrt{F(t)} \cdot \sqrt{Z(t)},$$

which further gives

$$\big(\sqrt{F(t)} - B\sqrt{Z(t)}\big)^2 \leq \mathcal{O}\Big(\frac{1}{t^{1-\sigma}}\Big) + \mathcal{O}\Big(\frac{(\log t)^2}{t^\sigma}\Big) + \mathcal{O}(\epsilon_{\text{app}}) + B^2 Z(t).$$

Note that for a general function $H(t) = A(t) + B(t)$ (with each positive), we have

$$H^2(t) = \mathcal{O}\big(A^2(t)\big) + \mathcal{O}\big(B^2(t)\big),$$
$$\sqrt{H(t)} = \mathcal{O}\big(\sqrt{A(t)}\big) + \mathcal{O}\big(\sqrt{B(t)}\big).$$

This means

$$\min_{0 \leq k \leq t} \mathbb{E}\big\|\nabla J(\boldsymbol{\theta}_k)\big\|^2 \leq \frac{1}{1 + t - \tau_t} \sum_{k=\tau_t}^{t} \mathbb{E}\big\|\nabla J(\boldsymbol{\theta}_k)\big\|^2$$

$$= \mathcal{O}\Big(\frac{1}{t^{1-\sigma}}\Big) + \mathcal{O}\Big(\frac{1}{t^\sigma}\Big) + \mathcal{O}(\epsilon_{\text{app}}) + \mathcal{O}\big(\mathcal{E}(t)\big).$$

## A.2 Proof Sketch of Theorem 4.7

The proof of Theorem 4.7 can be divided into the following two parts.

### A.2.1 Estimating the Average Reward $\eta_k$

We denote $y_k := \eta_k - r(\boldsymbol{\theta}_k)$. First, we shall mention that many components in this step is uses the same framework and partial result as the proof regarding $\boldsymbol{\omega}_t$ in the next part. Also, part of the proof is intriguingly similar with the proof of Theorem 4.5. For simplicity, here we only present the final result regarding $\eta_k$. Please refer to Section C.2 for the detailed proof. By setting $\gamma_k = (1 + t)^{-\nu}$, we have that

$$\sum_{k=\tau_t}^{t} \mathbb{E}[y_k^2] = \mathcal{O}(t^\nu) + \mathcal{O}(\log t \cdot t^{1-\nu}) + \mathcal{O}(t^{1-2(\sigma-\nu)}).$$

### A.2.2 Approximating the TD Fixed Point

**Step 1: decomposition of the estimation error.** For simplicity, we denote $\mathbf{z}_t := \boldsymbol{\omega}_t - \boldsymbol{\omega}_t^*$, where the $\boldsymbol{\omega}_t^*$ denotes the exact parameter under policy $\pi_{\boldsymbol{\theta}_t}$. By the critic update in Line 7 of Algorithm 1, we have

$$\|\mathbf{z}_{t+1}\|^2 = \|\mathbf{z}_t\|^2 + 2\beta_t \langle \mathbf{z}_t, \bar{g}(\boldsymbol{\omega}_t, \boldsymbol{\theta}_t) \rangle + 2\beta_t \Lambda(O_t, \boldsymbol{\omega}_t, \boldsymbol{\theta}_t) + 2\beta_t \langle \mathbf{z}_t, \Delta g(O_t, \eta_t, \boldsymbol{\theta}_t) \rangle$$
$$+ 2\langle \mathbf{z}_t, \boldsymbol{\omega}_t^* - \boldsymbol{\omega}_{t+1}^* \rangle + \left\| \beta_t(g(O_t, \boldsymbol{\omega}_t, \boldsymbol{\theta}_t) + \Delta g(O_t, \eta_t, \boldsymbol{\theta}_t)) + (\boldsymbol{\omega}_t^* - \boldsymbol{\omega}_{t+1}^*) \right\|^2. \tag{A.3}$$

where $O_t := (s_t, a_t, s_{t+1})$ is a tuple of observations, $g(O_t, \boldsymbol{\omega}_t)$ and $\bar{g}(\boldsymbol{\theta}_t, \boldsymbol{\omega}_t)$ are the estimated gradient and the true gradient respectively. $\Lambda(O_t, \boldsymbol{\omega}_t, \boldsymbol{\theta}_t) := \langle \boldsymbol{\omega}_t - \boldsymbol{\omega}_t^*, g(O_t, \boldsymbol{\omega}_t) - \bar{g}(\boldsymbol{\theta}_t, \boldsymbol{\omega}_t) \rangle$ can be seen as the error induced by the Markovian noise. Please refer to (C.7) for formal definition of each notation.

The second term on the right hand side of (A.3) can be bounded by $-2\lambda\beta_t \|\mathbf{z}_t\|^2$ due to Assumption 4.1. The third term is a bias term caused by the Markovian noise. The fourth term $\Delta g(O_t, \eta_t, \boldsymbol{\theta}_t)$ is another bias term caused by inaccurate average reward estimator $\eta_t$. The fifth term is caused by the slowly drifting policy parameter $\boldsymbol{\theta}_t$. And the last term can be considered as the variance term.

Rewriting (A.3) and telescoping from $\tau = \tau_t$ to $t$, we have

$$2\lambda \sum_{k=\tau_t}^{t} \mathbb{E}\|\mathbf{z}_k\|^2 \leq \underbrace{\sum_{k=\tau_t}^{t} \frac{1}{\beta_k} \left( \mathbb{E}\|\mathbf{z}_k\|^2 - \mathbb{E}\|\mathbf{z}_{k+1}\|^2 \right)}_{I_1} + 2\underbrace{\sum_{k=\tau_t}^{t} \mathbb{E}\Lambda(\boldsymbol{\theta}_k, \boldsymbol{\omega}_k, O_k)}_{I_2}$$
$$+ 2L_* G_{\boldsymbol{\theta}} \underbrace{\sum_{k=\tau_t}^{t} \frac{\alpha_k}{\beta_k} \sqrt{\mathbb{E}\|\mathbf{z}_k\|}}_{I_3} + \underbrace{\sum_{k=\tau_t}^{t} \sqrt{\mathbb{E}[y_k^2]} \cdot \sqrt{\mathbb{E}\|\mathbf{z}_k\|}}_{I_4} + C_q \underbrace{\sum_{k=\tau_t}^{t} \beta_k}_{I_5}. \tag{A.4}$$

We will see that the Markovian noise $I_2$, the "slowly drifting policy" term $I_3$ and the estimation bias $I_4$ from $\eta_t$ are significant, and bounding the Markovian term is another challenge.

**Step 2: bounding the Markovian bias.** We first decompose $\Lambda(\boldsymbol{\theta}_t, \boldsymbol{\omega}_t, O_t)$ as follows.

$$\Lambda(\boldsymbol{\theta}_t, \boldsymbol{\omega}_t, O_t) = \big( \Lambda(\boldsymbol{\theta}_t, \boldsymbol{\omega}_t, O_t) - \Lambda(\boldsymbol{\theta}_{t-\tau}, \boldsymbol{\omega}_t, O_t) \big) + \big( \Lambda(\boldsymbol{\theta}_{t-\tau}, \boldsymbol{\omega}_t, O_t) - \Lambda(\boldsymbol{\theta}_{t-\tau}, \boldsymbol{\omega}_{t-\tau}, O_t) \big)$$
$$+ \big( \Lambda(\boldsymbol{\theta}_{t-\tau}, \boldsymbol{\omega}_{t-\tau}, O_t) - \Lambda(\boldsymbol{\theta}_{t-\tau}, \boldsymbol{\omega}_{t-\tau}, \widetilde{O}_t) \big) + \Lambda(\boldsymbol{\theta}_{t-\tau}, \boldsymbol{\omega}_{t-\tau}, \widetilde{O}_t). \tag{A.5}$$

The motivation is to employ the uniform ergodicity defined by Assumption 4.2. This technique was first introduced by Bhandari et al. [3] to address the Markovian noise in policy evaluation. Zou et al. [44] extended to the Q-learning setting where the parameter itself both keeps updated and determines the behavior policy. In this work we take one step further to consider that the policy parameter $\boldsymbol{\theta}_t$ is changing, and the evaluation parameter $\boldsymbol{\omega}_t$ is updated. The analysis relies on the auxiliary Markov chain constructed by Zou et al. [44], which is obtained by repeatedly applying policy $\pi_{\boldsymbol{\theta}_{t-\tau}}$:

$$s_{t-\tau} \xrightarrow{\boldsymbol{\theta}_{t-\tau}} a_{t-\tau} \xrightarrow{\mathcal{P}} s_{t-\tau+1} \xrightarrow{\boldsymbol{\theta}_{t-\tau}} \widetilde{a}_{t-\tau+1} \xrightarrow{\mathcal{P}} \widetilde{s}_{t-\tau+2} \xrightarrow{\boldsymbol{\theta}_{t-\tau}} \widetilde{a}_{t-\tau+2} \xrightarrow{\mathcal{P}} \cdots \xrightarrow{\mathcal{P}} \widetilde{s}_t \xrightarrow{\boldsymbol{\theta}_{t-\tau}} \widetilde{a}_t \xrightarrow{\mathcal{P}} \widetilde{s}_{t+1}.$$

For reference, recall that the original Markov chain is given by:

$$s_{t-\tau} \xrightarrow{\boldsymbol{\theta}_{t-\tau}} a_{t-\tau} \xrightarrow{\mathcal{P}} s_{t-\tau+1} \xrightarrow{\boldsymbol{\theta}_{t-\tau+1}} a_{t-\tau+1} \xrightarrow{\mathcal{P}} s_{t-\tau+2} \xrightarrow{\boldsymbol{\theta}_{t-\tau+2}} a_{t-\tau+2} \xrightarrow{\mathcal{P}} \cdots \xrightarrow{\mathcal{P}} s_t \xrightarrow{\boldsymbol{\theta}_t} a_t \xrightarrow{\mathcal{P}} s_{t+1}.$$

By Lipschitz conditions, we can bound the first two terms in (A.5). The third term will be bounded by the total variation between $s_k$ and $\widetilde{s}_k$, which is achieved by recursively bounding total variation between $s_{k-1}$ and $\widetilde{s}_{k-1}$.

In fact, the Markovian noise $\Gamma(O_t, \boldsymbol{\theta}_t)$ in Section C.1 is obtained in a similar way. Due to the space limit, we only present how to bound the more complicated $\Lambda(\boldsymbol{\theta}_t, \boldsymbol{\omega}_t, O_t)$.
We have the final form as:

$$\Lambda(\boldsymbol{\theta}_t, \boldsymbol{\omega}_t, O_t) \leq C_1(\tau + 1)\|\boldsymbol{\theta}_t - \boldsymbol{\theta}_{t-\tau}\| + C_2 m\rho^{\tau-1} + C_3\|\boldsymbol{\omega}_t - \boldsymbol{\omega}_{t-\tau}\|, \tag{A.6}$$

where $C_1 = 2U_\delta^2 |\mathcal{A}| L(1 + \lceil \log_\rho m^{-1} \rceil + 1/(1-\rho)) + 2U_\delta L_*, C_2 = 2U_\delta^2, C_3 = 4U_\delta$ are constants.

**Step 3: integrating the results.** By some calculation, terms $I_1$, $I_2$ and $I_4$ can be respectively bounded as follows (set $\tau = \tau_t$ defined in (4.1)). The detailed derivation can be found in Appendix C.3,

$$I_1 = 4R_\omega^2 \frac{1}{\beta_t} = \mathcal{O}(t^\nu),$$

$$I_2 \le C_1 G_{\boldsymbol{\theta}}(\tau_t + 1)^2 \sum_{k=0}^{t-\tau_t} \alpha_k + C_2(t - \tau_t + 1)\alpha_t + C_3 U_\delta \tau_t \sum_{k=0}^{t-\tau_t} \beta_k$$

$$= \mathcal{O}\big((\log t)^2 t^{1-\sigma}\big) + \mathcal{O}(t^{1-\sigma}) + \mathcal{O}\big((\log t)t^{1-\nu}\big)$$

$$= \mathcal{O}\big((\log t)t^{1-\nu}\big),$$

$$I_5 = \sum_{k=0}^{t-\tau_t} \beta_k = \mathcal{O}(t^{1-\nu}).$$

The $\log t$ comes from $\tau_t = \mathcal{O}(\log t)$. Performing the same technique on $I_3$ as in Step 3 in the proof sketch of Theorem 4.5, we have

$$I_3 \le \left( \sum_{k=0}^{t-\tau_t} \frac{\alpha_k^2}{\beta_k^2} \right)^{\frac{1}{2}} \left( \sum_{k=\tau_t}^{t} \mathbb{E}\|\mathbf{z}_k\|^2 \right)^{\frac{1}{2}},$$

$$I_4 \le \left( \sum_{k=\tau_t}^{t} \mathbb{E}[y_k^2] \right)^{\frac{1}{2}} \left( \sum_{k=\tau_t}^{t} \mathbb{E}\|\mathbf{z}_k\|^2 \right)^{\frac{1}{2}}.$$

After plugging each term into (A.4), we have that

$$2\lambda \sum_{k=\tau_t}^{t} \mathbb{E}\|\mathbf{z}_k\|^2 \le \mathcal{O}(t^\nu) + \mathcal{O}\big((\log t)t^{1-\nu}\big)$$

$$+ 2L_* G_{\boldsymbol{\theta}} \left( \sum_{k=0}^{t-\tau_t} \frac{\alpha_k^2}{\beta_k^2} \right)^{\frac{1}{2}} \left( \sum_{k=\tau_t}^{t} \mathbb{E}\|\mathbf{z}_k\|^2 \right)^{\frac{1}{2}} + \left( \sum_{k=0}^{t-\tau_t} \mathbb{E}[y_k^2] \right)^{\frac{1}{2}} \left( \sum_{k=\tau_t}^{t} \mathbb{E}\|\mathbf{z}_k\|^2 \right)^{\frac{1}{2}}.$$

This inequality actually resembles (A.2). Following the same procedure as the proof of Theorem 4.5, starting from (A.2), we can finally get

$$\frac{1}{1 + t - \tau_t} \sum_{k=\tau_t}^{t} \mathbb{E}\|\mathbf{z}_k\|^2 = \mathcal{O}\left(\frac{1}{t^{1-\nu}}\right) + \mathcal{O}\left(\frac{\log t}{t^\nu}\right) + \mathcal{O}\left(\frac{1}{t^{2(\sigma-\nu)}}\right).$$

Note that this requires the step sizes $\gamma_t$ and $\beta_t$ should be of the same order $\mathcal{O}(t^{-\nu})$.

# B Preliminary Lemmas

These useful lemmas are frequently applied throughout the proof.

## B.1 Probabilistic Lemmas

The first two statements in the following lemma come from Zou et al. [44].

**Lemma B.1.** For any $\boldsymbol{\theta}_1$ and $\boldsymbol{\theta}_2$, it holds that

$$d_{TV}(\mu_{\boldsymbol{\theta}_1}, \mu_{\boldsymbol{\theta}_2}) \le |\mathcal{A}| L \left( \lceil \log_\rho m^{-1} \rceil + \frac{1}{1-\rho} \right) \|\boldsymbol{\theta}_1 - \boldsymbol{\theta}_2\|,$$

$$d_{TV}(\mu_{\boldsymbol{\theta}_1} \otimes \pi_{\boldsymbol{\theta}_1}, \mu_{\boldsymbol{\theta}_2} \otimes \pi_{\boldsymbol{\theta}_2}) \le |\mathcal{A}| L \left( 1 + \lceil \log_\rho m^{-1} \rceil + \frac{1}{1-\rho} \right) \|\boldsymbol{\theta}_1 - \boldsymbol{\theta}_2\|,$$

$$d_{TV}(\mu_{\boldsymbol{\theta}_1} \otimes \pi_{\boldsymbol{\theta}_1} \otimes \mathcal{P}, \mu_{\boldsymbol{\theta}_2} \otimes \pi_{\boldsymbol{\theta}_2} \otimes \mathcal{P}) \le |\mathcal{A}| L \left( 1 + \lceil \log_\rho m^{-1} \rceil + \frac{1}{1-\rho} \right) \|\boldsymbol{\theta}_1 - \boldsymbol{\theta}_2\|.$$

*Proof.* The proof of the first two inequality is exactly the same as Lemma A.3 in Zou et al. [44], which mainly depends on Theorem 3.1 in Mitrophanov [19]. Here we provide the proof of the third

inequality. Note that

$$d_{TV}(\mu_{\boldsymbol{\theta}_1} \otimes \pi_{\boldsymbol{\theta}_1} \otimes \mathcal{P}, \mu_{\boldsymbol{\theta}_2} \otimes \pi_{\boldsymbol{\theta}_2} \otimes \mathcal{P})$$

$$= \frac{1}{2} \int_{\mathcal{S}} \sum_{\mathcal{A}} \int_{\mathcal{S}} |\mu_{\boldsymbol{\theta}_1}(ds)\pi_{\boldsymbol{\theta}_1}(a|s)\mathcal{P}(ds'|s,a) - \mu_{\boldsymbol{\theta}_2}(ds)\pi_{\boldsymbol{\theta}_2}(a|s)\mathcal{P}(ds'|s,a)|$$

$$= \frac{1}{2} \int_{\mathcal{S}} \sum_{\mathcal{A}} \int_{\mathcal{S}} \mathcal{P}(ds'|s,a)|\mu_{\boldsymbol{\theta}_1}(ds)\pi_{\boldsymbol{\theta}_1}(a|s) - \mu_{\boldsymbol{\theta}_2}(ds)\pi_{\boldsymbol{\theta}_2}(a|s)|$$

$$= \frac{1}{2} \int_{\mathcal{S}} \sum_{\mathcal{A}} |\mu_{\boldsymbol{\theta}_1}(ds)\pi_{\boldsymbol{\theta}_1}(a|s) - \mu_{\boldsymbol{\theta}_2}(ds)\pi_{\boldsymbol{\theta}_2}(a|s)|$$

$$= d_{TV}(\mu_{\boldsymbol{\theta}_1} \otimes \pi_{\boldsymbol{\theta}_1}, \mu_{\boldsymbol{\theta}_2} \otimes \pi_{\boldsymbol{\theta}_2}), \tag{B.1}$$

so it has the same upper bound as the second inequality. $\qquad\square$

**Lemma B.2.** Given time indexes $t$ and $\tau$ such that $t \geq \tau > 0$, consider the auxiliary Markov chain starting from $s_{t-\tau}$. Conditioning on $s_{t-\tau+1}$ and $\boldsymbol{\theta}_{t-\tau}$, the Markov chain is obtained by repeatedly applying policy $\pi_{\boldsymbol{\theta}_{t-\tau}}$.

$$s_{t-\tau} \xrightarrow{\boldsymbol{\theta}_{t-\tau}} a_{t-\tau} \xrightarrow{\mathcal{P}} s_{t-\tau+1} \xrightarrow{\boldsymbol{\theta}_{t-\tau}} \widetilde{a}_{t-\tau+1} \xrightarrow{\mathcal{P}} \widetilde{s}_{t-\tau+2} \xrightarrow{\boldsymbol{\theta}_{t-\tau}} \widetilde{a}_{t-\tau+2} \xrightarrow{\mathcal{P}} \cdots \xrightarrow{\mathcal{P}} \widetilde{s}_t \xrightarrow{\boldsymbol{\theta}_{t-\tau}} \widetilde{a}_t \xrightarrow{\mathcal{P}} \widetilde{s}_{t+1}.$$

For reference, recall that the original Markov chain is given as:

$$s_{t-\tau} \xrightarrow{\boldsymbol{\theta}_{t-\tau}} a_{t-\tau} \xrightarrow{\mathcal{P}} s_{t-\tau+1} \xrightarrow{\boldsymbol{\theta}_{t-\tau+1}} a_{t-\tau+1} \xrightarrow{\mathcal{P}} s_{t-\tau+2} \xrightarrow{\boldsymbol{\theta}_{t-\tau+2}} a_{t-\tau+2} \xrightarrow{\mathcal{P}} \cdots \xrightarrow{\mathcal{P}} s_t \xrightarrow{\boldsymbol{\theta}_t} a_t \xrightarrow{\mathcal{P}} s_{t+1}.$$

Throughout this lemma, we always condition the expectation on $s_{t-\tau+1}$ and $\boldsymbol{\theta}_{t-\tau}$ and omit this in order to simplify the presentation. Under the setting introduced above, we have:

$$d_{TV}(\mathbb{P}(s_{t+1} \in \cdot), \mathbb{P}(\widetilde{s}_{t+1} \in \cdot)) \leq d_{TV}(\mathbb{P}(O_t \in \cdot), \mathbb{P}(\widetilde{O}_t \in \cdot)), \tag{B.2}$$

$$d_{TV}(\mathbb{P}(O_t \in \cdot), \mathbb{P}(\widetilde{O}_t \in \cdot)) = d_{TV}(\mathbb{P}((s_t, a_t) \in \cdot), \mathbb{P}((\widetilde{s}_t, \widetilde{a}_t) \in \cdot)), \tag{B.3}$$

$$d_{TV}(\mathbb{P}((s_t, a_t) \in \cdot), \mathbb{P}((\widetilde{s}_t, \widetilde{a}_t) \in \cdot)) \leq d_{TV}(\mathbb{P}(s_t \in \cdot), \mathbb{P}((\widetilde{s}_t \in \cdot)) + \frac{1}{2}|\mathcal{A}|LE[\|\boldsymbol{\theta}_t - \boldsymbol{\theta}_{t-\tau}\|]. \tag{B.4}$$

*Proof of* (B.2). By the Law of Total Probability,

$$\mathbb{P}(s_{t+1} \in \cdot) = \int_{\mathcal{S}} \sum_{\mathcal{A}} \mathbb{P}(s_t = ds, a_t = a, s_{t+1} \in \cdot),$$

and a similar argument also holds for $\widetilde{O}_t$. Then we have

$$2d_{TV}(\mathbb{P}(s_{t+1} \in \cdot), \mathbb{P}(\widetilde{s}_{t+1} \in \cdot))$$

$$= \int_{\mathcal{S}} \left| \int_{\mathcal{S}} \sum_{\mathcal{A}} \mathbb{P}(s_t = ds, a_t = a, s_{t+1} = ds') - \int_{\mathcal{S}} \sum_{\mathcal{A}} \mathbb{P}(s_t = ds, a_t = a, s_{t+1} = ds') \right|$$

$$\leq \int_{\mathcal{S}} \int_{\mathcal{S}} \sum_{\mathcal{A}} |\mathbb{P}(s_t = ds, a_t = a, s_{t+1} = ds') - \mathbb{P}(s_t = ds, a_t = a, s_{t+1} = ds')|$$

$$= \int_{\mathcal{S}} \int_{\mathcal{S}} \sum_{\mathcal{A}} |\mathbb{P}(O_t = (ds, a, ds')) - \mathbb{P}(\widetilde{O}_t = (ds, a, ds'))|$$

$$= 2d_{TV}(\mathbb{P}(O_t \in \cdot), \mathbb{P}(\widetilde{O}_t \in \cdot)).$$

The last equality requires exchange of integral, which should be guaranteed by the regularity. $\qquad\square$

*Proof of* (B.3).

$$2d_{TV}\big(\mathbb{P}(O_t \in \cdot), \mathbb{P}(\widetilde{O}_t \in \cdot)\big)$$

$$= \int_{\mathcal{S}} \sum_{\mathcal{A}} \int_{\mathcal{S}} \big|\mathbb{P}(O_t = (ds, a, ds')) - \mathbb{P}(\widetilde{O}_t = (ds, a, ds'))\big|$$

$$= \int_{\mathcal{S}} \sum_{\mathcal{A}} \int_{\mathcal{S}} \big|\mathcal{P}(ds'|s, a)\mathbb{P}((s_t, a_t) = (ds, a)) - \mathcal{P}(ds'|s, a)\mathbb{P}((\widetilde{s}_t, \widetilde{a}_t) = (ds, a))\big|$$

$$= \int_{\mathcal{S}} \sum_{\mathcal{A}} \int_{\mathcal{S}} \mathcal{P}(ds'|s, a)\big|\mathbb{P}((s_t, a_t) = (ds, a)) - \mathbb{P}((\widetilde{s}_t, \widetilde{a}_t) = (ds, a))\big|$$

$$= \int_{\mathcal{S}} \sum_{\mathcal{A}} \big|\mathbb{P}((s_t, a_t) = (ds, a)) - \mathbb{P}((\widetilde{s}_t, \widetilde{a}_t) = (ds, a))\big|$$

$$= 2d_{TV}\big(\mathbb{P}((s_t, a_t) \in \cdot), \mathbb{P}((\widetilde{s}_t, \widetilde{a}_t) \in \cdot)\big).$$

$\square$

*Proof of* (B.4). Because $\boldsymbol{\theta}_t$ is also dependent on $s_t$, we make it clear here that

$$\mathbb{P}\big((s_t, a_t) = (ds, a)\big) = \int_{\boldsymbol{\theta} \in \mathbb{R}^d} \mathbb{P}(s_t = ds)\mathbb{P}(\boldsymbol{\theta}_t = d\boldsymbol{\theta}|s_t = ds)\mathbb{P}(a_t = a|s_t = ds, \boldsymbol{\theta}_t = d\boldsymbol{\theta})$$

$$= \int_{\boldsymbol{\theta} \in \mathbb{R}^d} \mathbb{P}(s_t = ds)\mathbb{P}(\boldsymbol{\theta}_t = d\boldsymbol{\theta}|s_t = ds)\pi_{\boldsymbol{\theta}_t}(a|ds)$$

$$= \mathbb{P}(s_t = ds) \int_{\boldsymbol{\theta} \in \mathbb{R}^d} \mathbb{P}(\boldsymbol{\theta}_t = d\boldsymbol{\theta}|s_t = ds)\pi_{\boldsymbol{\theta}_t}(a|ds)$$

$$= \mathbb{P}(s_t = ds)\mathbb{E}\big[\pi_{\boldsymbol{\theta}_t}(a|ds)|s_t = ds\big].$$

Therefore, the total variance can be bounded as

$$2d_{TV}\big(\mathbb{P}((s_t, a_t) \in \cdot), \mathbb{P}((\widetilde{s}_t, \widetilde{a}_t) \in \cdot)\big)$$

$$= \int_{\mathcal{S}} \sum_{\mathcal{A}} \big|\mathbb{P}(s_t = ds)\mathbb{E}[\pi_{\boldsymbol{\theta}_t}(a|ds)|s_t = ds] - \mathbb{P}(\widetilde{s}_t = ds)\pi_{\boldsymbol{\theta}_{t-\tau}}(a|ds)\big|$$

$$= \int_{\mathcal{S}} \sum_{\mathcal{A}} \big|\mathbb{P}(s_t = ds)\mathbb{E}[\pi_{\boldsymbol{\theta}_t}(a|ds)|s_t = ds] - \mathbb{P}(s_t = ds)\pi_{\boldsymbol{\theta}_{t-\tau}}(a|ds)\big|$$

$$+ \int_{\mathcal{S}} \sum_{\mathcal{A}} \big|\mathbb{P}(s_t = ds)\pi_{\boldsymbol{\theta}_{t-\tau}}(a|ds) - \mathbb{P}(\widetilde{s}_t = ds)\pi_{\boldsymbol{\theta}_{t-\tau}}(a|ds)\big|$$

$$= \int_{\mathcal{S}} \mathbb{P}(s_t = ds) \sum_{\mathcal{A}} \big|\mathbb{E}[\pi_{\boldsymbol{\theta}_t}(a|ds)|s_t = ds] - \pi_{\boldsymbol{\theta}_{t-\tau}}(a|ds)\big|$$

$$+ 2d_{TV}\big(\mathbb{P}(s_t \in \cdot), \mathbb{P}((\widetilde{s}_t \in \cdot)\big)$$

$$\leq |\mathcal{A}|L\mathbb{E}\big[\|\boldsymbol{\theta}_t - \boldsymbol{\theta}_{t-\tau}\|\big] + 2d_{TV}\big(\mathbb{P}(s_t \in \cdot), \mathbb{P}((\widetilde{s}_t \in \cdot)\big),$$

where the inequality holds due to the Lipschitz continuity of the policy as in Assumption 4.3. $\square$

## B.2 Lipschitzness of the Optimal Parameter

This section is used to present the proof of Proposition 4.4.

*Proof of Proposition 4.4.* Sutton and Barto [27] has proved in Chapter 9 the fact that the linear TD(0) will converge to the optimal point (w.r.t. Mean Square Projected Bellman Error) which satisfies

$$\mathbf{A}_i \boldsymbol{\omega}^*(\boldsymbol{\theta}_i) = \mathbf{b}_i,$$

where $\mathbf{A}_i := \mathbb{E}[\boldsymbol{\phi}(s)(\boldsymbol{\phi}(s) - \boldsymbol{\phi}(s'))^\top]$ and $\mathbf{b}_i := \mathbb{E}[(r(s, a) - r(\boldsymbol{\theta}_i))\boldsymbol{\phi}(s)]$. The expectation is taken over the stationary distribution $s \sim \mu_{\boldsymbol{\theta}_i}$, the action $a \sim \pi_{\boldsymbol{\theta}_i}(\cdot|s)$ and the transition probability matrix $s' \sim \mathcal{P}(\cdot|s, a)$.

Now we denote $\boldsymbol{\omega}_1^*, \boldsymbol{\omega}_2^*, \widehat{\boldsymbol{\omega}}_1$ as the unique solutions of the following equations respectively:
$$\mathbf{A}_1\boldsymbol{\omega}_1^* = \mathbf{b}_1,$$
$$\mathbf{A}_2\widehat{\boldsymbol{\omega}}_1 = \mathbf{b}_1,$$
$$\mathbf{A}_2\boldsymbol{\omega}_2^* = \mathbf{b}_2.$$
First we bound $\|\boldsymbol{\omega}_1^* - \widehat{\boldsymbol{\omega}}_1\|$. By definition, we have
$$\|\boldsymbol{\omega}_1^* - \widehat{\boldsymbol{\omega}}_1\| \le \|\mathbf{A}_1^{-1} - \mathbf{A}_2^{-1}\|\|\mathbf{b}_1\|.$$
It can be easily shown that
$$\mathbf{A}_1^{-1} - \mathbf{A}_2^{-1} = \mathbf{A}_1^{-1}(\mathbf{A}_2 - \mathbf{A}_1)\mathbf{A}_2^{-1},$$
which further gives
$$\|\boldsymbol{\omega}_1^* - \widehat{\boldsymbol{\omega}}_1\| \le \|\mathbf{A}_1^{-1}\|\|\mathbf{A}_1 - \mathbf{A}_2\|\|\mathbf{A}_2^{-1}\|\|\mathbf{b}_1\|.$$
Then we bound $\|\widehat{\boldsymbol{\omega}}_1 - \boldsymbol{\omega}_2^*\|$,
$$\|\widehat{\boldsymbol{\omega}}_1 - \boldsymbol{\omega}_2^*\| \le \|\mathbf{A}_2^{-1}\|\|\mathbf{b}_1 - \mathbf{b}_2\|.$$
By Assumption 4.1, the eigenvalues of $\mathbf{A}_i$ are bounded from below by $\lambda > 0$, therefore $\|\mathbf{A}_i^{-1}\| \le \lambda^{-1}$. Also $\|\mathbf{b}_1\| \le U_r$ due to the assumption that $|r(s,a)| \le U_r$ and $\|\boldsymbol{\phi}(s)\| \le 1$. To bound $\|\mathbf{A}_1 - \mathbf{A}_2\|$ and $\|\mathbf{b}_1 - \mathbf{b}_2\|$, we first note that
$$\|\mathbf{A}_1 - \mathbf{A}_2\|_2 \le \sup_{s,s'\in\mathcal{S}} \left\|\boldsymbol{\phi}(s)(\boldsymbol{\phi}(s) - \boldsymbol{\phi}(s'))^\top\right\|_2 \cdot 2d_{TV}\big(\mathbb{P}(O^1 \in \cdot), \mathbb{P}(O^2 \in \cdot)\big),$$
$$\le 4d_{TV}\big(\mathbb{P}(O^1 \in \cdot), \mathbb{P}(O^2 \in \cdot)\big)$$
$$\|\mathbf{b}_1 - \mathbf{b}_2\| \le \left\|\mathbb{E}[r(s^1, a^1)\boldsymbol{\phi}(s^1)] - \mathbb{E}[r(s^2, a^2)\boldsymbol{\phi}(s^2)]\right\| + \left\|r(\boldsymbol{\theta}_1)\mathbb{E}[\boldsymbol{\phi}(s^1)] - r(\boldsymbol{\theta}_2)\mathbb{E}[\boldsymbol{\phi}(s^2)]\right\|$$
$$\le 6U_r d_{TV}\big(\mathbb{P}(O^1 \in \cdot), \mathbb{P}(O^2 \in \cdot)\big),$$
where $O^i$ is the tuple obtained by $s^i \sim \mu_{\boldsymbol{\theta}_i}(\cdot)$, $a^i \sim \pi_{\boldsymbol{\theta}_i}(\cdot|s^i)$ and $(s')^i \sim \mathcal{P}(\cdot|s^i, a^i)$. And the total variation norm can be bounded by Lemma B.1 as:
$$d_{TV}\big(\mathbb{P}(O^1 \in \cdot), \mathbb{P}(O^2 \in \cdot)\big) \le |\mathcal{A}|L\left(1 + \lceil\log_\rho m^{-1}\rceil + \frac{1}{1-\rho}\right)\|\boldsymbol{\theta}_1 - \boldsymbol{\theta}_2\|.$$
Collecting the results above gives
$$\|\boldsymbol{\omega}_1^* - \boldsymbol{\omega}_2^*\| \le \|\boldsymbol{\omega}_1^* - \widehat{\boldsymbol{\omega}}_1\| + \|\widehat{\boldsymbol{\omega}}_1 - \boldsymbol{\omega}_2^*\|$$
$$\le (2\lambda^{-2}U_r + 3\lambda^{-1}U_r)|\mathcal{A}|L\left(1 + \lceil\log_\rho m^{-1}\rceil + \frac{1}{1-\rho}\right)\|\boldsymbol{\theta}_1 - \boldsymbol{\theta}_2\|,$$
and we set $L_* := (2\lambda^{-2}U_r + 3\lambda^{-1}U_r)|\mathcal{A}|L(1 + \lceil\log_\rho m^{-1}\rceil + 1/(1-\rho))$ to obtain the final result. $\qquad\square$

## B.3 Asymptotic Equivalence

**Lemma B.3.** Suppose $\{a_i\}$ is a non-negative, bounded sequence, $\tau := C_1 + C_2 \log t(C_2 > 0)$, then for any large enough $t$ such that $t \ge \tau > 0$, we have:
$$\frac{1}{1+t-\tau}\sum_{k=\tau}^{t} a_i = \mathcal{O}\left(\frac{1}{t}\sum_{k=1}^{t} a_i\right),$$
$$\frac{1}{t}\sum_{k=1}^{t} a_i = \mathcal{O}\left(\frac{\log t}{t}\right) + \mathcal{O}\left(\frac{1}{1+t-\tau}\sum_{k=\tau}^{t} a_i\right).$$

*Proof.* We know that $\tau = \mathcal{O}(\log t)$ and the sequence is bounded: $0 < a_i < B$. For the first equation, we have
$$\frac{1}{1+t-\tau}\sum_{k=\tau}^{t} a_i \le \frac{1}{1+t-\tau}\sum_{k=1}^{t} a_i \le \frac{t}{1+t-\tau} \cdot \frac{1}{t}\sum_{k=1}^{t} a_i \le \mathcal{O}\left(\frac{1}{t}\sum_{k=1}^{t} a_i\right),$$
and further assuming $t \ge 2\tau - 2$ gives a constant 2. For the second equation, we have
$$\frac{1}{t}\sum_{k=1}^{t} a_i \le \frac{1}{t}\left((\tau-1)B + \sum_{k=\tau}^{t} a_i\right) = \frac{\tau-1}{t}B + \frac{1}{t}\sum_{k=\tau}^{t} a_i = \mathcal{O}\left(\frac{\log t}{t}\right) + \mathcal{O}\left(\frac{1}{1+t-\tau}\sum_{k=\tau}^{t} a_i\right).$$
$\qquad\square$

# C  Proof of Main Theorems and Propositions

## C.1  Proof of Theorem 4.5

We first define several notations to clarify the dependence:

$$O_t := (s_t, a_t, s_{t+1}),$$
$$\eta^* := \eta(\boldsymbol{\theta}) = \mathbb{E}_{s \sim \mu_{\boldsymbol{\theta}}, a \sim \pi_{\boldsymbol{\theta}}(\cdot|s)}[r(s,a)]$$
$$\Delta h(O, \eta, \boldsymbol{\omega}, \boldsymbol{\theta}) := \Big(\eta(\boldsymbol{\theta}) - \eta + \big(\phi(s') - \phi(s)\big)^\top (\boldsymbol{\omega} - \boldsymbol{\omega}^*)\Big) \nabla \log \pi_{\boldsymbol{\theta}}(a|s),$$
$$\Delta h'(O, \boldsymbol{\theta}) := \Big(\big(\phi(s')^\top \boldsymbol{\omega}^* - V^{\pi_{\boldsymbol{\theta}}}(s')\big) - \big(\phi(s)^\top \boldsymbol{\omega}^* - V^{\pi_{\boldsymbol{\theta}}}(s)\big)\Big) \nabla \log \pi_{\boldsymbol{\theta}}(a|s),$$
$$h(O, \boldsymbol{\theta}) := \big(r(s,a) - \eta(\boldsymbol{\theta}) + V^{\pi_{\boldsymbol{\theta}}}(s') - V^{\pi_{\boldsymbol{\theta}}}(s)\big) \nabla \log \pi_{\boldsymbol{\theta}}(a|s),$$
$$\Gamma(O, \boldsymbol{\theta}) := \big\langle \nabla J(\boldsymbol{\theta}), h(O, \boldsymbol{\theta}) - \nabla J(\boldsymbol{\theta}) \big\rangle. \tag{C.1}$$

Note that $\Delta h$, $\Delta h'$ and $h$ together gives a decomposition of the actual gradient. They each correspond to the error caused by the critic $\boldsymbol{\omega}_t$, the approximation error of the linear class and the stochastic policy gradient.

There are several lemmas that will be used in the proof.

**Lemma C.1.** For the performance function defined in (3.1), there exists a constant $L_J > 0$ such that for all $\boldsymbol{\theta}_1, \boldsymbol{\theta}_2 \in \mathbb{R}^d$, it holds that

$$\big\|\nabla J(\boldsymbol{\theta}_1) - \nabla J(\boldsymbol{\theta}_2)\big\| \leq L_J \|\boldsymbol{\theta}_1 - \boldsymbol{\theta}_2\|,$$

which by the definition of smoothness [21] implies

$$J(\boldsymbol{\theta}_2) \geq J(\boldsymbol{\theta}_1) + \big\langle \nabla J(\boldsymbol{\theta}_1), \boldsymbol{\theta}_2 - \boldsymbol{\theta}_1 \big\rangle - \frac{L_J}{2} \|\boldsymbol{\theta}_1 - \boldsymbol{\theta}_2\|^2.$$

The following two lemmas characterize the bias introduced by the critic's approximation and the Markovian noise.

**Lemma C.2.** For any $t \geq 0$,

$$\big\|\Delta h(O_t, \eta_t, \boldsymbol{\omega}_t, \boldsymbol{\theta}_t)\big\|^2 \leq B^2 \big(8\|\boldsymbol{\omega}_t - \boldsymbol{\omega}_t^*\|^2 + 2(\eta_t - \eta_t^*)^2\big).$$

**Lemma C.3.** For any $\boldsymbol{\theta} \in \mathbb{R}^d$, we have $\|\delta \nabla \log \pi_{\boldsymbol{\theta}}(a|s)\| \leq G_{\boldsymbol{\theta}} := U_\delta \cdot B$, where $U_\delta = 2U_r + 2R_{\boldsymbol{\omega}}$. Furthermore, for any $t \geq 0$, it holds that

$$\mathbb{E}\big[\Gamma(O_t, \boldsymbol{\theta}_t)\big] \geq -G_{\boldsymbol{\theta}}\Big(D_1(\tau+1) \sum_{k=t-\tau+1}^{t} \mathbb{E}\|\boldsymbol{\theta}_k - \boldsymbol{\theta}_{k-1}\| + D_2 m \rho^{\tau-1}\Big),$$

where $D_1 = \max\{(U_\delta L_l + 2L_* B + 3L_J), 2U_\delta B|\mathcal{A}|L\}$ and $D_2 = 4U_\delta B$.

*Proof of Theorem 4.5.* Under the update rule of Algorithm 1, we have by Lemma C.1

$$\begin{aligned}
J(\boldsymbol{\theta}_{t+1}) &\geq J(\boldsymbol{\theta}_t) + \alpha_t \big\langle \nabla J(\boldsymbol{\theta}_t), \delta_t \nabla \log \pi_{\boldsymbol{\theta}_t}(a_t|s_t) \big\rangle - L_J \alpha_t^2 \big\|\delta_t \nabla \log \pi_{\boldsymbol{\theta}_t}(a_t|s_t)\big\|^2 \\
&= J(\boldsymbol{\theta}_t) + \alpha_t \big\langle \nabla J(\boldsymbol{\theta}_t), \Delta h(O_t, \eta_t, \boldsymbol{\omega}_t, \boldsymbol{\theta}_t) \big\rangle + \alpha_t \big\langle \nabla J(\boldsymbol{\theta}_t), \Delta h'(O_t, \boldsymbol{\theta}_t) \big\rangle \\
&\quad + \alpha_t \big\langle \nabla J(\boldsymbol{\theta}_t), h(O_t, \boldsymbol{\theta}_t) \big\rangle - L_J \alpha_t^2 \big\|\delta_t \nabla \log \pi_{\boldsymbol{\theta}_t}(a_t|s_t)\big\|^2 \\
&= J(\boldsymbol{\theta}_t) + \alpha_t \big\langle \nabla J(\boldsymbol{\theta}_t), \Delta h(O_t, \eta_t, \boldsymbol{\omega}_t, \boldsymbol{\theta}_t) \big\rangle + \alpha_t \big\langle \nabla J(\boldsymbol{\theta}_t), \Delta h'(O_t, \boldsymbol{\theta}_t) \big\rangle \\
&\quad + \alpha_t \Gamma(O_t, \boldsymbol{\theta}_t) + \alpha_t \big\|\nabla J(\boldsymbol{\theta}_t)\big\|^2 - L_J \alpha_t^2 \big\|\delta_t \nabla \log \pi_{\boldsymbol{\theta}_t}(a_t|s_t)\big\|^2. \tag{C.2}
\end{aligned}$$

We will bound the expectation of each term on the right hand side of (C.2) as follows. First, we have

$$\mathbb{E}\big\langle \nabla J(\boldsymbol{\theta}_t), \Delta h(O_t, \eta_t, \boldsymbol{\omega}_t, \boldsymbol{\theta}_t) \big\rangle \geq -B\sqrt{\mathbb{E}\big\|\nabla J(\boldsymbol{\theta}_t)\big\|^2} \sqrt{8\mathbb{E}\|\mathbf{z}_t\|^2 + 2\mathbb{E}[y_t^2]},$$

where $\mathbf{z}_t := \boldsymbol{\omega}_t - \boldsymbol{\omega}_t^*$ and $y_t := \eta_t - \eta_t^*$, and the inequality is due to Cauchy inequality and Lemma C.2.

Second, we have

$$\mathbb{E}[\Gamma(O_t, \boldsymbol{\theta}_t)] \geq -G_{\boldsymbol{\theta}}\left( D_1(\tau + 1) \sum_{k=t-\tau+1}^{t} \mathbb{E}\|\boldsymbol{\theta}_k - \boldsymbol{\theta}_{k-1}\| + D_2 m \rho^{\tau-1} \right),$$

$$\geq -G_{\boldsymbol{\theta}}\left( D_1(\tau + 1) G_{\boldsymbol{\theta}} \sum_{k=t-\tau+1}^{t-1} \alpha_k + D_2 m \rho^{\tau-1} \right),$$

where the first inequality is due to Lemma C.3, and the second inequality is due to $\left\| \delta_t \nabla \log \pi_{\boldsymbol{\theta}_t}(a_t|s_t) \right\| \leq G_{\boldsymbol{\theta}}$ by Lemma C.3.

Third, taking expectation over the approximation error term containing $\Delta h'$, we have

$$\mathbb{E}\langle \nabla J(\boldsymbol{\theta}_t), \Delta h'(O_t, \boldsymbol{\theta}_t) \rangle \geq -G_{\boldsymbol{\theta}} \sqrt{\mathbb{E}\left\| \Delta h'(O_t, \boldsymbol{\theta}_t) \right\|^2}$$

$$\geq -G_{\boldsymbol{\theta}} \cdot 2B \sqrt{\mathbb{E}\left( \phi(s)^{\top} \boldsymbol{\omega}_t^* - V^{\pi_{\boldsymbol{\theta}_t}}(s) \right)^2}$$

$$\geq -2BG_{\boldsymbol{\theta}} \epsilon_{\text{app}},$$

Taking the expectation of (C.2) and plugging the above terms back into it gives

$$\mathbb{E}[J(\boldsymbol{\theta}_{t+1})] \geq \mathbb{E}[J(\boldsymbol{\theta}_t)] - \alpha_t B \sqrt{\mathbb{E}\left\| \nabla J(\boldsymbol{\theta}_t) \right\|^2} \sqrt{8\mathbb{E}\|\mathbf{z}_t\|^2 + 2\mathbb{E}[y_t^2]} - 2BG_{\boldsymbol{\theta}} \epsilon_{\text{app}} \alpha_t$$

$$- \alpha_t G_{\boldsymbol{\theta}}\left( D_1(\tau + 1) G_{\boldsymbol{\theta}} \sum_{k=t-\tau}^{t-1} \alpha_k + D_2 m \rho^{\tau-1} \right) + \alpha_t \mathbb{E}\|\nabla J(\boldsymbol{\theta}_t)\|^2 - L_J G_{\boldsymbol{\theta}}^2 \alpha_t^2.$$

Rearranging the above inequality gives

$$\mathbb{E}\left\| \nabla J(\boldsymbol{\theta}_t) \right\|^2 \leq \frac{1}{\alpha_t}\left( \mathbb{E}[J(\boldsymbol{\theta}_{t+1})] - \mathbb{E}[J(\boldsymbol{\theta}_t)] \right) + B \sqrt{\mathbb{E}\left\| \nabla J(\boldsymbol{\theta}_t) \right\|^2} \sqrt{8\mathbb{E}\|\mathbf{z}_t\|^2 + 2\mathbb{E}[y_t^2]}$$

$$+ D_1 G_{\boldsymbol{\theta}}^2 (\tau + 1) \sum_{k=t-\tau}^{t-1} \alpha_k + D_2 G_{\boldsymbol{\theta}} m \rho^{\tau-1} + L_J G_{\boldsymbol{\theta}}^2 \alpha_t.$$

By setting $\tau = \tau_t$, we get

$$\mathbb{E}\left\| \nabla J(\boldsymbol{\theta}_t) \right\|^2 \leq \frac{1}{\alpha_t}\left( \mathbb{E}\left[ J(\boldsymbol{\theta}_{t+1}) \right] - \mathbb{E}\left[ J(\boldsymbol{\theta}_t) \right] \right) + B \sqrt{\mathbb{E}\left\| \nabla J(\boldsymbol{\theta}_t) \right\|^2} \sqrt{8\mathbb{E}\|\mathbf{z}_t\|^2 + 2\mathbb{E}[y_t^2]}$$

$$+ 2BG_{\boldsymbol{\theta}} \epsilon_{\text{app}} + D_1 G_{\boldsymbol{\theta}}^2 (\tau_t + 1)^2 \alpha_{t-\tau_t} + D_2 G_{\boldsymbol{\theta}} \alpha_t + L_J G_{\boldsymbol{\theta}}^2 \alpha_t.$$

Summing over $k$ from $\tau_t$ to $t$ gives

$$\sum_{k=\tau_t}^{t} \mathbb{E}\left\| \nabla J(\boldsymbol{\theta}_t) \right\|^2 \leq \underbrace{\sum_{k=\tau_t}^{t} \frac{1}{\alpha_k}\left( \mathbb{E}[J(\boldsymbol{\theta}_{k+1})] - \mathbb{E}[J(\boldsymbol{\theta}_k)] \right)}_{I_1} + B \sum_{k=\tau_t}^{t} \sqrt{\mathbb{E}\left\| \nabla J(\boldsymbol{\theta}_t) \right\|^2} \sqrt{8\mathbb{E}\|\mathbf{z}_t\|^2 + 2\mathbb{E}[y_t^2]}$$

$$+ \underbrace{\sum_{k=\tau_t}^{t} D_1 G_{\boldsymbol{\theta}}^2 (\tau_t + 1)^2 \alpha_{k-\tau_t} + \sum_{k=\tau_t}^{t} (D_2 G_{\boldsymbol{\theta}} + L_J G_{\boldsymbol{\theta}}^2)\alpha_k}_{I_2} + 2BG_{\boldsymbol{\theta}} \epsilon_{\text{app}} (t - \tau_t + 1).$$

For the term $I_1$, we have,

$$\sum_{k=\tau_t}^{t} \frac{1}{\alpha_k}\left( J(\boldsymbol{\theta}_{k+1}) - J(\boldsymbol{\theta}_k) \right) = \sum_{k=\tau_t}^{t} \left( \frac{1}{\alpha_{k-1}} - \frac{1}{\alpha_k} \right) \mathbb{E}[J(\boldsymbol{\theta}_k)] - \frac{1}{\alpha_{\tau_t-1}} \mathbb{E}[J(\boldsymbol{\theta}_{\tau_t})] + \frac{1}{\alpha_t} \mathbb{E}[J(\boldsymbol{\theta}_{t+1})]$$

$$\leq \sum_{k=\tau_t}^{t} \left( \frac{1}{\alpha_k} - \frac{1}{\alpha_{k-1}} \right) U_r + \frac{1}{\alpha_{\tau_t-1}} U_r + \frac{1}{\alpha_t} U_r$$

$$= U_r \left[ \sum_{k=\tau_t}^{t} \left( \frac{1}{\alpha_k} - \frac{1}{\alpha_{k-1}} \right) + \frac{1}{\alpha_{\tau_t-1}} + \frac{1}{\alpha_t} \right]$$

$$= 2U_r \alpha_t^{-1},$$

where the inequality holds due to $|\mathbb{E}[J(\boldsymbol{\theta})]| \leq U_r$.

For the term $I_2$, we have

$$
\begin{aligned}
\sum_{k=\tau_t}^{t} D_1 G_{\boldsymbol{\theta}}^2 (\tau_t + 1)^2 \alpha_{k-\tau_t} &= D_1 G_{\boldsymbol{\theta}}^2 (\tau_t + 1)^2 \sum_{k=\tau_t}^{t} \alpha_{k-\tau_t} \\
&= D_1 G_{\boldsymbol{\theta}}^2 (\tau_t + 1)^2 \sum_{k=0}^{t-\tau_t} \alpha_k \\
&= D_1 G_{\boldsymbol{\theta}}^2 (\tau_t + 1)^2 c_\alpha \sum_{k=0}^{t-\tau_t} \frac{1}{(1+k)^\sigma},
\end{aligned}
$$

and

$$
\begin{aligned}
\sum_{k=\tau_t}^{t} (D_2 G_{\boldsymbol{\theta}} + L_J G_{\boldsymbol{\theta}}^2) \alpha_k &= (D_2 G_{\boldsymbol{\theta}} + L_J G_{\boldsymbol{\theta}}^2) \sum_{k=\tau_t}^{t} \alpha_k \\
&\leq (D_2 G_{\boldsymbol{\theta}} + L_J G_{\boldsymbol{\theta}}^2) \sum_{k=0}^{t-\tau_t} \alpha_k \\
&= (D_2 G_{\boldsymbol{\theta}} + L_J G_{\boldsymbol{\theta}}^2) c_\alpha \sum_{k=0}^{t-\tau_t} \frac{1}{(1+k)^\sigma}.
\end{aligned}
$$

Note that both upper bounds rely on the summation $\sum_{k=0}^{t-\tau_t} 1/(1+k)^\sigma \leq \int_0^{t-\tau_t+1} x^{-\sigma} dx = 1/(1-\sigma)(t - \tau_t + 1)^{1-\sigma}$. Combining the results for terms $I_1$ and $I_2$, we have

$$
\begin{aligned}
\sum_{k=\tau_t}^{t} \mathbb{E}\|\nabla J(\boldsymbol{\theta}_t)\|^2 \leq{}& \frac{2U_r}{c_\alpha}(1+t)^\sigma \\
&+ \left(D_1 G_{\boldsymbol{\theta}}^2 (\tau_t + 1)^2 + D_2 G_{\boldsymbol{\theta}} + L_J G_{\boldsymbol{\theta}}^2\right) \frac{c_\alpha}{1-\sigma}(t - \tau_t + 1)^{1-\sigma} \\
&+ B \sum_{k=\tau_t}^{t} \sqrt{\mathbb{E}\|\nabla J(\boldsymbol{\theta}_t)\|^2} \sqrt{8\mathbb{E}\|\mathbf{z}_t\|^2 + 2\mathbb{E}[y_t^2]} \\
&+ 2BG_{\boldsymbol{\theta}} \epsilon_{\text{app}} (t - \tau_t + 1).
\end{aligned}
$$

Dividing $(1 + t - \tau_t)$ at both sides and assuming $t > 2\tau_t - 1$, we can express the result as

$$
\begin{aligned}
\frac{1}{1+t-\tau_t} \sum_{k=\tau_t}^{t} \mathbb{E}\|\nabla J(\boldsymbol{\theta}_t)\|^2 \leq{}& \frac{4U_r}{c_\alpha} \frac{1}{(t+1)^{1-\sigma}} \\
&+ \left(D_1 G_{\boldsymbol{\theta}}^2 (\tau_t + 1)^2 + D_2 G_{\boldsymbol{\theta}} + L_J G_{\boldsymbol{\theta}}^2\right) \frac{c_\alpha}{1-\sigma} \frac{1}{(t - \tau_t + 1)^\sigma} \\
&+ \frac{2B}{1+t-\tau_t} \sum_{k=\tau_t}^{t} \sqrt{\mathbb{E}\|\nabla J(\boldsymbol{\theta}_t)\|^2} \sqrt{8\mathbb{E}\|\mathbf{z}_t\|^2 + 2\mathbb{E}[y_t^2]} \\
&+ 2BG_{\boldsymbol{\theta}} \epsilon_{\text{app}}. \quad\quad\quad\quad\quad\quad\quad (\text{C.3})
\end{aligned}
$$

By Cauchy-Schwartz inequality, we have

$$
\begin{aligned}
&\frac{1}{1+t-\tau_t} \sum_{k=\tau_t}^{t} \sqrt{\mathbb{E}\|\nabla J(\boldsymbol{\theta}_t)\|^2} \sqrt{8\mathbb{E}\|\mathbf{z}_t\|^2 + 2\mathbb{E}[y_t^2]} \\
&\leq \left(\frac{1}{1+t-\tau_t} \sum_{k=\tau_t}^{t} \mathbb{E}\|\nabla J(\boldsymbol{\theta}_t)\|^2\right)^{\frac{1}{2}} \left(\frac{1}{1+t-\tau_t} \sum_{k=\tau_t}^{t} \left(8\mathbb{E}\|\mathbf{z}_t\|^2 + 2\mathbb{E}[y_t^2]\right)\right)^{\frac{1}{2}}.
\end{aligned}
$$

Now, denote $F(t) := 1/(1 + t - \tau_t) \sum_{k=\tau_t}^{t} \mathbb{E}\|\nabla J(\boldsymbol{\theta}_k)\|^2$ and $Z(t) := 1/(1 + t - \tau_t) \sum_{k=\tau_t}^{t} \left( 8\mathbb{E}\|\mathbf{z}_t\|^2 + 2\mathbb{E}[y_t^2] \right)$, and putting them back to (C.3) ($\mathcal{O}$-notation for simplicity):

$$F(t) \leq \mathcal{O}\left(\frac{1}{t^{1-\sigma}}\right) + \mathcal{O}\left(\frac{(\log t)^2}{t^\sigma}\right) + \mathcal{O}(\epsilon_{\text{app}}) + 2B\sqrt{F(t)} \cdot \sqrt{Z(t)},$$

which further gives

$$\left(\sqrt{F(t)} - B\sqrt{Z(t)}\right)^2 \leq \mathcal{O}\left(\frac{1}{t^{1-\sigma}}\right) + \mathcal{O}\left(\frac{(\log t)^2}{t^\sigma}\right) + \mathcal{O}(\epsilon_{\text{app}}) + B^2 Z(t). \quad \text{(C.4)}$$

Note that for a general function $H(t) \leq A(t) + B(t)$ (with each positive), we have

$$H^2(t) \leq 2A^2(t) + 2B^2(t),$$
$$\sqrt{H(t)} \leq \sqrt{A(t)} + \sqrt{B(t)}.$$

This means (C.4) implies

$$\sqrt{F(t)} - B\sqrt{Z(t)} \leq \sqrt{A(t)} + B\sqrt{Z(t)},$$
$$\sqrt{F(t)} \leq \sqrt{A(t)} + 2B\sqrt{Z(t)},$$
$$F(t) \leq 2A(t) + 8B^2 Z(t).$$

By Lemma B.3, assuming $t \geq 2\tau_t - 1$, it holds that

$$Z(t) = \frac{1}{1 + t - \tau_t} \sum_{k=\tau_t}^{t} 8\mathbb{E}\|\mathbf{z}_k\|^2 + 2\mathbb{E}[y_t^2] \leq \frac{2}{t} \sum_{k=1}^{t} 8\mathbb{E}\|\mathbf{z}_k\|^2 + 2\mathbb{E}[y_t^2] = 2\mathcal{E}(t).$$

And finally, we have

$$\begin{aligned}
\min_{0 \leq k \leq t} \mathbb{E}\|\nabla J(\boldsymbol{\theta}_k)\|^2 &\leq \frac{1}{1 + t - \tau_t} \sum_{k=\tau_t}^{t} \mathbb{E}\|\nabla J(\boldsymbol{\theta}_k)\|^2 \\
&\leq \frac{8U_r}{c_\alpha} \frac{1}{(t+1)^{1-\sigma}} \\
&\quad + \left(D_1 G_{\boldsymbol{\theta}}^2 (\tau_t + 1)^2 + D_2 G_{\boldsymbol{\theta}} + L_J G_{\boldsymbol{\theta}}^2\right) \frac{2c_\alpha}{1-\sigma} \frac{1}{(t - \tau_t + 1)^\sigma} \\
&\quad + 4BG_{\boldsymbol{\theta}} \epsilon_{\text{app}} \\
&\quad + 16B^2 \mathcal{E}(t) \\
&= \mathcal{O}\left(\frac{1}{t^{1-\sigma}}\right) + \mathcal{O}\left(\frac{1}{t^\sigma}\right) + \mathcal{O}(\epsilon_{\text{app}}) + \mathcal{O}(\mathcal{E}(t)).
\end{aligned}$$

$\square$

## C.2  Proof of Theorem 4.7: Estimating the Average Reward

The two time-scale analysis with Markovian noise and moving behavior policy can be complicated, so we define some useful notations here that could hopefully clarify the probabilistic dependency.

$$\begin{aligned}
O_t &:= (s_t, a_t, s_{t+1}), \\
\eta_t^* &:= \eta^*(\boldsymbol{\theta}_t) = J(\boldsymbol{\theta}_t), \\
y_t &:= \eta_t - \eta_t^*, \\
\Xi(O, \eta, \boldsymbol{\theta}) &:= y_t(r_t - \eta_t^*).
\end{aligned} \quad \text{(C.5)}$$

We also write $J(\boldsymbol{\theta}_t) = r(\boldsymbol{\theta}_t)$ sometimes in the proof.

**Lemma C.4.** For any $\boldsymbol{\theta}_1, \boldsymbol{\theta}_2$, we have

$$\left| J(\boldsymbol{\theta}_1) - J(\boldsymbol{\theta}_2) \right| \leq C_J \|\boldsymbol{\theta}_1 - \boldsymbol{\theta}_2\|,$$

where $C_J = 2U_r |\mathcal{A}| L(1 + \lceil \log_\rho m^{-1} \rceil + 1/(1 - \rho))$.

**Lemma C.5.** Given the definition of $\Xi(O_t, \eta_t, \boldsymbol{\theta}_t)$, for any $t > 0$, we have

$$\mathbb{E}[\Xi(O_t, \eta_t, \boldsymbol{\theta}_t)] \leq 4U_r C_J \|\boldsymbol{\theta}_t - \boldsymbol{\theta}_{t-\tau}\| + 2U_r|\eta_t - \eta_{t-\tau}| + 2U_r^2|\mathcal{A}|L \sum_{i=t-\tau}^{t} \mathbb{E}\|\boldsymbol{\theta}_i - \boldsymbol{\theta}_{t-\tau}\|. + 4U_r^2 m\rho^{\tau-1}.$$

*Proof.* From the definition, $\eta_t$ is the average reward estimator, $\eta_t^* = J(\boldsymbol{\theta}_t) = \mathbb{E}[r(s,a)]$ is the average reward under the stationary distribution $\mu_{\boldsymbol{\theta}_t} \otimes \pi_{\boldsymbol{\theta}_t}$, and $y_t = \eta_t - \eta_t^*$. From the algorithm we have the update rule as

$$\eta_{t+1} := \eta_t + \gamma_t\big(r(s_t, a_t) - \eta_t\big),$$

where we leave the step size $\gamma_t$ unspecified for now. Unrolling the recursive definition we have

$$
\begin{aligned}
y_{t+1}^2 &= \big(y_t + \eta_t^* - \eta_{t+1}^* + \gamma_t(r_t - \eta_t)\big)^2 \\
&\leq y_t^2 + 2\gamma_t y_t(r_t - \eta_t) + 2y_t(\eta_t^* - \eta_{t+1}^*) + 2(\eta_t^* - \eta_{t+1}^*)^2 + 2\gamma_t^2(r_t - \eta_t)^2 \\
&= (1 - 2\gamma_t)y_t^2 + 2\gamma_t y_t(r_t - \eta_t^*) + 2y_t(\eta_t^* - \eta_{t+1}^*) + 2(\eta_t^* - \eta_{t+1}^*)^2 + 2\gamma_t^2(r_t - \eta_t)^2 \\
&= (1 - 2\gamma_t)y_t^2 + 2\gamma_t\Xi(O_k, \eta_k, \boldsymbol{\theta}_k) + 2y_t(\eta_t^* - \eta_{t+1}^*) + 2(\eta_t^* - \eta_{t+1}^*)^2 + 2\gamma_t^2(r_t - \eta_t)^2.
\end{aligned}
$$

Rearranging and summing from $\tau_t$ to $t$, we have

$$\sum_{k=\tau_t}^{t} \mathbb{E}[y_k^2] \leq \underbrace{\sum_{k=\tau_t}^{t} \frac{1}{2\gamma_k}\mathbb{E}(y_k^2 - y_{k+1}^2)}_{I_1} + \underbrace{\sum_{k=\tau_t}^{t} \mathbb{E}[\Xi(O_k, \eta_k, \boldsymbol{\theta}_k)]}_{I_2}$$

$$+ \underbrace{\sum_{k=\tau_t}^{t} \frac{1}{\gamma_k}\mathbb{E}[y_k(\eta_k^* - \eta_{k+1}^*)]}_{I_3} + \underbrace{\sum_{k=\tau_t}^{t} \frac{1}{\gamma_k}\mathbb{E}[(\eta_k^* - \eta_{k+1}^*)^2]}_{I_4} + \underbrace{\sum_{k=\tau_t}^{t} \gamma_k \mathbb{E}[(r_k - \eta_k)^2]}_{I_5}.$$

For $I_1$, following the Abel summation formula, we have

$$
\begin{aligned}
I_1 &= \sum_{k=\tau_t}^{t} \frac{1}{2\gamma_k}(y_k^2 - y_{k+1}^2) \\
&= \sum_{k=\tau_t}^{t} \left(\frac{1}{2\gamma_k} - \frac{1}{2\gamma_{k-1}}\right)y_k^2 + \frac{1}{2\gamma_{\tau_t-1}}y_{\tau_t}^2 - \frac{1}{2\gamma_t}y_{t+1}^2 \\
&\leq \frac{2U_r^2}{\gamma_t}.
\end{aligned}
$$

For $I_2$, from Lemma C.5, we have

$$\mathbb{E}[\Xi(O_t, \eta_t, \boldsymbol{\theta}_t)] \leq 4U_r C_J \|\boldsymbol{\theta}_t - \boldsymbol{\theta}_{t-\tau}\| + 2U_r|\eta_t - \eta_{t-\tau}| + 2U_r^2|\mathcal{A}|L \sum_{i=t-\tau}^{t} \mathbb{E}\|\boldsymbol{\theta}_i - \boldsymbol{\theta}_{t-\tau}\|. + 4U_r^2 m\rho^{\tau-1}$$

$$\leq 4U_r C_J G_{\boldsymbol{\theta}}\tau\alpha_{t-\tau} + 4U_r^2\tau\gamma_{t-\tau} + 2U_r^2|\mathcal{A}|L\tau(\tau+1)G_{\boldsymbol{\theta}}\alpha_{t-\tau} + 4U_r^2 m\rho^{\tau-1}$$

$$\leq C_1\tau^2\alpha_{t-\tau} + C_2\tau\gamma_{t-\tau} + C_3 m\rho^{\tau-1}.$$

By the choice of $\tau_t$, we have

$$I_2 = \sum_{k=\tau_t}^{t} \mathbb{E}[\Xi(O_k, \eta_k, \boldsymbol{\theta}_k)] \leq (C_1\tau_t^2 + C_3)\sum_{k=\tau_t}^{t} \alpha_k + C_2\tau_t \sum_{k=\tau_t}^{t} \gamma_k.$$

For $I_3$, we have

$$I_3 \leq \left(\sum_{k=\tau_t}^{t} \mathbb{E}[y_k^2]\right)^{1/2} \left(C_J^2 G_{\boldsymbol{\theta}}^2 \sum_{k=\tau_t}^{t} \frac{\alpha_k^2}{\gamma_k^2}\right)^{1/2},$$

which is because by Lemma C.4, $(\eta_k^* - \eta_{k+1}^*)$ can be linearly bounded by $\|\boldsymbol{\theta}_k - \boldsymbol{\theta}_{k+1}\| \leq G_{\boldsymbol{\theta}} \cdot \alpha_k$. For $I_4$, by the same argument it holds that

$$
\begin{aligned}
I_4 &= \sum_{k=\tau_t}^{t} \frac{1}{\gamma_k} \mathbb{E}[(\eta_k^* - \eta_{k+1}^*)^2] \\
&= \sum_{k=\tau_t}^{t} \frac{1}{\gamma_k} \mathbb{E}\big[\big(J(\boldsymbol{\theta}_k) - J(\boldsymbol{\theta}_{k+1})\big)^2\big] \\
&\leq \sum_{k=\tau_t}^{t} \frac{1}{\gamma_k} C_J^2 \|\boldsymbol{\theta}_k - \boldsymbol{\theta}_{k+1}\|^2 \\
&\leq \sum_{k=\tau_t}^{t} \frac{1}{\gamma_k} C_J^2 G_{\boldsymbol{\theta}}^2 \alpha_k^2 \\
&= \mathcal{O}\bigg( \sum_{k=\tau_t}^{t} \frac{\alpha_k^2}{\gamma_k} \bigg).
\end{aligned}
$$

For $I_5$, we have

$$
\begin{aligned}
I_5 &= \sum_{k=\tau_t}^{t} \gamma_k \mathbb{E}[(r_k - \eta_k)^2] \\
&\leq \sum_{k=\tau_t}^{t} 4 U_r^2 \gamma_k \\
&= \mathcal{O}\bigg( \sum_{k=\tau_t}^{t} \gamma_k \bigg),
\end{aligned}
$$

by bounding the expectation uniformly.

Now, we set $\gamma_k = 1/(1+t)^\nu$ and combine all the terms together to get

$$
\begin{aligned}
\sum_{k=\tau_t}^{t} \mathbb{E}[y_k^2] &\leq 2 U_r^2 (1+t)^\nu + (C_1 \tau_t^2 + C_3) c_\alpha \sum_{k=\tau_t}^{t} (1+k)^{-\sigma} + C_2 \tau_t \sum_{k=\tau_t}^{t} (1+k)^{-\nu} \\
&\quad + C_J G_{\boldsymbol{\theta}} c_\alpha \bigg( \sum_{k=\tau_t}^{t} \mathbb{E}[y_k^2] \bigg)^{1/2} \bigg( \sum_{k=\tau_t}^{t} (1+k)^{-2(\sigma-\nu)} \bigg)^{1/2} \\
&\quad + C_J^2 G_{\boldsymbol{\theta}}^2 c_\alpha^2 \sum_{k=\tau_t}^{t} (1+k)^{\nu-2\sigma} + 4 U_r^2 \sum_{k=\tau_t}^{t} (1+k)^{-\nu} \\
&\leq 2 U_r^2 (1+t)^\nu + \big[ (C_1 \tau^2 + C_3) c_\alpha + C_2 \tau_t + C_J^2 G_{\boldsymbol{\theta}}^2 c_\alpha^2 + 4 U_r^2 \big] \sum_{k=\tau_t}^{t} (1+k)^{-\nu} \\
&\quad + C_J G_{\boldsymbol{\theta}} c_\alpha \bigg( \sum_{k=\tau_t}^{t} \mathbb{E}[y_k^2] \bigg)^{1/2} \bigg( \sum_{k=\tau_t}^{t} (1+k)^{-2(\sigma-\nu)} \bigg)^{1/2} \\
&\leq 2 U_r^2 (1+t)^\nu + \big[ (C_1 \tau^2 + C_3) c_\alpha + C_2 \tau_t + C_J^2 G_{\boldsymbol{\theta}}^2 c_\alpha^2 + 4 U_r^2 \big] \frac{(1+t-\tau_t)^{1-\nu}}{1-\nu} \\
&\quad + C_J G_{\boldsymbol{\theta}} c_\alpha \bigg( \sum_{k=\tau_t}^{t} \mathbb{E}[y_k^2] \bigg)^{1/2} \bigg( \frac{(1+t-\tau_t)^{1-2(\sigma-\nu)}}{1-2(\sigma-\nu)} \bigg)^{1/2}
\end{aligned}
$$

By applying the squaring technique already stated in the proof of Theorem 4.5, we have that

$$\sum_{k=\tau_t}^{t} \mathbb{E}[y_k^2] \le 4U_r^2(1+t)^\nu + 2\big[(C_1\tau_t^2 + C_3)c_\alpha + C_2\tau_t + C_J^2 G_{\boldsymbol{\theta}}^2 c_\alpha^2 + 4U_r^2\big] \frac{(1+t-\tau_t)^{1-\nu}}{1-\nu}$$

$$+ 8C_J^2 G_{\boldsymbol{\theta}}^2 c_\alpha^2 \frac{(1+t-\tau_t)^{1-2(\sigma-\nu)}}{1-2(\sigma-\nu)} \tag{C.6}$$

$$= \mathcal{O}(t^\nu) + \mathcal{O}(\log^2 t \cdot t^{1-\nu}) + \mathcal{O}(t^{1-2(\sigma-\nu)}).$$

$\square$

## C.3 Proof of Theorem 4.7: Approximating the TD Fixed Point

Now we deal with the critic's parameter $\boldsymbol{\omega}_t$. The two time-scale analysis with Markovian noise and moving behavior policy can be complicated, so we define some useful notations here that could hopefully clarify the probabilistic dependency.

$$O_t := (s_t, a_t, s_{t+1}),$$
$$g(O, \boldsymbol{\omega}, \boldsymbol{\theta}) := [r(s,a) - J(\boldsymbol{\theta}) + (\boldsymbol{\phi}(s') - \boldsymbol{\phi}(s))^\top \boldsymbol{\omega}]\boldsymbol{\phi}(s),$$
$$\Delta g(O, \eta, \boldsymbol{\theta}) := [J(\boldsymbol{\theta}) - \eta]\boldsymbol{\phi}(s),$$
$$\bar{g}(\boldsymbol{\omega}, \boldsymbol{\theta}) := \mathbb{E}_{s\sim\mu_{\boldsymbol{\theta}}, a\sim\pi_{\boldsymbol{\theta}}, s'\sim\mathcal{P}}\Big[\big[r(s,a) - J(\boldsymbol{\theta}) + \big(\boldsymbol{\phi}(s') - \boldsymbol{\phi}(s)\big)^\top \boldsymbol{\omega}\big]\boldsymbol{\phi}(s)\Big],$$
$$\boldsymbol{\omega}_t^* := \boldsymbol{\omega}^*(\boldsymbol{\theta}_t),$$
$$\eta_t^* := \eta^*(\boldsymbol{\theta}_t) = J(\boldsymbol{\theta}_t)$$
$$\Lambda(O, \boldsymbol{\omega}, \boldsymbol{\theta}) := \big\langle \boldsymbol{\omega} - \boldsymbol{\omega}^*(\boldsymbol{\theta}), g(O, \boldsymbol{\omega}, \boldsymbol{\theta}) - \bar{g}(\boldsymbol{\omega}, \boldsymbol{\theta})\big\rangle,$$
$$\mathbf{z}_t := \boldsymbol{\omega}_t - \boldsymbol{\omega}_t^*$$
$$y_t := \eta_t - \eta_t^*. \tag{C.7}$$

A bounded lemma is used frequently in this section.

**Lemma C.6.** Under Assumption 4.3, for any $\boldsymbol{\theta}$, $\boldsymbol{\omega}$, $O = (s, a, s')$ such that $\|\boldsymbol{\omega}\| \le R_{\boldsymbol{\omega}}$,

$$\big\|g(O, \boldsymbol{\omega}, \boldsymbol{\theta})\big\| \le U_\delta := 2U_r + 2R_{\boldsymbol{\omega}},$$
$$\big\|\Delta g(O, \eta, \boldsymbol{\theta})\big\| \le 2U_r,$$
$$\big|\Lambda(O, \boldsymbol{\omega}, \boldsymbol{\theta})\big| \le 2R_{\boldsymbol{\omega}} \cdot 2U_\delta \le 2U_\delta^2.$$

The following lemma is used to control the bias due to Markovian noise.

**Lemma C.7.** Given the definition of $\Lambda(\boldsymbol{\theta}_t, \boldsymbol{\omega}_t, O_t)$, for any $0 \le \tau \le t$, we have

$$\mathbb{E}[\Lambda(O_t, \boldsymbol{\omega}_t, \boldsymbol{\theta}_t)] \le C_1(\tau+1)\|\boldsymbol{\theta}_t - \boldsymbol{\theta}_{t-\tau}\| + C_2 m\rho^{\tau-1} + C_3\|\boldsymbol{\omega}_t - \boldsymbol{\omega}_{t-\tau}\|,$$

where $C_1 = 2U_\delta^2|\mathcal{A}|L(1 + \lceil\log_\rho m^{-1}\rceil + 1/(1-\rho)) + 2U_\delta L_*, C_2 = 2U_\delta^2, C_3 = 4U_\delta$ are constants.

*Proof of Theorem 4.7.* By the updating rule of $\boldsymbol{\omega}_t$ in Algorithm 1, unrolling and decomposing the squared error gives

$$\|\mathbf{z}_{t+1}\|^2 = \big\|\mathbf{z}_t + \beta_t(g(O_t, \boldsymbol{\omega}_t, \boldsymbol{\theta}_t) + \Delta g(O_t, \eta_t, \boldsymbol{\theta}_t)) + (\boldsymbol{\omega}_t^* - \boldsymbol{\omega}_{t+1}^*)\big\|^2$$
$$= \|\mathbf{z}_t\|^2 + 2\beta_t\langle\mathbf{z}_t, g(O_t, \boldsymbol{\omega}_t, \boldsymbol{\theta}_t)\rangle + 2\beta_t\langle\mathbf{z}_t, \Delta g(O_t, \eta_t, \boldsymbol{\theta}_t)\rangle$$
$$+ 2\langle\mathbf{z}_t, \boldsymbol{\omega}_t^* - \boldsymbol{\omega}_{t+1}^*\rangle + \big\|\beta_t(g(O_t, \boldsymbol{\omega}_t, \boldsymbol{\theta}_t) + \Delta g(O_t, \eta_t, \boldsymbol{\theta}_t)) + (\boldsymbol{\omega}_t^* - \boldsymbol{\omega}_{t+1}^*)\big\|^2$$
$$= \|\mathbf{z}_t\|^2 + 2\beta_t\langle\mathbf{z}_t, \bar{g}(\boldsymbol{\omega}_t, \boldsymbol{\theta}_t)\rangle + 2\beta_t\Lambda(O_t, \boldsymbol{\omega}_t, \boldsymbol{\theta}_t) + 2\beta_t\langle\mathbf{z}_t, \Delta g(O_t, \eta_t, \boldsymbol{\theta}_t)\rangle$$
$$+ 2\langle\mathbf{z}_t, \boldsymbol{\omega}_t^* - \boldsymbol{\omega}_{t+1}^*\rangle + \big\|\beta_t(g(O_t, \boldsymbol{\omega}_t, \boldsymbol{\theta}_t) + \Delta g(O_t, \eta_t, \boldsymbol{\theta}_t)) + (\boldsymbol{\omega}_t^* - \boldsymbol{\omega}_{t+1}^*)\big\|^2$$
$$\le \|\mathbf{z}_t\|^2 + 2\beta_t\langle\mathbf{z}_t, \bar{g}(\boldsymbol{\omega}_t, \boldsymbol{\theta}_t)\rangle + 2\beta_t\Lambda(O_t, \boldsymbol{\omega}_t, \boldsymbol{\theta}_t) + 2\beta_t\langle\mathbf{z}_t, \Delta g(O_t, \eta_t, \boldsymbol{\theta}_t)\rangle$$
$$+ 2\langle\mathbf{z}_t, \boldsymbol{\omega}_t^* - \boldsymbol{\omega}_{t+1}^*\rangle + 2\beta_t^2\big\|g(O_t, \boldsymbol{\omega}_t, \boldsymbol{\theta}_t) + \Delta g(O_t, \eta_t, \boldsymbol{\theta}_t)\big\|^2 + 2\|\boldsymbol{\omega}_t^* - \boldsymbol{\omega}_{t+1}^*\|^2$$
$$\le \|\mathbf{z}_t\|^2 + 2\beta_t\langle\mathbf{z}_t, \bar{g}(\boldsymbol{\omega}_t, \boldsymbol{\theta}_t)\rangle + 2\beta_t\Lambda(O_t, \boldsymbol{\omega}_t, \boldsymbol{\theta}_t) + 2\beta_t\langle\mathbf{z}_t, \Delta g(O_t, \eta_t, \boldsymbol{\theta}_t)\rangle$$
$$+ 2\langle\mathbf{z}_t, \boldsymbol{\omega}_t^* - \boldsymbol{\omega}_{t+1}^*\rangle + 2U_\delta^2\beta_t^2 + 2\|\boldsymbol{\omega}_t^* - \boldsymbol{\omega}_{t+1}^*\|^2,$$

where the first inequality is due to $\|\mathbf{x}+\mathbf{y}\|^2 \leq 2\|\mathbf{x}\|^2+2\|\mathbf{y}\|^2$ and the second is due to $\|g(O_t, \boldsymbol{\omega}_t, \boldsymbol{\theta}_t)+\Delta g(O_t, \eta_t, \boldsymbol{\theta}_t)\| \leq U_\delta$. First, note that due to Assumption 4.1, we have

$$
\begin{aligned}
\langle \mathbf{z}_t, \bar{g}(\boldsymbol{\omega}_t, \boldsymbol{\theta}_t) \rangle &= \langle \mathbf{z}_t, \bar{g}(\boldsymbol{\omega}_t, \boldsymbol{\theta}_t) - \bar{g}(\boldsymbol{\omega}_t^*, \boldsymbol{\theta}_t) \rangle \\
&= \Big\langle \mathbf{z}_t, \mathbb{E}\big[ \big(\boldsymbol{\phi}(s') - \boldsymbol{\phi}(s)\big)^\top (\boldsymbol{\omega}_t - \boldsymbol{\omega}_t^*) \boldsymbol{\phi}(s) \big] \Big\rangle \\
&= \mathbf{z}_t^\top \mathbb{E}\big[ \boldsymbol{\phi}(s) \big(\boldsymbol{\phi}(s') - \boldsymbol{\phi}(s)\big)^\top \big] \mathbf{z}_t \\
&= \mathbf{z}_t^\top \mathbf{A} \mathbf{z}_t \\
&\leq -\lambda \|\mathbf{z}_t\|^2,
\end{aligned}
$$

where the first equation is due to the fact that $\bar{g}(\boldsymbol{\omega}^*, \boldsymbol{\theta}) = 0$ [27]. Taking expectation up to $s_{t+1}$, we have

$$
\begin{aligned}
\mathbb{E}\|\mathbf{z}_{t+1}\|^2 &\leq \mathbb{E}\|\mathbf{z}_t\|^2 + 2\beta_t \mathbb{E}\langle \mathbf{z}_t, \bar{g}(\boldsymbol{\omega}_t, \boldsymbol{\theta}_t) \rangle + 2\beta_t \mathbb{E}\Lambda(O_t, \boldsymbol{\omega}_t, \boldsymbol{\theta}_t) + 2\beta_t \mathbb{E}\langle \mathbf{z}_t, \Delta g(O_t, \eta_t, \boldsymbol{\theta}_t) \rangle \\
&\quad + 2\mathbb{E}\langle \mathbf{z}_t, \boldsymbol{\omega}_t^* - \boldsymbol{\omega}_{t+1}^* \rangle + 2U_\delta^2 \beta_t^2 + 2\mathbb{E}\|\boldsymbol{\omega}_t^* - \boldsymbol{\omega}_{t+1}^*\|^2 \\
&\leq (1 - 2\lambda\beta_t)\mathbb{E}\|\mathbf{z}_t\|^2 + 2\beta_t \mathbb{E}\Lambda(O_t, \boldsymbol{\omega}_t, \boldsymbol{\theta}_t) + 2\beta_t \mathbb{E}\langle \mathbf{z}_t, \Delta g(O_t, \eta_t, \boldsymbol{\theta}_t) \rangle \\
&\quad + 2\mathbb{E}\langle \mathbf{z}_t, \boldsymbol{\omega}_t^* - \boldsymbol{\omega}_{t+1}^* \rangle + 2U_\delta^2 \beta_t^2 + 2\mathbb{E}\|\boldsymbol{\omega}_t^* - \boldsymbol{\omega}_{t+1}^*\|^2.
\end{aligned}
$$

Based on the result above, we can further rewrite it as:

$$
\begin{aligned}
\mathbb{E}\|\mathbf{z}_{t+1}\|^2 &\leq (1 - 2\lambda\beta_t)\mathbb{E}\|\mathbf{z}_t\|^2 + 2\beta_t \mathbb{E}\Lambda(O_t, \boldsymbol{\omega}_t, \boldsymbol{\theta}_t) + 2\beta_t \mathbb{E}\|\mathbf{z}_t\| \cdot |y_t| \\
&\quad + 2L_* \mathbb{E}\|\mathbf{z}_t\| \cdot \|\boldsymbol{\theta}_t - \boldsymbol{\theta}_{t+1}\| + 2U_\delta^2 \beta_t^2 + 2L_*^2 \mathbb{E}\|\boldsymbol{\theta}_t - \boldsymbol{\theta}_{t+1}\|^2 \\
&\leq (1 - 2\lambda\beta_t)\mathbb{E}\|\mathbf{z}_t\|^2 + 2\beta_t \mathbb{E}\Lambda(O_t, \boldsymbol{\omega}_t, \boldsymbol{\theta}_t) + 2\beta_t \mathbb{E}\|\mathbf{z}_t\| \cdot |y_t| \\
&\quad + 2L_* G_{\boldsymbol{\theta}} \alpha_t \mathbb{E}\|\mathbf{z}_t\| + 2U_\delta^2 \beta_t^2 + 2L_*^2 G_{\boldsymbol{\theta}}^2 \alpha_t^2 \\
&\leq (1 - 2\lambda\beta_t)\mathbb{E}\|\mathbf{z}_t\|^2 + 2\beta_t \mathbb{E}\Lambda(O_t, \boldsymbol{\omega}_t, \boldsymbol{\theta}_t) + 2\beta_t \mathbb{E}\|\mathbf{z}_t\| \cdot |y_t| \\
&\quad + 2L_* G_{\boldsymbol{\theta}} \alpha_t \mathbb{E}\|\mathbf{z}_t\| + \left( 2U_\delta^2 + 2L_*^2 G_{\boldsymbol{\theta}}^2 \Big( \max_t \frac{\alpha_t}{\beta_t} \Big)^2 \right) \beta_t^2 \\
&= (1 - 2\lambda\beta_t)\mathbb{E}\|\mathbf{z}_t\|^2 + 2\beta_t \mathbb{E}\Lambda(O_t, \boldsymbol{\omega}_t, \boldsymbol{\theta}_t) + 2\beta_t \mathbb{E}\|\mathbf{z}_t\| \cdot |y_t| + 2L_* G_{\boldsymbol{\theta}} \alpha_t \mathbb{E}\|\mathbf{z}_t\| + C_q \beta_t^2,
\end{aligned}
$$

where we denote the constant coefficient before the quadratic stepsize $\beta_t^2$ as $C_q$ at the last step. The first inequality is due to Proposition 4.4 and Cauchy-Schwartz inequality. The second inequality is due to the update of $\boldsymbol{\theta}_t$ is bounded by $G_{\boldsymbol{\theta}}\alpha_t$. The third inequality is from employing the fact that $\sigma > \nu$ so $\alpha_t/\beta_t$ is bounded. Rearranging the inequality yields

$$
\begin{aligned}
2\lambda \mathbb{E}\|\mathbf{z}_t\|^2 &\leq \frac{1}{\beta_t} \big( \mathbb{E}\|\mathbf{z}_t\|^2 - \mathbb{E}\|\mathbf{z}_{t+1}\|^2 \big) + 2\mathbb{E}\Lambda(O_t, \boldsymbol{\omega}_t, \boldsymbol{\theta}_t) + \mathbb{E}\|\mathbf{z}_t\| \cdot |y_t| + 2L_* G_{\boldsymbol{\theta}} \frac{\alpha_t}{\beta_t} \mathbb{E}\|\mathbf{z}_t\| + C_q \beta_t \\
&\leq \frac{1}{\beta_t} \big( \mathbb{E}\|\mathbf{z}_t\|^2 - \mathbb{E}\|\mathbf{z}_{t+1}\|^2 \big) + 2\mathbb{E}\Lambda(O_t, \boldsymbol{\omega}_t, \boldsymbol{\theta}_t) + \sqrt{\mathbb{E}y_t^2} \cdot \sqrt{\mathbb{E}\|\mathbf{z}_t\|^2} + 2L_* G_{\boldsymbol{\theta}} \frac{\alpha_t}{\beta_t} \sqrt{\mathbb{E}\|\mathbf{z}_t\|^2} + C_q \beta_t,
\end{aligned}
$$

where the second inequality is due to the concavity of square root function. Telescoping from $\tau_t$ to $t$ gives:

$$
2\lambda \sum_{k=\tau_t}^{t} \mathbb{E}\|\mathbf{z}_k\|^2 \leq \underbrace{\sum_{k=\tau_t}^{t} \frac{1}{\beta_k} \big( \mathbb{E}\|\mathbf{z}_k\|^2 - \mathbb{E}\|\mathbf{z}_{k+1}\|^2 \big)}_{I_1} + 2\underbrace{\sum_{k=\tau_t}^{t} \mathbb{E}\Lambda(\boldsymbol{\theta}_k, \boldsymbol{\omega}_k, O_k)}_{I_2}
$$

$$
+ 2L_* G_{\boldsymbol{\theta}} \underbrace{\sum_{k=\tau_t}^{t} \frac{\alpha_k}{\beta_k} \sqrt{\mathbb{E}\|\mathbf{z}_k\|^2}}_{I_3} + \underbrace{\sum_{k=\tau_t}^{t} \sqrt{\mathbb{E}y_k^2} \cdot \sqrt{\mathbb{E}\|\mathbf{z}_k\|^2}}_{I_4} + C_q \underbrace{\sum_{k=\tau_t}^{t} \beta_k}_{I_5}. \quad \text{(C.8)}
$$

From (C.8), we can see the proof of the critic again shares the same spirit with the proof of Theorem 4.5. For term $I_1$, we have

$$I_1 := \sum_{k=\tau_t}^{t} \frac{1}{\beta_k} (\mathbb{E}\|\mathbf{z}_k\|^2 - \mathbb{E}\|\mathbf{z}_{k+1}\|^2)$$

$$= \sum_{k=\tau_t}^{t} \left( \frac{1}{\beta_k} - \frac{1}{\beta_{k-1}} \right) \mathbb{E}\|\mathbf{z}_k\|^2 + \frac{1}{\beta_{\tau_t - 1}} \mathbb{E}\|\mathbf{z}_{\tau_t}\|^2 - \frac{1}{\beta_t} \mathbb{E}\|\mathbf{z}_{t+1}\|^2$$

$$\leq \sum_{k=\tau_t}^{t} \left( \frac{1}{\beta_k} - \frac{1}{\beta_{k-1}} \right) \mathbb{E}\|\mathbf{z}_k\|^2 + \frac{1}{\beta_{\tau_t - 1}} \mathbb{E}\|\mathbf{z}_{\tau_t}\|^2$$

$$\leq 4R_{\boldsymbol{\omega}}^2 \left( \sum_{k=\tau_t}^{t} \left( \frac{1}{\beta_k} - \frac{1}{\beta_{k-1}} \right) + \frac{1}{\beta_{\tau_t - 1}} \right)$$

$$= 4R_{\boldsymbol{\omega}}^2 \frac{1}{\beta_t}$$

$$= 4R_{\boldsymbol{\omega}}^2 (1+t)^{\nu} = \mathcal{O}(t^{\nu}),$$

where the first inequality is due to discarding the last term, and the second inequality is due to $\mathbb{E}\|\mathbf{z}_k\|^2 \leq (R_{\boldsymbol{\omega}} + R_{\boldsymbol{\omega}})^2$.

For term $I_2$, note that due to Lemma C.7, we actually have

$$\Lambda(O_k, \boldsymbol{\omega}_k, \boldsymbol{\theta}_k) \leq C_1(\tau_t + 1)\|\boldsymbol{\theta}_k - \boldsymbol{\theta}_{k-\tau_t}\| + C_2 m \rho^{\tau_t - 1} + C_3\|\boldsymbol{\omega}_k - \boldsymbol{\omega}_{k-\tau_t}\|$$

$$\leq C_1(\tau_t + 1) \sum_{i=k-\tau_t}^{k-1} G_{\boldsymbol{\theta}} \alpha_i + C_2 m \rho^{\tau_t - 1} + C_3 \sum_{i=k-\tau_t}^{k-1} U_{\delta} \beta_i$$

$$\leq C_1 G_{\boldsymbol{\theta}} (\tau_t + 1)^2 \alpha_{k-\tau_t} + C_2 \alpha_t + C_3 U_{\delta} \tau_t \beta_k,$$

and the summation is

$$I_2 := \sum_{k=\tau_t}^{t} \mathbb{E}\Lambda(O_k, \boldsymbol{\omega}_k, \boldsymbol{\theta}_k)$$

$$\leq C_1 G_{\boldsymbol{\theta}} (\tau_t + 1)^2 \sum_{k=\tau_t}^{t} \alpha_{k-\tau_t} + C_2 \sum_{k=\tau_t}^{t} \alpha_t + C_3 U_{\delta} \tau_t \sum_{k=\tau_t}^{t} \beta_k$$

$$\leq C_1 G_{\boldsymbol{\theta}} (\tau_t + 1)^2 \sum_{k=0}^{t-\tau_t} \alpha_k + C_2 (t - \tau_t + 1)\alpha_t + C_3 U_{\delta} \tau_t \sum_{k=0}^{t-\tau_t} \beta_k$$

$$\leq C_1 G_{\boldsymbol{\theta}} (\tau_t + 1)^2 c_{\alpha} \frac{(1+t-\tau_t)^{1-\sigma}}{1-\sigma} + C_2 (t - \tau_t + 1) c_{\alpha} (1+t)^{-\sigma} + C_3 U_{\delta} \tau_t \frac{(1+t-\tau_t)^{1-\nu}}{1-\nu}$$

$$\leq \left[ \frac{C_1 G_{\boldsymbol{\theta}} (\tau_t + 1)^2 c_{\alpha}}{1-\sigma} + C_2 c_{\alpha} + \frac{C_3 U_{\delta} \tau_t}{1-\nu} \right] (1+t)^{1-\nu}$$

$$= \mathcal{O}\big((\log t)^2 t^{1-\nu}\big),$$

where the second inequality is due to the monotonicity of $\alpha_k$ and $\beta_k$. The $\mathcal{O}(\cdot)$ comes from that $\tau = \mathcal{O}(\log t)$ and $\sum k^{-\nu} = \mathcal{O}(t^{1-\nu})$.

For term $I_3$ and $I_4$, we will instead show it can be bounded in a different form. Using Cauchy-Schwartz inequality we have

$$I_3 := \sum_{k=\tau_t}^{t} \frac{\alpha_k}{\beta_k} \sqrt{\mathbb{E}\|\mathbf{z}_k\|^2} \leq \left( \sum_{k=\tau_t}^{t} \frac{\alpha_k^2}{\beta_k^2} \right)^{\frac{1}{2}} \left( \sum_{k=\tau_t}^{t} \mathbb{E}\|\mathbf{z}_k\|^2 \right)^{\frac{1}{2}} \leq \left( \sum_{k=0}^{t-\tau_t} \frac{\alpha_k^2}{\beta_k^2} \right)^{\frac{1}{2}} \left( \sum_{k=\tau_t}^{t} \mathbb{E}\|\mathbf{z}_k\|^2 \right)^{\frac{1}{2}},$$

$$I_4 := \sum_{k=\tau_t}^{t} \sqrt{\mathbb{E}y_k^2} \cdot \sqrt{\mathbb{E}\|\mathbf{z}_k\|^2} \leq \left( \sum_{k=\tau_t}^{t} \mathbb{E}y_k^2 \right)^{\frac{1}{2}} \left( \sum_{k=\tau_t}^{t} \mathbb{E}\|\mathbf{z}_k\|^2 \right)^{\frac{1}{2}} \leq \left( \sum_{k=0}^{t-\tau_t} \mathbb{E}y_k^2 \right)^{\frac{1}{2}} \left( \sum_{k=\tau_t}^{t} \mathbb{E}\|\mathbf{z}_k\|^2 \right)^{\frac{1}{2}}.$$

For term $I_5$, simply bound it as $\sum_{k=0}^{t-\tau_t} \beta_k \leq (1+t)^{1-\nu}/(1-\nu)$.

Collecting the upper bounds of the above five terms, and writing them using $\mathcal{O}(\cdot)$ notation give

$$2\lambda \sum_{k=\tau_t}^{t} \mathbb{E}\|\mathbf{z}_k\|^2 \leq 4R_{\boldsymbol{\omega}}^2(1+t)^{\nu} + 2\left[\frac{C_1 G_{\boldsymbol{\theta}}(\tau_t+1)^2 c_{\alpha}}{1-\sigma} + C_2 c_{\alpha} + \frac{C_3 U_{\delta}\tau_t + C_q}{1-\nu}\right](1+t)^{1-\nu}$$

$$+ 2L_* G_{\boldsymbol{\theta}}\left(\sum_{k=0}^{t-\tau_t} \frac{\alpha_k^2}{\beta_k^2}\right)^{\frac{1}{2}}\left(\sum_{k=\tau_t}^{t} \mathbb{E}\|\mathbf{z}_k\|^2\right)^{\frac{1}{2}}$$

$$+ \left(\sum_{k=0}^{t-\tau_t} \mathbb{E}y_k^2\right)^{\frac{1}{2}}\left(\sum_{k=\tau_t}^{t} \mathbb{E}\|\mathbf{z}_k\|^2\right)^{\frac{1}{2}}. \tag{C.9}$$

Now, we first divide both sides by $(1+t-\tau_t)$, and denote

$$Z(t) := \frac{1}{1+t-\tau_t}\sum_{k=\tau_t}^{t}\mathbb{E}\|\mathbf{z}_k\|^2,$$

$$F(t) := \frac{1}{1+t-\tau_t}\sum_{k=0}^{t-\tau_t}\frac{\alpha_k^2}{\beta_k^2} \leq \frac{t^{-2(\sigma-\nu)}}{1-2(\sigma-\nu)} = \mathcal{O}(t^{-2(\sigma-\nu)}),$$

$$G(t) := \frac{1}{1+t-\tau_t}\sum_{k=0}^{t-\tau_t}\mathbb{E}[y_k^2] = \mathcal{O}(t^{\nu-1}) + \mathcal{O}(\log t \cdot t^{-\nu}) + \mathcal{O}(t^{-2(\sigma-\nu)}),$$

and the rest as $A(t) = \mathcal{O}(t^{\nu}) + \mathcal{O}(t^{1-\nu})$. $G(t)$'s constants appear at (C.6) in exact form. This simplification leads to

$$2\lambda\left(\sqrt{Z(t)} - \frac{L_* G_{\boldsymbol{\theta}}}{2\lambda}\cdot\sqrt{F(t)} - \frac{1}{4\lambda}\sqrt{G(t)}\right)^2 \leq A(t) + 2\lambda\left(\frac{L_* G_{\boldsymbol{\theta}}}{2\lambda}\sqrt{F(t)} + \frac{1}{4\lambda}\sqrt{G(t)}\right)^2,$$

which further gives

$$Z(t) \leq A(t)/\lambda + 16F(t) + 16G(t).$$

This is again a similar reasoning as in the end of the proof of Theorem 4.5. We actually show that

$$\frac{1}{1+t-\tau_t}\sum_{k=\tau_t}^{t}\mathbb{E}\|\boldsymbol{\omega}_k - \boldsymbol{\omega}_k^*\|^2 = \mathcal{O}\left(\frac{1}{t^{1-\nu}}\right) + \mathcal{O}\left(\frac{\log t}{t^{\nu}}\right) + \mathcal{O}\left(\frac{1}{t^{2(\sigma-\nu)}}\right).$$

This completes the proof. To obtain the exact constant, please refer to (C.6) and (C.9). $\qquad\square$

### C.4 Proof of Corollary 4.9

*Proof of Corollary 4.9.* By Theorem 4.7, we have

$$\frac{1}{1+t-\tau_t}\sum_{k=\tau_t}^{t}\mathbb{E}\|\boldsymbol{\omega}_k - \boldsymbol{\omega}_k^*\|^2 = \mathcal{O}\left(\frac{1}{t^{1-\nu}}\right) + \mathcal{O}\left(\frac{\log t}{t^{\nu}}\right) + \mathcal{O}\left(\frac{1}{t^{2(\sigma-\nu)}}\right).$$

By Lemma B.3, $\mathcal{E}(t)$ in Theorem 4.5 is of the equivalent order:

$$\mathcal{E}_1(t) = \frac{1}{t}\sum_{k=1}^{t}\mathbb{E}\|\boldsymbol{\omega}_k - \boldsymbol{\omega}_k^*\|^2$$

$$= \mathcal{O}\left(\frac{1}{1+t-\tau_t}\sum_{k=\tau_t}^{t}\mathbb{E}\|\boldsymbol{\omega}_k - \boldsymbol{\omega}_k^*\|^2\right) + \mathcal{O}\left(\frac{\log t}{t}\right)$$

$$= \mathcal{O}\left(\frac{1}{t^{1-\nu}}\right) + \mathcal{O}\left(\frac{\log t}{t^{\nu}}\right) + \mathcal{O}\left(\frac{1}{t^{2(\sigma-\nu)}}\right) + \mathcal{O}\left(\frac{\log t}{t}\right)$$

$$= \mathcal{O}\left(\frac{1}{t^{1-\nu}}\right) + \mathcal{O}\left(\frac{\log t}{t^{\nu}}\right) + \mathcal{O}\left(\frac{1}{t^{2(\sigma-\nu)}}\right).$$

The same reasoning also applies to

$$\mathcal{E}_2(t) = \frac{1}{t} \sum_{k=1}^{t} \mathbb{E}(\eta_k - r(\boldsymbol{\theta}_k))^2$$

$$= \mathcal{O}\left(\frac{1}{t^{1-\nu}}\right) + \mathcal{O}\left(\frac{\log t}{t^{\nu}}\right) + \mathcal{O}\left(\frac{1}{t^{2(\sigma-\nu)}}\right).$$

Plugging the above results into Theorem 4.5, and optimizing over the choice of $\sigma$ and $\nu$ (which gives $\sigma = 3/5$ and $\nu = 2/5$), we have

$$\min_{0 \leq k \leq t} \mathbb{E}\|\nabla J(\boldsymbol{\theta}_k)\|^2 = \mathcal{O}\left(\frac{1}{t^{1-\sigma}}\right) + \mathcal{O}\left(\frac{\log^2 t}{t^{\sigma}}\right) + \mathcal{O}\left(\frac{1}{t^{1-\nu}}\right) + \mathcal{O}\left(\frac{\log t}{t^{\nu}}\right) + \mathcal{O}\left(\frac{1}{t^{2(\sigma-\nu)}}\right) + \mathcal{O}(\epsilon_{\text{app}})$$

$$= \mathcal{O}\left(\frac{1}{t^{1-\sigma}}\right) + \mathcal{O}\left(\frac{\log t}{t^{\nu}}\right) + \mathcal{O}\left(\frac{1}{t^{2(\sigma-\nu)}}\right) + \mathcal{O}(\epsilon_{\text{app}})$$

$$= \mathcal{O}\left(\frac{\log t}{t^{2/5}}\right) + \mathcal{O}(\epsilon_{\text{app}}).$$

Therefore, in order to obtain an $\epsilon$-approximate(ignoring the approximation error) stationary point of $J$, namely,

$$\min_{0 \leq k \leq T} \mathbb{E}\|\nabla J(\boldsymbol{\theta}_k)\|^2 = \mathcal{O}\left(\frac{\log T}{T^{2/5}}\right) + \mathcal{O}(\epsilon_{\text{app}}) \leq \mathcal{O}(\epsilon_{\text{app}}) + \epsilon,$$

we need to set $T = \widetilde{\mathcal{O}}(\epsilon^{-2.5})$.  $\square$

# D Proof of Technical Lemmas

## D.1 Proof of Lemma C.1

*Proof of Lemma C.1.* The first inequality comes from Lemma 3.2 in Zhang et al. [42].

The second inequality is well known as a partial result of $[-L, L]$-smoothness of non-convex functions.
$\square$

## D.2 Proof of Lemma C.2

*Proof of Lemma C.2.* Applying the definition of $\Delta h()$ and Cauchy-Schwartz inequality immediately yields the result. $\square$

## D.3 Proof of Lemma C.3

The proof of Lemma C.3 will be built on the following supporting lemmas.

**Lemma D.1.** For any $t \geq 0$,
$$\left|\Gamma(O_t, \boldsymbol{\theta}_t) - \Gamma(O_t, \boldsymbol{\theta}_{t-\tau})\right| \leq G_{\boldsymbol{\theta}}(U_\delta L_l + 2L_*B + 3L_J)\|\boldsymbol{\theta}_t - \boldsymbol{\theta}_{t-\tau}\|.$$

**Lemma D.2.** For any $t \geq 0$,
$$\left|\mathbb{E}[\Gamma(O_t, \boldsymbol{\theta}_{t-\tau}) - \Gamma(\widetilde{O}_t, \boldsymbol{\theta}_{t-\tau})]\right| \leq 2U_\delta BG_{\boldsymbol{\theta}}|\mathcal{A}|L \sum_{i=t-\tau}^{t} \|\boldsymbol{\theta}_i - \boldsymbol{\theta}_{t-\tau}\|.$$

**Lemma D.3.** For any $t \geq 0$,
$$\left|\mathbb{E}[\Gamma(\widetilde{O}_t, \boldsymbol{\theta}_{t-\tau}) - \Gamma(O'_t, \boldsymbol{\theta}_{t-\tau})]\right| \leq 4U_\delta BG_{\boldsymbol{\theta}}m\rho^{\tau-1}.$$

*Proof of Lemma C.3.* First note that
$$\delta = \left|r(s,a) - J(\boldsymbol{\theta}) + \boldsymbol{\phi}^\top(s')\boldsymbol{\omega} - \boldsymbol{\phi}^\top(s)\boldsymbol{\omega}\right|$$
$$\leq \left|r(s,a)\right| + \left|J(\boldsymbol{\theta})\right| + \left|\boldsymbol{\phi}^\top(s')\boldsymbol{\omega}\right| + \left|\boldsymbol{\phi}^\top(s)\boldsymbol{\omega}\right|$$
$$= 2U_r + 2R_{\boldsymbol{\omega}}$$
$$=: U_\delta,$$

which immediately implies

$$\big\|\delta \nabla \log \pi_{\boldsymbol{\theta}}(a|s)\big\| \le |\delta| \cdot \big\|\nabla \log \pi_{\boldsymbol{\theta}}(a|s)\big\| \le U_{\delta} \cdot B,$$

where the last inequality is due to Assumption 4.3. We decompose the Markovian bias as

$$\mathbb{E}[\Gamma(O_t, \boldsymbol{\theta}_t)] = \mathbb{E}[\Gamma(O_t, \boldsymbol{\theta}_t) - \Gamma(O_t, \boldsymbol{\theta}_{t-\tau})] + \mathbb{E}[\Gamma(O_t, \boldsymbol{\theta}_{t-\tau}) - \Gamma(\widetilde{O}_t, \boldsymbol{\theta}_{t-\tau})]$$
$$+ \mathbb{E}[\Gamma(\widetilde{O}_t, \boldsymbol{\theta}_{t-\tau}) - \Gamma(O_t', \boldsymbol{\theta}_{t-\tau})] + \mathbb{E}[\Gamma(O_t', \boldsymbol{\theta}_{t-\tau})],$$

where $\widetilde{O}_t$ is from the auxiliary Markovian chain and $O_t'$ is from the stationary distribution which actually satisfy $\mathbb{E}[\Gamma(O_t', \boldsymbol{\theta}_{t-\tau})] = 0$. By collecting the corresponding bounds from Lemmas D.1, D.2 and D.3, we have that

$$\mathbb{E}[\Gamma(O_t, \boldsymbol{\theta}_t)] \ge -G_{\boldsymbol{\theta}}(U_{\delta}L_l + 2L_*B + 3L_J)\mathbb{E}\|\boldsymbol{\theta}_t - \boldsymbol{\theta}_{t-\tau}\| - 2U_{\delta}BG_{\boldsymbol{\theta}}|\mathcal{A}|L \sum_{i=t-\tau}^{t} \mathbb{E}\|\boldsymbol{\theta}_i - \boldsymbol{\theta}_{t-\tau}\|$$

$$- 4U_{\delta}BG_{\boldsymbol{\theta}}m\rho^{\tau-1}$$

$$\ge -G_{\boldsymbol{\theta}}(U_{\delta}L_l + 2L_*B + 3L_J) \sum_{i=t-\tau+1}^{t} \mathbb{E}\|\boldsymbol{\theta}_i - \boldsymbol{\theta}_{i-1}\|$$

$$- 2U_{\delta}BG_{\boldsymbol{\theta}}|\mathcal{A}|L \sum_{i=t-\tau+1}^{t} \sum_{j=t-\tau+1}^{i} \mathbb{E}\|\boldsymbol{\theta}_j - \boldsymbol{\theta}_{j-1}\| - 4U_{\delta}BG_{\boldsymbol{\theta}}m\rho^{\tau-1}$$

$$\ge -G_{\boldsymbol{\theta}}(U_{\delta}L_l + 2L_*B + 3L_J) \sum_{i=t-\tau+1}^{t} \mathbb{E}\|\boldsymbol{\theta}_i - \boldsymbol{\theta}_{i-1}\|$$

$$- 2U_{\delta}BG_{\boldsymbol{\theta}}|\mathcal{A}|L\tau \sum_{j=t-\tau+1}^{t} \mathbb{E}\|\boldsymbol{\theta}_j - \boldsymbol{\theta}_{j-1}\| - 4U_{\delta}BG_{\boldsymbol{\theta}}m\rho^{\tau-1}$$

$$\ge -G_{\boldsymbol{\theta}}\left( D_1(\tau+1) \sum_{k=t-\tau+1}^{t} \mathbb{E}\|\boldsymbol{\theta}_k - \boldsymbol{\theta}_{k-1}\| + D_2 m\rho^{\tau-1} \right),$$

where $D_1 := \max\{(U_{\delta}L_l + 2L_*B + 3L_J), 2U_{\delta}B|\mathcal{A}|L\}$ and $D_2 := 4U_{\delta}B$, which completes the proof. $\qquad\square$

## D.4 Proof of Lemma C.4

*Proof of Lemma C.4.* By definition, we have

$$J(\boldsymbol{\theta}_1) - J(\boldsymbol{\theta}_2) = \mathbb{E}[r(s^{(1)}, a^{(1)}) - r(s^{(2)}, a^{(2)})],$$

where $s^{(i)} \sim \mu_{\boldsymbol{\theta}_i}, a^{(i)} \sim \pi_{\boldsymbol{\theta}_i}$. Therefore, it holds that

$$J(\boldsymbol{\theta}_1) - J(\boldsymbol{\theta}_2) = \mathbb{E}[r(s^{(1)}, a^{(1)}) - r(s^{(2)}, a^{(2)})]$$
$$\le 2U_r d_{TV}(\mu_{\boldsymbol{\theta}_1} \otimes \pi_{\boldsymbol{\theta}_1}, \mu_{\boldsymbol{\theta}_2} \otimes \pi_{\boldsymbol{\theta}_2})$$
$$\le 2U_r|\mathcal{A}|L\left( 1 + \lceil \log_{\rho} m^{-1} \rceil + \frac{1}{1-\rho} \right)\|\boldsymbol{\theta}_1 - \boldsymbol{\theta}_2\|$$
$$= C_J\|\boldsymbol{\theta}_1 - \boldsymbol{\theta}_2\|.$$

$\qquad\square$

## D.5 Proof of Lemma C.5

The proof of this lemma depends on several auxiliary lemmas as follows.

**Lemma D.4.** For any $\boldsymbol{\theta}_1, \boldsymbol{\theta}_2, eta, O = (s, a, s')$, we have

$$\big|\Xi(O, \eta, \boldsymbol{\theta}_1) - \Xi(O, \eta, \boldsymbol{\theta}_2)\big| \le 4U_r C_J\|\boldsymbol{\theta}_1 - \boldsymbol{\theta}_2\|.$$

**Lemma D.5.** For any $\boldsymbol{\theta}, \eta_1, \eta_2, O$, we have
$$\left|\Xi(O, \eta_1, \boldsymbol{\theta}) - \Xi(O, \eta_2, \boldsymbol{\theta})\right| \le 2U_r|\eta_1 - \eta_2|.$$

**Lemma D.6.** Consider original tuples $O_t = (s_t, a_t, s_{t+1})$ and the auxiliary tuples $\widetilde{O}_t = (\widetilde{s}_t, \widetilde{a}_t, \widetilde{s}_{t+1})$. Conditioned on $s_{t-\tau+1}$ and $\boldsymbol{\theta}_{t-\tau}$, we have

$$\left|\mathbb{E}[\Xi(O_t, \eta_{t-\tau}, \boldsymbol{\theta}_{t-\tau}) - \Xi(\widetilde{O}_t, \eta_{t-\tau}, \boldsymbol{\theta}_{t-\tau})]\right| \le 2U_r^2|\mathcal{A}|L \sum_{i=t-\tau}^{t} \mathbb{E}\|\boldsymbol{\theta}_i - \boldsymbol{\theta}_{t-\tau}\|.$$

**Lemma D.7.** Conditioned on $s_{t-\tau+1}$ and $\boldsymbol{\theta}_{t-\tau}$, we have

$$\mathbb{E}[\Xi(\widetilde{O}_t, \eta_{t-\tau}, \boldsymbol{\theta}_{t-\tau})] \le 4U_r^2 m \rho^{\tau-1}.$$

*Proof.* By the Lemma D.4, D.5, D.6 and D.7, we can collect the corresponding term and get the bound

$$\mathbb{E}[\Xi(O_t, \eta_t, \boldsymbol{\theta}_t)] = \mathbb{E}[\Xi(O_t, \eta_t, \boldsymbol{\theta}_t) - \Xi(O_t, \eta_t, \boldsymbol{\theta}_{t-\tau})] + \mathbb{E}[\Xi(O_t, \eta_t, \boldsymbol{\theta}_{t-\tau}) - \Xi(O_t, \eta_{t-\tau}, \boldsymbol{\theta}_{t-\tau})]$$
$$+ \mathbb{E}[\Xi(O_t, \eta_{t-\tau}, \boldsymbol{\theta}_{t-\tau}) - \Xi(\widetilde{O}_t, \eta_{t-\tau}, \boldsymbol{\theta}_{t-\tau})] + \mathbb{E}[\Xi(\widetilde{O}_t, \eta_{t-\tau}, \boldsymbol{\theta}_{t-\tau})]$$
$$\le 4U_r C_J \|\boldsymbol{\theta}_t - \boldsymbol{\theta}_{t-\tau}\| + 2U_r|\eta_t - \eta_{t-\tau}| + 2U_r^2|\mathcal{A}|L \sum_{i=t-\tau}^{t} \mathbb{E}\|\boldsymbol{\theta}_i - \boldsymbol{\theta}_{t-\tau}\| + 4U_r^2 m \rho^{\tau-1}.$$

$\square$

## D.6 Proof of Lemma C.6

*Proof of Lemma C.6.* For the first inequality, apply the property of norm and the Cauchy-Schwartz inequality:

$$\|g(O, \boldsymbol{\omega}, \boldsymbol{\theta})\| = \|(r(s, a) - J(\boldsymbol{\theta}) + \boldsymbol{\phi}^\top(s')\boldsymbol{\omega} - \boldsymbol{\phi}^\top(s)\boldsymbol{\omega})\boldsymbol{\phi}(s)\|$$
$$\le |r(s, a)| + \|J(\boldsymbol{\theta})\| + |\boldsymbol{\phi}^\top(s')\boldsymbol{\omega}| \cdot \|\boldsymbol{\phi}^\top(s)\| + |\boldsymbol{\phi}^\top(s)\boldsymbol{\omega}| \cdot \|\boldsymbol{\phi}^\top(s)\|$$
$$= U_r + U_r + R_{\boldsymbol{\omega}} + R_{\boldsymbol{\omega}} \le 2U_r + 2R_{\boldsymbol{\omega}}.$$

For the second inequality, we can directly apply Cauchy-Schwartz inequality and obtain the result. For the third inequality, apply Cauchy-Schwartz inequality as we have

$$\left|\Lambda(O, \boldsymbol{\omega}, \boldsymbol{\theta})\right| = \left|\langle \boldsymbol{\omega} - \boldsymbol{\omega}^*, g(O, \boldsymbol{\omega}, \boldsymbol{\theta}) - \bar{g}(\boldsymbol{\omega}, \boldsymbol{\theta})\rangle\right|$$
$$\le \|\boldsymbol{\omega} - \boldsymbol{\omega}^*\| \cdot \|g(O, \boldsymbol{\omega}, \boldsymbol{\theta}) - \bar{g}(\boldsymbol{\omega}, \boldsymbol{\theta})\|$$
$$\le 2R_{\boldsymbol{\omega}} \cdot 2U_\delta \le 2U_\delta^2,$$

which completes the proof. $\square$

## D.7 Proof of Lemma C.7

This Lemma is actually a combination of several auxiliary lemmas listed here:

**Lemma D.8.** For any $\boldsymbol{\theta}_1, \boldsymbol{\theta}_2, \boldsymbol{\omega}$ and tuple $O = (s, a, s')$,
$$\left|\Lambda(O, \boldsymbol{\omega}, \boldsymbol{\theta}_1) - \Lambda(O, \boldsymbol{\omega}, \boldsymbol{\theta}_2)\right| \le K_1\|\boldsymbol{\theta}_1 - \boldsymbol{\theta}_2\|,$$
where $K_1 = 2U_\delta^2|\mathcal{A}|L(1 + \lceil\log_\rho m^{-1}\rceil + 1/(1-\rho)) + 2U_\delta L_*$.

**Lemma D.9.** For any $\boldsymbol{\theta}, \boldsymbol{\omega}_1, \boldsymbol{\omega}_2$ and tuple $O = (s, a, s')$,
$$\left|\Lambda(O, \boldsymbol{\omega}_1, \boldsymbol{\theta}) - \Lambda(O, \boldsymbol{\omega}_2, \boldsymbol{\theta})\right| \le 6U_\delta\|\boldsymbol{\omega}_1 - \boldsymbol{\omega}_2\|.$$

**Lemma D.10.** Consider original tuples $O_t = (s_t, a_t, s_{t+1})$ and the auxiliary tuples $\widetilde{O}_t = (\widetilde{s}_t, \widetilde{a}_t, \widetilde{s}_{t+1})$. Conditioned on $s_{t-\tau+1}$ and $\boldsymbol{\theta}_{t-\tau}$, we have

$$\mathbb{E}[\Lambda(O_t, \boldsymbol{\omega}_{t-\tau}, \boldsymbol{\theta}_{t-\tau}) - \Lambda(\widetilde{O}_t, \boldsymbol{\omega}_{t-\tau}, \boldsymbol{\theta}_{t-\tau})] \le U_\delta^2|\mathcal{A}|L \sum_{i=t-\tau}^{t} \mathbb{E}\|\boldsymbol{\theta}_i - \boldsymbol{\theta}_{t-\tau}\| \qquad \text{(D.1)}$$

**Lemma D.11.** Conditioned on $s_{t-\tau+1}$ and $\boldsymbol{\theta}_{t-\tau}$,
$$\mathbb{E}[\Lambda(\widetilde{O}_t, \boldsymbol{\omega}_{t-\tau}, \boldsymbol{\theta}_{t-\tau})] \leq 2U_\delta^2 m\rho^{\tau-1}.$$

*Proof of Lemma C.7.* By the Lemma D.8, D.9, D.10 and D.11, we can collect the corresponding term and get the bound

$$\mathbb{E}[\Lambda(O_t, \boldsymbol{\omega}_t, \boldsymbol{\theta}_t)] = \mathbb{E}[\Lambda(O_t, \boldsymbol{\omega}_t, \boldsymbol{\theta}_t) - \Lambda(O_t, \boldsymbol{\omega}_t, \boldsymbol{\theta}_{t-\tau})] + \mathbb{E}[\Lambda(O_t, \boldsymbol{\omega}_t, \boldsymbol{\theta}_{t-\tau}) - \Lambda(O_t, \boldsymbol{\omega}_{t-\tau}, \boldsymbol{\theta}_{t-\tau})]$$
$$+ \mathbb{E}[\Lambda(O_t, \boldsymbol{\omega}_{t-\tau}, \boldsymbol{\theta}_{t-\tau}) - \Lambda(\widetilde{O}_t, \boldsymbol{\omega}_{t-\tau}, \boldsymbol{\theta}_{t-\tau})] + \mathbb{E}[\Lambda(\widetilde{O}_t, \boldsymbol{\omega}_{t-\tau}, \boldsymbol{\theta}_{t-\tau})]$$
$$\leq C_1(\tau+1)\|\boldsymbol{\theta}_t - \boldsymbol{\theta}_{t-\tau}\| + C_2 m\rho^{\tau-1} + C_3\|\boldsymbol{\omega}_t - \boldsymbol{\omega}_{t-\tau}\|,$$

where $C_1 = 2U_\delta^2|\mathcal{A}|L(1 + \lceil\log_\rho m^{-1}\rceil + 1/(1-\rho)) + 2U_\delta L_*, C_2 = 2U_\delta^2, C_3 = 4U_\delta$. $\qquad\square$

# E    Proof of Auxiliary Lemmas

## E.1    Proof of Lemma D.1

*Proof of Lemma D.1.* Let $\delta(O_t, \boldsymbol{\theta}) := r(s_t, a_t) + (\phi(s_{t+1}) - \phi(s_t))^\top \boldsymbol{\omega}^* - r(\boldsymbol{\theta})$ and it can be shown that $\delta(O_t, \boldsymbol{\theta}_1) - \delta(O_t, \boldsymbol{\theta}_2) = (\phi(s_{t+1}) - \phi(s_t))^\top(\boldsymbol{\omega}_1^* - \boldsymbol{\omega}_2^*) - (r(\boldsymbol{\theta}_1) - r(\boldsymbol{\theta}_2))$.

$$\|h(O_t, \boldsymbol{\theta}_t) - h(O_t, \boldsymbol{\theta}_{t-\tau})\| = \|\delta(O_t, \boldsymbol{\theta}_t)\nabla\log\pi_{\boldsymbol{\theta}_t}(a_t|s_t) - \delta(O_t, \boldsymbol{\theta}_{t-\tau})\nabla\log\pi_{\boldsymbol{\theta}_{t-\tau}}(a_t|s_t)\|$$
$$\leq \|\delta(O_t, \boldsymbol{\theta}_t)\nabla\log\pi_{\boldsymbol{\theta}_t}(a_t|s_t) - \delta(O_t, \boldsymbol{\theta}_t)\nabla\log\pi_{\boldsymbol{\theta}_{t-\tau}}(a_t|s_t)\|$$
$$+ \|\delta(O_t, \boldsymbol{\theta}_t)\nabla\log\pi_{\boldsymbol{\theta}_{t-\tau}}(a_t|s_t) - \delta(O_t, \boldsymbol{\theta}_{t-\tau})\nabla\log\pi_{\boldsymbol{\theta}_{t-\tau}}(a_t|s_t)\|$$
$$\leq U_\delta L_l\|\boldsymbol{\theta}_t - \boldsymbol{\theta}_{t-\tau}\| + 2L_*B\|\boldsymbol{\theta}_t - \boldsymbol{\theta}_{t-\tau}\|.$$

By triangle inequality, we have

$$|\Gamma(O_t, \boldsymbol{\theta}_t) - \Gamma(O_t, \boldsymbol{\theta}_{t-\tau})| \leq G_{\boldsymbol{\theta}}\|h(O_t, \boldsymbol{\theta}_t) - h(O_t, \boldsymbol{\theta}_{t-\tau})\| + 3G_{\boldsymbol{\theta}}\|\nabla J(\boldsymbol{\theta}_t) - \nabla J(\boldsymbol{\theta}_{t-\tau})\|$$
$$\leq G_{\boldsymbol{\theta}}(U_\delta L_l + 2L_*B + 3L_J)\|\boldsymbol{\theta}_t - \boldsymbol{\theta}_{t-\tau}\|.$$

$\qquad\square$

## E.2    Proof of Lemma D.2

*Proof of Lemma D.2.* By the definition of in (C.1),

$$\mathbb{E}[\Gamma(O_t, \boldsymbol{\theta}_{t-\tau}) - \Gamma(\widetilde{O}_t, \boldsymbol{\theta}_{t-\tau})] = \mathbb{E}[\langle\nabla J(\boldsymbol{\theta}_{t-\tau}), h(O_t, \boldsymbol{\theta}_{t-\tau}) - h(\widetilde{O}_t, \boldsymbol{\theta}_{t-\tau})\rangle]$$
$$= \mathbb{E}[\langle\nabla J(\boldsymbol{\theta}_{t-\tau}), h(O_t, \boldsymbol{\theta}_{t-\tau})\rangle - \langle\nabla J(\boldsymbol{\theta}_{t-\tau}), h(\widetilde{O}_t, \boldsymbol{\theta}_{t-\tau})\rangle]$$
$$\leq 4U_\delta BG_{\boldsymbol{\theta}}d_{TV}(\mathbb{P}(O_t = \cdot|s_{t-\tau+1}, \boldsymbol{\theta}_{t-\tau}), \mathbb{P}(\widetilde{O}_t = \cdot|s_{t-\tau+1}, \boldsymbol{\theta}_{t-\tau})),$$
$$\text{(E.1)}$$

where the inequality is by the definition of total variation. By Lemma B.2 we have

$$d_{TV}(\mathbb{P}(O_t \in \cdot|s_{t-\tau+1}, \boldsymbol{\theta}_{t-\tau}), \mathbb{P}(\widetilde{O}_t \in \cdot|s_{t-\tau+1}, \boldsymbol{\theta}_{t-\tau}))$$
$$= d_{TV}(\mathbb{P}((s_t, a_t) \in \cdot|s_{t-\tau+1}, \boldsymbol{\theta}_{t-\tau}), \mathbb{P}((\widetilde{s}_t, \widetilde{a}_t) \in \cdot|s_{t-\tau+1}, \boldsymbol{\theta}_{t-\tau}))$$
$$\leq d_{TV}(\mathbb{P}(s_t \in \cdot|s_{t-\tau+1}, \boldsymbol{\theta}_{t-\tau}), \mathbb{P}(\widetilde{s}_t \in \cdot|s_{t-\tau+1}, \boldsymbol{\theta}_{t-\tau})) + \frac{1}{2}|\mathcal{A}|L\mathbb{E}\|\boldsymbol{\theta}_t - \boldsymbol{\theta}_{t-\tau}\|$$
$$\leq d_{TV}(\mathbb{P}(O_{t-1} \in \cdot|s_{t-\tau+1}, \boldsymbol{\theta}_{t-\tau}), \mathbb{P}(\widetilde{O}_{t-1} \in \cdot|s_{t-\tau+1}, \boldsymbol{\theta}_{t-\tau})) + \frac{1}{2}|\mathcal{A}|L\mathbb{E}\|\boldsymbol{\theta}_t - \boldsymbol{\theta}_{t-\tau}\|.$$

Repeat the inequality above over $t$ to $t-\tau+1$ we have

$$d_{TV}(\mathbb{P}(O_t \in \cdot|s_{t-\tau+1}, \boldsymbol{\theta}_{t-\tau}), \mathbb{P}(\widetilde{O}_t \in \cdot|s_{t-\tau+1}, \boldsymbol{\theta}_{t-\tau})) \leq \frac{1}{2}|\mathcal{A}|L\sum_{i=t-\tau}^t \mathbb{E}\|\boldsymbol{\theta}_i - \boldsymbol{\theta}_{t-\tau}\|. \quad \text{(E.2)}$$

Plugging (E.2) into (E.1) we get

$$\mathbb{E}[\Gamma(O_t, \boldsymbol{\theta}_{t-\tau}) - \Gamma(\widetilde{O}_t, \boldsymbol{\theta}_{t-\tau})] \leq 2U_\delta BG_{\boldsymbol{\theta}}|\mathcal{A}|L\sum_{i=t-\tau}^t \|\boldsymbol{\theta}_i - \boldsymbol{\theta}_{t-\tau}\|.$$

$\qquad\square$

### E.3 Proof of Lemma D.3

*Proof of Lemma D.3.*

$$\mathbb{E}\big[\Gamma(\widetilde{O}_t, \boldsymbol{\theta}_{t-\tau}) - \Gamma(O'_t, \boldsymbol{\theta}_{t-\tau})\big] \leq 4U_\delta B G_{\boldsymbol{\theta}} d_{TV}\big(\mathbb{P}(\widetilde{O}_t = \cdot | s_{t-\tau+1}, \boldsymbol{\theta}_{t-\tau}), \mu_{\boldsymbol{\theta}_{t-\tau}} \otimes \pi_{\boldsymbol{\theta}_{t-\tau}} \otimes \mathcal{P}\big)$$
$$\leq 4U_\delta B G_{\boldsymbol{\theta}} m \rho^{\tau-1}.$$

The first inequality is by the definition of total variation norm and the second inequality is shown in Lemma D.11. □

### E.4 Proof of Lemma D.4

*Proof of Lemma D.4.* By the definition of $\Xi(O, \eta, \boldsymbol{\theta})$ in (C.5), we have

$$\begin{aligned}
\big|\Xi(O, \eta, \boldsymbol{\theta}_1) - \Xi(O, \eta, \boldsymbol{\theta}_2)\big| &= \big|(\eta - \eta_1^*)(r - \eta_1^*) - (\eta - \eta_2^*)(r - \eta_2^*)\big| \\
&\leq \big|(\eta - \eta_1^*)(r - \eta_1^*) - (\eta - \eta_1^*)(r - \eta_2^*)\big| \\
&\quad + \big|(\eta - \eta_1^*)(r - \eta_2^*) - (\eta - \eta_2^*)(r - \eta_2^*)\big| \\
&\leq 4U_r|\eta_1^* - \eta_2^*| \\
&= 4U_r\big|J(\boldsymbol{\theta}_1) - J(\boldsymbol{\theta}_2)\big| \\
&\leq 4U_r C_J \|\boldsymbol{\theta}_1 - \boldsymbol{\theta}_2\|.
\end{aligned}$$

□

### E.5 Proof of Lemma D.5

*Proof of Lemma D.5.* By definition,

$$\begin{aligned}
\big|\Xi(O, \eta_1, \boldsymbol{\theta}) - \Xi(O, \eta_2, \boldsymbol{\theta})\big| &= \big|(\eta_1 - \eta^*)(r - \eta^*) - (\eta_2 - \eta^*)(r - \eta^*)\big| \\
&\leq 2U_r|\eta_1 - \eta_2|.
\end{aligned}$$

□

### E.6 Proof of Lemma D.6

*Proof of Lemma D.6.* By the Cauchy-Schwartz inequality and the definition of total variation norm, we have

$$\mathbb{E}\big[\Xi(O_t, \eta_{t-\tau}, \boldsymbol{\theta}_{t-\tau}) - \Xi(\widetilde{O}_t, \eta_{t-\tau}, \boldsymbol{\theta}_{t-\tau})\big] = (\eta_{t-\tau} - \eta_{t-\tau}^*)\mathbb{E}[r(s_t, a_t) - r(\widetilde{s}_t, \widetilde{a}_t)].$$

Since

$$\mathbb{E}[r(s_t, a_t) - r(\widetilde{s}_t, \widetilde{a}_t)] \leq 2U_r d_{TV}\big(\mathbb{P}(O_t \in \cdot | s_{t-\tau+1}, \boldsymbol{\theta}_{t-\tau}), \mathbb{P}(\widetilde{O}_t \in \cdot | s_{t-\tau+1}, \boldsymbol{\theta}_{t-\tau})\big),$$

the total variation between $O_t$ and $\widetilde{O}_t$ has appeared in (E.2), in the proof of Lemma D.2, which is

$$d_{TV}\big(\mathbb{P}(O_t \in \cdot | s_{t-\tau+1}, \boldsymbol{\theta}_{t-\tau}), \mathbb{P}(\widetilde{O}_t \in \cdot | s_{t-\tau+1}, \boldsymbol{\theta}_{t-\tau})\big) \leq \frac{1}{2}|\mathcal{A}|L \sum_{i=t-\tau}^{t} \mathbb{E}\|\boldsymbol{\theta}_i - \boldsymbol{\theta}_{t-\tau}\|.$$

Plugging this bound, we have

$$\big|\mathbb{E}[\Xi(O_t, \eta_{t-\tau}, \boldsymbol{\theta}_{t-\tau}) - \Xi(\widetilde{O}_t, \eta_{t-\tau}, \boldsymbol{\theta}_{t-\tau})]\big| \leq 2U_r^2|\mathcal{A}|L \sum_{i=t-\tau}^{t} \mathbb{E}\|\boldsymbol{\theta}_i - \boldsymbol{\theta}_{t-\tau}\|.$$

□

## E.7 Proof of Lemma D.7

*Proof of Lemma D.7.* We first note that according to the definition,
$$\mathbb{E}[\eta(O'_t, \eta_{t-\tau}, \boldsymbol{\theta}_{t-\tau})|\boldsymbol{\theta}_{t-\tau}] = 0,$$
where $O'_t = (s'_t, a'_t, s'_{t+1})$ is the tuple generated by $s'_t \sim \mu_{\boldsymbol{\theta}_{t-\tau}}, a'_t \sim \pi_{\boldsymbol{\theta}_{t-\tau}}, s'_{t+1} \sim \mathcal{P}$. By the ergodicity in Assumption 4.2, it holds that
$$d_{TV}\big(\mathbb{P}(\widetilde{s}_t = \cdot|s_{t-\tau+1}, \boldsymbol{\theta}_{t-\tau}), \mu_{\boldsymbol{\theta}_{t-\tau}}\big) \leq m\rho^{\tau-1}.$$
It can be shown that
$$
\begin{aligned}
\mathbb{E}[\Xi(\widetilde{O}_t, \eta_{t-\tau}, \boldsymbol{\theta}_{t-\tau})] &= \mathbb{E}\big[\Xi(\widetilde{O}_t, \eta_{t-\tau}, \boldsymbol{\theta}_{t-\tau}) - \Xi(O'_t, \eta_{t-\tau}, \boldsymbol{\theta}_{t-\tau})\big] \\
&= \mathbb{E}\big[(\eta_{t-\tau} - \eta^*_{t-\tau})\big(r(\widetilde{s}_t, \widetilde{a}_t) - r(s', a')\big)\big] \\
&\leq 4U_r^2 d_{TV}\big(\mathbb{P}(\widetilde{O}_t = \cdot|s_{t-\tau+1}, \boldsymbol{\theta}_{t-\tau}), \mu_{\boldsymbol{\theta}_{t-\tau}} \otimes \pi_{\boldsymbol{\theta}_{t-\tau}} \otimes \mathcal{P}\big) \\
&\leq 4U_r^2 m\rho^{\tau-1}.
\end{aligned}
$$
The argument used here also appears in the proof of Lemma D.11 and explained in detail there. $\square$

## E.8 Proof of Lemma D.8

*Proof of Lemma D.8.*
$$
\begin{aligned}
\big|\Lambda(O, \boldsymbol{\omega}, \boldsymbol{\theta}_1) - \Lambda(O, \boldsymbol{\omega}, \boldsymbol{\theta}_2)\big| &= \Big|\langle \boldsymbol{\omega} - \boldsymbol{\omega}_1^*, g(O, \boldsymbol{\omega}) - \bar{g}(\boldsymbol{\theta}_1, \boldsymbol{\omega})\rangle - \langle \boldsymbol{\omega} - \boldsymbol{\omega}_2^*, g(O, \boldsymbol{\omega}) - \bar{g}(\boldsymbol{\theta}_2, \boldsymbol{\omega})\rangle\Big| \\
&\leq \underbrace{\Big|\langle \boldsymbol{\omega} - \boldsymbol{\omega}_1^*, g(O, \boldsymbol{\omega}) - \bar{g}(\boldsymbol{\theta}_1, \boldsymbol{\omega})\rangle - \langle \boldsymbol{\omega} - \boldsymbol{\omega}_1^*, g(O, \boldsymbol{\omega}) - \bar{g}(\boldsymbol{\theta}_2, \boldsymbol{\omega})\rangle\Big|}_{I_1} \\
&\quad + \underbrace{\Big|\langle \boldsymbol{\omega} - \boldsymbol{\omega}_1^*, g(O, \boldsymbol{\omega}) - \bar{g}(\boldsymbol{\theta}_2, \boldsymbol{\omega})\rangle - \langle \boldsymbol{\omega} - \boldsymbol{\omega}_2^*, g(O, \boldsymbol{\omega}) - \bar{g}(\boldsymbol{\theta}_2, \boldsymbol{\omega})\rangle\Big|}_{I_2}.
\end{aligned}
$$
For the term $I_2$, we simply use the Cauchy-Schwartz inequality to get $2U_\delta\|\boldsymbol{\omega}_1^* - \boldsymbol{\omega}_2^*\|$.
For the term $I_1$, it can be bounded as:
$$
\begin{aligned}
&\Big|\langle \boldsymbol{\omega} - \boldsymbol{\omega}_1^*, g(O, \boldsymbol{\omega}) - \bar{g}(\boldsymbol{\theta}_1, \boldsymbol{\omega})\rangle - \langle \boldsymbol{\omega} - \boldsymbol{\omega}_1^*, g(O, \boldsymbol{\omega}) - \bar{g}(\boldsymbol{\theta}_2, \boldsymbol{\omega})\rangle\Big| \\
&= \Big|\langle \boldsymbol{\omega} - \boldsymbol{\omega}_1^*, \bar{g}(\boldsymbol{\theta}_1, \boldsymbol{\omega}) - \bar{g}(\boldsymbol{\theta}_2, \boldsymbol{\omega})\rangle\Big| \\
&\leq 2R_{\boldsymbol{\omega}}\big\|\bar{g}(\boldsymbol{\theta}_1, \boldsymbol{\omega}) - \bar{g}(\boldsymbol{\theta}_2, \boldsymbol{\omega})\big\| \\
&\leq 2R_{\boldsymbol{\omega}} \cdot 2U_\delta \cdot d_{TV}(\mu_{\boldsymbol{\theta}_1} \otimes \pi_{\boldsymbol{\theta}_1} \otimes \mathcal{P}, \mu_{\boldsymbol{\theta}_2} \otimes \pi_{\boldsymbol{\theta}_2} \otimes \mathcal{P}) \\
&\leq 2U_\delta^2 d_{TV}(\mu_{\boldsymbol{\theta}_1} \otimes \pi_{\boldsymbol{\theta}_1} \otimes \mathcal{P}, \mu_{\boldsymbol{\theta}_2} \otimes \pi_{\boldsymbol{\theta}_2} \otimes \mathcal{P}),
\end{aligned}
$$
where the first inequality is due to Cauchy-Schwartz; the second inequality is by the definition of total variation norm; the third inequality is due to the fact $U_\delta \geq 2R_{\boldsymbol{\omega}}$. Therefore, we have
$$
\begin{aligned}
\big|\Lambda(\boldsymbol{\theta}_1, \boldsymbol{\omega}, O) - \Lambda(\boldsymbol{\theta}_2, \boldsymbol{\omega}, O)\big| &\leq 2U_\delta^2 d_{TV}(\mu_{\boldsymbol{\theta}_1} \otimes \pi_{\boldsymbol{\theta}_1} \otimes \mathcal{P}, \mu_{\boldsymbol{\theta}_2} \otimes \pi_{\boldsymbol{\theta}_2} \otimes \mathcal{P}) + 2U_\delta\|\boldsymbol{\omega}_1^* - \boldsymbol{\omega}_2^*\| \\
&\leq 2U_\delta^2 |\mathcal{A}| L\Big(1 + \lceil\log_\rho m^{-1}\rceil + \frac{1}{1-\rho}\Big)\|\boldsymbol{\theta}_1 - \boldsymbol{\theta}_2\| + 2U_\delta L_*\|\boldsymbol{\theta}_1 - \boldsymbol{\theta}_2\| \\
&= K_1\|\boldsymbol{\theta}_1 - \boldsymbol{\theta}_2\|,
\end{aligned}
$$
where the second inequality is due to Lemma B.1 and Proposition 4.4. $\square$

## E.9 Proof of Lemma D.9

*Proof of Lemma D.9.* By definition,
$$
\begin{aligned}
\big|\Lambda(O, \boldsymbol{\omega}_1, \boldsymbol{\theta}) - \Lambda(O, \boldsymbol{\omega}_2, \boldsymbol{\theta})\big| &= \Big|\langle \boldsymbol{\omega}_1 - \boldsymbol{\omega}^*, g(O, \boldsymbol{\omega}_1) - \bar{g}(\boldsymbol{\omega}_1, \boldsymbol{\theta})\rangle - \langle \boldsymbol{\omega}_2 - \boldsymbol{\omega}^*, g(O, \boldsymbol{\omega}_2) - \bar{g}(\boldsymbol{\omega}_2, \boldsymbol{\theta})\rangle\Big| \\
&\leq \Big|\langle \boldsymbol{\omega}_1 - \boldsymbol{\omega}^*, g(O, \boldsymbol{\omega}_1) - \bar{g}(\boldsymbol{\omega}_1, \boldsymbol{\theta})\rangle - \langle \boldsymbol{\omega}_1 - \boldsymbol{\omega}^*, g(O, \boldsymbol{\omega}_2) - \bar{g}(\boldsymbol{\omega}_2, \boldsymbol{\theta})\rangle\Big| \\
&\quad + \Big|\langle \boldsymbol{\omega}_1 - \boldsymbol{\omega}^*, g(O, \boldsymbol{\omega}_2) - \bar{g}(\boldsymbol{\omega}_2, \boldsymbol{\theta})\rangle - \langle \boldsymbol{\omega}_2 - \boldsymbol{\omega}^*, g(O, \boldsymbol{\omega}_2) - \bar{g}(\boldsymbol{\omega}_2, \boldsymbol{\theta})\rangle\Big| \\
&\leq 2R_{\boldsymbol{\omega}}\big\|(g(O, \boldsymbol{\omega}_1) - g(O, \boldsymbol{\omega}_2)) - (\bar{g}(\boldsymbol{\omega}_1, \boldsymbol{\theta}) - \bar{g}(\boldsymbol{\omega}_2, \boldsymbol{\theta}))\big\| + 2U_\delta\|\boldsymbol{\omega}_1 - \boldsymbol{\omega}_2\|.
\end{aligned}
$$

Note that we have $\|g(O, \boldsymbol{\omega}_1, \boldsymbol{\theta}) - g(O, \boldsymbol{\omega}_2, \boldsymbol{\theta})\| = |(\boldsymbol{\phi}(s') - \boldsymbol{\phi}(s))^\top (\boldsymbol{\omega}_1 - \boldsymbol{\omega}_2)| \le 2\|\boldsymbol{\omega}_1 - \boldsymbol{\omega}_2\|$ and similarly $\|\bar{g}(\boldsymbol{\omega}_1, \boldsymbol{\theta}) - \bar{g}(\boldsymbol{\omega}_2, \boldsymbol{\theta})\| \le |\mathbb{E}[(\boldsymbol{\phi}(s') - \boldsymbol{\phi}(s))^\top (\boldsymbol{\omega}_1 - \boldsymbol{\omega}_2)]| \le 2\|\boldsymbol{\omega}_1 - \boldsymbol{\omega}_2\|$. Therefore,

$$
\begin{aligned}
\left| \Lambda(O, \boldsymbol{\omega}_1, \boldsymbol{\theta}) - \Lambda(O, \boldsymbol{\omega}_2, \boldsymbol{\theta}) \right| &\le 2R_{\boldsymbol{\omega}} \left\| \big(g(O, \boldsymbol{\omega}_1) - g(O, \boldsymbol{\omega}_2)\big) - \big(\bar{g}(\boldsymbol{\omega}_1, \boldsymbol{\theta}) - \bar{g}(\boldsymbol{\omega}_2, \boldsymbol{\theta})\big) \right\| + 2U_\delta \|\boldsymbol{\omega}_1 - \boldsymbol{\omega}_2\| \\
&\le 2R_{\boldsymbol{\omega}} \cdot 4\|\boldsymbol{\omega}_1 - \boldsymbol{\omega}_2\| + 2U_\delta \|\boldsymbol{\omega}_1 - \boldsymbol{\omega}_2\| \\
&\le 6U_\delta \|\boldsymbol{\omega}_1 - \boldsymbol{\omega}_2\|.
\end{aligned}
$$

$\square$

## E.10  Proof of Lemma D.10

*Proof of Lemma D.10.* By the Cauchy-Schwartz inequality and the definition of total variation norm, we have

$$
\begin{aligned}
\mathbb{E}[\Lambda(O_t, \boldsymbol{\omega}_{t-\tau}, \boldsymbol{\theta}_{t-\tau}) - \Lambda(\widetilde{O}_t, \boldsymbol{\omega}_{t-\tau}, \boldsymbol{\theta}_{t-\tau})] &= \mathbb{E}\big[\langle \boldsymbol{\omega}_{t-\tau} - \boldsymbol{\omega}_{t-\tau}^*, g(O_t, \boldsymbol{\omega}_{t-\tau}) - g(\widetilde{O}_t, \boldsymbol{\omega}_{t-\tau}) \rangle\big] \\
&\le 2U_\delta^2 d_{TV}\big(\mathbb{P}(O_t \in \cdot | s_{t-\tau+1}, \boldsymbol{\theta}_{t-\tau}), \mathbb{P}(\widetilde{O}_t \in \cdot | s_{t-\tau+1}, \boldsymbol{\theta}_{t-\tau})\big).
\end{aligned}
$$

(E.3)

The total variation between $O_t$ and $\widetilde{O}_t$ has appeared in (E.2), in the proof of Lemma D.2, which is

$$
d_{TV}\big(\mathbb{P}(O_t \in \cdot | s_{t-\tau+1}, \boldsymbol{\theta}_{t-\tau}), \mathbb{P}(\widetilde{O}_t \in \cdot | s_{t-\tau+1}, \boldsymbol{\theta}_{t-\tau})\big) \le \frac{1}{2}|\mathcal{A}|L \sum_{i=t-\tau}^{t} \mathbb{E}\|\boldsymbol{\theta}_i - \boldsymbol{\theta}_{t-\tau}\|.
$$

Plugging this bound into (E.3), we have

$$
\mathbb{E}\big|\Lambda(O_t, \boldsymbol{\omega}_{t-\tau}, \boldsymbol{\theta}_{t-\tau}) - \Lambda(\widetilde{O}_t, \boldsymbol{\omega}_{t-\tau}, \boldsymbol{\theta}_{t-\tau})\big| \le U_\delta^2 |\mathcal{A}|L \sum_{i=t-\tau}^{t} \mathbb{E}\|\boldsymbol{\theta}_i - \boldsymbol{\theta}_{t-\tau}\|.
$$

$\square$

## E.11  Proof of Lemma D.11

*Proof of Lemma D.11.* We first note that according to the definition in Section C.3,

$$
\mathbb{E}[\Lambda(O_t', \boldsymbol{\omega}_{t-\tau}, \boldsymbol{\theta}_{t-\tau}) | s_{t-\tau+1}, \boldsymbol{\theta}_{t-\tau}] = 0,
$$

where $O_t' = (s_t', a_t', s_{t+1}')$ is the tuple generated by $s_t' \sim \mu_{\boldsymbol{\theta}_{t-\tau}}, a_t' \sim \pi_{\boldsymbol{\theta}_{t-\tau}}, s_{t+1}' \sim \mathcal{P}$. By the ergodicity in Assumption 4.2, it holds that

$$
d_{TV}\big(\mathbb{P}(\widetilde{s}_t = \cdot | s_{t-\tau+1}, \boldsymbol{\theta}_{t-\tau}), \mu_{\boldsymbol{\theta}_{t-\tau}}\big) \le m\rho^{\tau-1}.
$$

It can be shown that

$$
\begin{aligned}
\mathbb{E}[\Lambda(\widetilde{O}_t, \boldsymbol{\omega}_{t-\tau}, \boldsymbol{\theta}_{t-\tau})] &= \mathbb{E}[\Lambda(\widetilde{O}_t, \boldsymbol{\omega}_{t-\tau}, \boldsymbol{\theta}_{t-\tau}) - \Lambda(O_t', \boldsymbol{\omega}_{t-\tau}, \boldsymbol{\theta}_{t-\tau})] \\
&= \mathbb{E}\langle \boldsymbol{\omega}_{t-\tau} - \boldsymbol{\omega}_{t-\tau}^*, g(\widetilde{O}_t, \boldsymbol{\omega}_{t-\tau}) - g(O_t', \boldsymbol{\omega}_{t-\tau}) \rangle \\
&\le 4R_{\boldsymbol{\omega}} U_\delta d_{TV}\big(\mathbb{P}(\widetilde{O}_t = \cdot | s_{t-\tau+1}, \boldsymbol{\theta}_{t-\tau}), \mu_{\boldsymbol{\theta}_{t-\tau}} \otimes \pi_{\boldsymbol{\theta}_{t-\tau}} \otimes \mathcal{P}\big) \\
&\le 2U_\delta^2 d_{TV}\big(\mathbb{P}(\widetilde{s}_t = \cdot | s_{t-\tau+1}, \boldsymbol{\theta}_{t-\tau}), \mu_{\boldsymbol{\theta}_{t-\tau}}\big) \\
&\le 2U_\delta^2 m\rho^{\tau-1}.
\end{aligned}
$$

The third inequality holds because $2R_{\boldsymbol{\omega}} < U_\delta$ and

$$
\begin{aligned}
d_{TV}\big(\mathbb{P}(\widetilde{O}_t = \cdot | s_{t-\tau+1}, \boldsymbol{\theta}_{t-\tau}), \mu_{\boldsymbol{\theta}_{t-\tau}} \otimes \pi_{\boldsymbol{\theta}_{t-\tau}} \otimes \mathcal{P}\big) &= d_{TV}\big(\mathbb{P}((\widetilde{s}_t, \widetilde{a}_t) = \cdot | s_{t-\tau+1}, \boldsymbol{\theta}_{t-\tau}), \mu_{\boldsymbol{\theta}_{t-\tau}} \otimes \pi_{\boldsymbol{\theta}_{t-\tau}}\big) \\
&= d_{TV}\big(\mathbb{P}(\widetilde{s}_t = \cdot | s_{t-\tau+1}, \boldsymbol{\theta}_{t-\tau}), \mu_{\boldsymbol{\theta}_{t-\tau}}\big).
\end{aligned}
$$

This can be shown following the same procedure in (B.1), because $\mathbb{P}(\widetilde{O}_t = \cdot | s_{t-\tau+1}, \boldsymbol{\theta}_{t-\tau}) = \mathbb{P}(\widetilde{s}_t = \cdot | s_{t-\tau+1}, \boldsymbol{\theta}_{t-\tau}) \otimes \pi_{\boldsymbol{\theta}_{t-\tau}} \otimes \mathcal{P}$. $\square$