[Reviews · NeurIPS 2020]

Review 1

Summary and Contributions: This paper analyzes the sample complexity of two time-scale actor-critic (AC) algorithms in a finite time case. The analysis is focused on seeking the proof of required iteration for an AC algorithm to reach the first-order stationary point (i.e., the gradient of the actor is smaller than ɛ). The proof techniques are characterizing the bounds of all estimation errors during the training process (shown in Appendix) and connecting those bounds to the actor’s gradient. The primary contribution is the sample complexity of actor-critic algorithms which could be used to estimate the training time of a reinforcement learning algorithm in several applications.

Strengths: Pros: 1. The finite-time analysis is more close to practical scenarios than infinite time analysis. 2. This paper doesn’t rely on the i.i.d. assumption of the data distribution.

Weaknesses: Cons: 1. This paper is hard to follow for me. I would kindly suggest the author should provide more background knowledge for this paper; otherwise, it’s difficult for the readers from other fields to understand the jargons in this paper. For example, what is Markovian noises. I searched for it on the Internet and got a few different answers. So, I guess it probably shouldn’t be presumed as common knowledge. Another example is a sentence in Line 205: “It is known that the TD limiting point satisfies:”, where I think there should be a reference to support this statement. 2. The derivation of theorems better has an accessible outline or sketch of proof in the main text. Currently, the author directs the readers to read the Appendix without explaining the techniques used for the proofs. For example, in Theorem 4.5, there are no explanations, even simple, for the former three terms. I do read the Appendix when reading mathematical proof, but there should be a rough explanation in the text. Otherwise, it hurts the reading experience as I have to search the explanation in the Appendix before proceeding. Moreover, the reference to the Appendix should be more clear. The current version just refers to Appendix but didn’t specifically point out the section number. 3. The discussion of related works should be elaborated. Without a detailed comparison with related works, it’s difficult to properly position this paper in the literature. I do understand the author has mentioned a lot of relevant works, but as a reader not familiar with the history of analysis to actor-critic algorithms, the limitations or drawbacks of previous works are unclear to me.

Correctness: The followings are the questions: 1. Why does this proof not require the “decouple actor and critic” assumption? It’s not obvious to me, but the author claims it’s the strength of this paper. So, I think it worth more explanation. 2. Why is the Equation 4.1 defined in this way? 3. Remark 4.6 should be elaborated for the following aspects: (i) How does the method relax the requirement for i.i.d. data assumption? (ii) Which term is Markovian noises? Or, what’s the definition of Markovian noises? 4. In Remark 4.8, how is “iterative refinement” used? 5. It would be nice to have a reference to how Corollary 4.9 is derived in the Appendix. 6. In Lemma B.3 (line 555), where is k on the right-hand side? 7. How is Lemma C.2 derived? To me, it’s not obvious.

Clarity: This paper is very hard to follow. I would kindly suggest the author to reorganize the paper and improve the clarity of use of notations. For example, the notations used in Appendix C are confusing to me. Minors: 1. r(s, a)/r(θ) and ɛ_app/ɛ_app(θ) are overloaded. 2. In Appendix C, the forms of h(O, η, ω, θ) and h′(O, θ) are similar. I do understand their uses in the proof, but it would be nice to simplify the notations used for the proof. That should definitely improve the readability. 3. 225: “such that” is repeated

Relation to Prior Work: Overall, I think the analysis presented in this paper might have a great impact on the theory of actor-critic algorithms. However, the paper is difficult to follow and thus the contribution is obscure to me. Thus, at this review phase, I recommend a “Borderline, reject” decision for this paper.

Reproducibility: No

Additional Feedback: Overall, I think the analysis presented in this paper could have a great impact on the theory of actor-critic algorithms. However, the paper is difficult to follow and thus the contribution is obscure to me.


Review 2

Summary and Contributions: This paper considers a two time-scale actor-critic algorithm, in which actor and critic update simultaneously but with different learning rates. The authors provide a non-asymptotic convergence rate under Markovian sampling, in contrast to iid samples considered in prior work.

Strengths: The actor-critic algorithm analyzed in this paper is the classic version---easy to implement and sample efficient, compared to other variants with theoretical performance guarantee. In particular, a finite sample analysis is provided for the Markovian sampling setting. As far as I know this constitutes a clear improvement over the state of the art.

Weaknesses: The contribution in Introduction emphasizes the technical novelty without explanation. Remark 4.8 mainly provides comparison of the results rather than the ‘new technique’ itself. It is still unclear to me what the technique. Also, the analysis appears to be similar to [Y]. It would be good if the authors comment on the difference. [Y] Kaledin, M. , Moulines, E. , Naumov, A. , Tadic, V. and Wai, H.-T. (2020). Finite time analysis of linear two-timescale stochastic approximation with Markovian noise.

Correctness: The result appears interesting and seems to be correct (although I did not go through all the proofs in the appendix).

Clarity: Overall the paper is easy to follow.

Relation to Prior Work: The authors should improve the way related work is reported. For instance, on stochastic bias characterization, [23] uses the technique of Lyapunov analysis, which is quite different from [3]. There are two other papers that should be cited: [X] also studied the convergence of two-timescale AC algorithm; [Y] provides non-asymptotic analysis of two time-scale linear SA under Markovian noise. [X] Zhang, S. , Liu, B. , Yao, H. and Whiteson, S. (2019). Provably convergent two-timescale off-policy actor-critic with function approximation.

Reproducibility: Yes

Additional Feedback: It would be good if authors could comment on where the first term in (4.3) and (4.4) come from. Authors are suggested to add a proof outline for the main results. ====Post-rebuttal update==== Overall the results are interesting and valuable. So I would suggest accepting the paper, though the authors did not fully address my concern of the technical novelty. p3, 130: ‘function r(s,a)’ should be ‘function r’ p6, 216: typo “TD(0) limiting point w satisfies Algorithm 1 contains a …”. P7, 248-249: the description is inconsistent with the statement in Theorem 4.5


Review 3

Summary and Contributions: The paper provides the first finite sample complexity analysis for twotime-scale on-policy actor-critic with non-i.i.d. samples. The algorithm the paper analyzed is more closely related to algorithms practitioners have been using than those analyzed by previous works.

Strengths: The paper is very well written and easy to follow. I checked all the proofs in the appendix and I think they are right. The paper makes a solid theoretical contribution by providing the first sample complexity analysis for twotime-scale on-policy actor-critic with non-i.i.d. samples, which, as far as I know, is novel. The results are significant and relevant to the RL community. I can imagine many possible extensions based on this work.

Weaknesses: 1. I think it is a bit overstated (Line 10 and Line 673) to use the term \epsilon-approximate stationary point of J -- there is still function approximation error as in Theorem 4.5. I think the existence of this function approximation error should be explicitly acknowledged whenever the conclusion about sample complexity is stated. Otherwise readers may have the impression that compatible features (Konda, 2002, Sutton et al 2000) are used to deal with these errors, which are not the case. 2. As shown by Konda (2002) and Sutton et al (2000), compatible features are useful tools to address the function approximation error of the critic. I'm wondering if it's possible to introduce compatible features and TD(1) critic in the finite sample complexity analysis in this paper to eliminate the \epsilon_app term. 3. I feel the analysis in the paper depends heavily on the property of the stationary distribution (e.g., Line 757). I'm wondering if it's possible to conduct a similar analysis for the discounted setting (instead of the average reward setting). Although a discounted problem can be solved by methods for the average reward problem (e.g., discarding each transition w.p. 1 - \gamma, see Konda 2002), solving the discounted problem directly is more common in the RL community. It would be beneficial to have a discussion w.r.t. the discounted objective. 4. Although using advantage instead of q value is more common in practice, I'm wondering if there is other technical consideration for conducting the analysis with advantage instead of q value. 5. The assumption about maximum eigenvalues in Line 215 seems artificial. I can understand this assumption, as well as the projection in Line 8 in Algorithm 1, is mainly used to ensure the boundedness of the critic. However, as in Line 219, R_w indeed depends on \lambda, which we do not know in practice. So it means we cannot implement the exact Algorithm 1 in practice. Instead of using this assumption and the projection, is it possible to use regularization (e.g., ridge) for the critic to ensure it's bounded, as done in asymptotic analysis in Zhang et al (2020)? Also Line 216 is a bit misleading. Only the first half (negative definiteness) is used to ensure solvability. But as far as I know, in policy evaluation setting, we do not need the second half (maximum eigenvalue). 6. Some typos: Line 463 should include \epsilon_app and replace the first + with \leq \epsilon_app (the last term of Line 587) is missing in Line 585 and 586 There shouldn't be (1 - \gamma) in Line 589 In Line 618, there should be no need to introduce the summation from k=0 to t - \tau_t, as the summation from k=\tau_t to t is still used in Line 624. In Line 625, it should be \tau_t instead of \tau In Line 640, I personally think it's not proper to cite [25] (the S & B book) -- that book includes too many. Referring to the definition of w^* should be more easy to follow. In Line 658, it should be ||z_k||^2 In Line 672, \epsilon_app is missing In Line 692, it should be E[....] = 0 In Line 708, there shouldn't be \theta_1, \theta_2, \eta_1, \eta_2 In Line 774, I think expectation is missing in the LHS Konda, V. R. Actor-critic algorithms. PhD thesis, Massachusetts Institute of Technology, 2002. Zhang, S., Liu, B., Yao, H., & Whiteson. Provably Convergent Two-Timescale Off-Policy Actor-Critic with Function Approximation. ICML 2020.

Correctness: Yes

Clarity: Yes

Relation to Prior Work: Yes

Reproducibility: Yes

Additional Feedback: I read the author response and would like to keep my score.


Review 4

Summary and Contributions: The paper under review performs a mathematical analysis of the loss of the A2C algorithm in the case where the actor and critic are simultaneously updated. The analysis is based on a number of assumptions which include the assumption that the critic uses linear function approximation. The main results are Theorem 4.5, Theorem 4.7 and Corollary 4.9. Theorem 4.5 decomposes the gradient of the loss function as a sum of 4 components. Theorem 4.7 characterizes the convergence of the critic. Corollary 4.9 puts the previous two theorems together and bounds the gradient of the loss by epsilon under the assumption that the A2C performed T steps, where T is Õ(epsilon^{-2.5}). In Remark 4.12 authors argue that the estimate in Corollary 4.9 is the lowest one can achieve under the assumptions considered in the paper.

Strengths: A well-written mathematical analysis of A2C.

Weaknesses: It would be good to include a brief description how authors ended up with the 4 summands in the formulation of Theorem 4.5 - I am under impression that now one has to dig into the proof in the appendix to get any intuition. The paper is interesting, though I am not sure to what extent this is a good fit for a conference presentation. The proofs are extending over more than 20 pages and they are marked as “sketches”.

Correctness: The appendix contains sketches of proofs. The paper does not contain any experiments.

Clarity: The paper is well-written, though I am afraid that the current presentation will satisfy only needs of readers interested in precise mathematical reasoning. Those interested in applications of RL likely will have hard time understanding merits of the paper.

Relation to Prior Work: The prior work is carefully discussed.

Reproducibility: Yes

Additional Feedback: A minor improvement of presentation may consist in inclusion of a more abstract version of A2C (without the specifics of the approximators used by the authors) side by side with the version considered in the mathematical analysis. In practice (e.g. in https://github.com/openai/baselines/blob/master/baselines/a2c/a2c.py#L68) one considers also a joint actor-critic loss. I wonder whether the analysis holds for this case. The rebuttal does not change my opinion about the paper.

[Author Response · NeurIPS 2020]

**To R1, R2, R4:** *About the proof sketch:* Due to the 8 pages limit, we had to put the proof sketch in Appendix A (Page
12-14) and use the main text to highlight the contribution. In the camera ready version, we can reorganize and move the
proof sketch into the main text, and also add references/pointers to the specific section in the appendix.

**To R1:** *The paper is hard to follow...:* We apologize we did not provide enough background. We will provide more
explanation to those terminologies in the revision.

*Contribution compared with previous works:* The main contributions of our work include: (1) We are the first to provide
a finite-time analysis for a practical actor-critic algorithm introduced in [25]; (2) Our analysis does not require the
unrealistic i.i.d. assumption and directly deals with the Markov decision process; (3) Our analysis provides a better
sample complexity $\mathcal{O}(\epsilon^{-2.5})$ than the best-known result $\mathcal{O}(\epsilon^{-4})$ in previous work under strong assumptions [16,21].

1.Why decoupled actor-critic assumption? The decoupled actor-critic is not an assumption, but a different algorithm
appearing in [16] and [21]. It is not a practical algorithm, but easier to analyze. Our two time-scale actor-critic algorithm
is more realistic and more sample efficient.

2.What is $\tau_t$? $\tau_t$ is the mixing time of the Markov chain, which characterizes the time it takes the ergodic Markov chain
in Assumption 4.2 to converge to its stationary distribution.

3.What is Markovian noise? Both actor and critic are updated based on observation tuples $\{O_t = (s_t, a_t, s_{t+1})\}_{t=0,1,...}$.
In previous work [16, 21], they assume that each tuple is sampled i.i.d. in order to simplify the analysis. However, this
is obviously not true in practice. In this paper, we follow [3] and directly deal with the true data which are sampled
from the Markov decision process. We refer to this setting as the "Markovian noise" setting.

4.How is "iterative refinement" used? The "iterative refinement" is not used in our paper. It is used in [36] and more
details can be found therein.

5.Proof of Corollary 4.9: The proof is in Section C.4, line 667.

6.Proof of Lemma B.3: This is a typo. All $a_i$ should be $a_k$.

7.Proof of Lemma C.2 The proof is as follows:
$$\left\| \Delta h(O, \eta, \boldsymbol{\omega}, \boldsymbol{\theta}) \right\|^2 := \left( \eta(\boldsymbol{\theta}) - \eta + (\boldsymbol{\phi}(s') - \boldsymbol{\phi}(s))^\top (\boldsymbol{\omega} - \boldsymbol{\omega}^*) \right)^2 \cdot \left\| \nabla \log \pi_{\boldsymbol{\theta}}(a|s) \right\|^2$$
$$\leq \left[ 2(\eta(\boldsymbol{\theta}) - \eta)^2 + 2\left( (\boldsymbol{\phi}(s') - \boldsymbol{\phi}(s))^\top (\boldsymbol{\omega} - \boldsymbol{\omega}^*) \right)^2 \right] B^2$$
$$\leq \left[ 2(\eta(\boldsymbol{\theta}) - \eta)^2 + 2\left\| \boldsymbol{\phi}(s') - \boldsymbol{\phi}(s) \right\|^2 \|\boldsymbol{\omega} - \boldsymbol{\omega}^*\|^2 \right] B^2 \leq \left[ 2(\eta(\boldsymbol{\theta}) - \eta)^2 + 2 \cdot 4 \cdot \|\boldsymbol{\omega} - \boldsymbol{\omega}^*\|^2 \right] B^2,$$

where the equality is by the definition of $\Delta h(O, \eta, \boldsymbol{\omega}, \boldsymbol{\theta})$, the first inequality is by $(a+b)^2 \leq 2a^2 + 2b^2$ and Assumption
4.3(a), the second inequality is by Cauchy-Schwartz, and the last inequality is by triangle inequality and $\|\boldsymbol{\phi}(s)\| \leq 1$.

**To R2:** *Technical novelty in Remark 4.8:* Take the estimation error $\mathbf{z}_t$ as an example, the inequality at line 642 involves
bounding $\mathbb{E}\|\mathbf{z}_t\|^2$ with $\mathbb{E}\|\mathbf{z}_t\|$ at its right hand side. Directly unrolling the equation at line 642 yields a loose result and
a complicated proof, as done in [36]. We find it is viable to postpone the unrolling and compute the average estimation
error which can give a tighter bound. We will elaborate it in Remark 4.8 in the revision.

*Technical difference with [Y]:* Thanks for pointing out the related works which we were not aware of previously. We
will compare with them in the revision. The problem settings of both works are very different. In specific, [Y] considers
updates that are linear in the two parameters $\boldsymbol{\theta}_t$ and $\boldsymbol{\omega}_t$. In contrast, the actor-critic updates in our paper is not a linear
function of $\boldsymbol{\theta}_t$ and $\boldsymbol{\omega}_t$, i.e., the policy gradient update for $\boldsymbol{\theta}_t$. So the analysis in [Y] is not directly applicable to our
setting, and our analysis on the actor $\boldsymbol{\theta}_t$ update requires a very different approach.

*Where the first term in (4.3) and (4.4) come from?* The first term in (4.3) comes from the term $I_1$ at line 648, which is
related to the estimation error of last iterate $\boldsymbol{\omega}_t$. Similarly, the term in (4.4) comes from $I_1$ at line 615, which is related
to the estimation error of the last iterate $\eta_t$ for the average reward.

**To R3:** *The $\epsilon$-approximate stationary point:* We will explicitly acknowledge the existence of function approximation
error to avoid any confusion.

*Compatible function approximation:* It is possible to use compatible function approximation instead of a fixed linear
function approximation. The potential difficulty is to efficiently estimate the Q function for a given state-action pair,
which might involve starting another sampling trajectory.

*Discounted setting:* It is possible to extend our analysis to discounted MDPs. As you suggested, we can discarding each
transition w.p. $1 - \gamma$ and restarting the episode. We will add a discussion in the future work section.

*Q-function or Advantage function:* Our analysis is applicable to both advantage function $\Delta(s, a)$ and Q-function, with a
very minor change in the analysis. We use the advantage function just following the convention of practice.

*Regularized critic:* Thank you for pointing out the related work [ICML2020] and suggesting this very promising idea.
We will comment on this work and study the regularized critic in our future work.

**To R4:** *The proofs are extending over more than 20 pages and they are marked as "sketches":* This is a misunderstanding.
To clarify, we actually have both sketches of proofs (Appendix A, pp. 12-14) and detailed proofs (Appendix C). So it is
not the sketch that is lengthy.

*About Theorem 4.5* The first term is the linear approximation error; the second term is from upper bounding the
performance function; the third term is due to the stochastic variance and Markovian noise; and the last term is the
critic's error. More details can be found in the proof sketch, at line 446.

*Joint loss:* In Open AI's implementation of A2C, the gradients of the joint loss w.r.t. the actor and the critic can actually
be separated, so our analysis still holds.

[Meta-Review · NeurIPS 2020]

It is important to understand the algorithms we aim to apply to many problems of interest. In that light, and given the popularity of these algorithms, a finite-time analysis of actor-critic methods is a nice contribution to the field, and we happily accept this paper for publication at NeurIPS.